# On the Comparison between Multi-modal and Single-modal Contrastive Learning

**Wei Huang**[*][†]
RIKEN AIP
wei.huang.vr@riken.jp

**Andi Han**[*]
RIKEN AIP
andi.han@riken.jp

**Yongqiang Chen**
The Chinese University of Hong Kong
yqchen@cse.cuhk.edu.hk

**Yuan Cao**
The University of Hong Kong
yuancao@hku.hk

**Zhiqiang Xu**[†]
MBZUAI
zhiqiang.xu@mbzuai.ac.ae

**Taiji Suzuki**
University of Tokyo & RIKEN AIP
taiji@mist.i.u-tokyo.ac.jp

## Abstract

Multi-modal contrastive learning with language supervision has presented a paradigm shift in modern machine learning. By pre-training on a web-scale dataset, multi-modal contrastive learning can learn high-quality representations that exhibit impressive robustness and transferability. Despite its empirical success, the theoretical understanding is still in its infancy, especially regarding its comparison with single-modal contrastive learning. In this work, we introduce a feature learning theory framework that provides a theoretical foundation for understanding the differences between multi-modal and single-modal contrastive learning. Based on a data generation model consisting of signal and noise, our analysis is performed on a ReLU network trained with the InfoMax objective function. Through a trajectory-based optimization analysis and generalization characterization on downstream tasks, we identify the critical factor, which is the signal-to-noise ratio (SNR), that impacts the generalizability in downstream tasks of both multi-modal and single-modal contrastive learning. Through the cooperation between the two modalities, multi-modal learning can achieve better feature learning, leading to improvements in performance in downstream tasks compared to single-modal learning. Our analysis provides a unified framework that can characterize the optimization and generalization of both single-modal and multi-modal contrastive learning. Empirical experiments on both synthetic and real-world datasets further consolidate our theoretical findings.

## 1   Introduction

Large-scale pre-trained models have achieved unprecedented success, including GPT series [6, 41], LLaMa [53], among many others. CLIP [42] as a typical example, uses a multi-modal contrastive learning framework to learn from a massive scale of image-caption data. The multi-modal contrastive learning in CLIP has shown significant capabilities to learn high-quality representations, which are ready to be adapted to a wide range of downstream tasks, forming the backbone of generative models

---

[*]Equal Contribution.
[†]Corresponding Author.

38th Conference on Neural Information Processing Systems (NeurIPS 2024).

like DALL-E2 [43], prompt learning [61] as well as general purpose multi-modal agents [62, 35]. Given the huge success of models like CLIP that have stellar zero-shot and few-shot capabilities on a wide range of out-of-distribution (OOD) benchmarks, they have been widely recognized as foundation models (FMs). More similar examples are given by ALIGN [28], Florence [59], BLIP [33], Flamingo [1].

Despite the unprecedented success achieved by multi-modal contrastive learning, the fundamental mechanism that leads to greater performance, especially compared to single-modal contrastive learning is still under-explored. Recently, several seminal works provided theoretical explanations for either single-modal [4, 5, 14, 48, 24, 7, 50, 49, 20] or multi-modal contrastive learning [38, 37, 44, 12]. For example, [56] studied how single-modal contrastive learning learns the feature representations for neural networks by analyzing its feature learning process. As for multi-modal contrastive learning, [12, 58] provided explanations for why multi-modal contrastive learning demonstrates zero-shot transferability, and robustness to distribution shifts, than supervised learning, which offer valuable insights. Although both lines of the existing works provide valid theoretical insights under the respective settings, rare work has compared the optimization and generalization of the two types of contrastive learning under a unified framework. This motivates us to establish a systematic feature learning analysis for both single-modal and multi-modal contrastive learning.

In particular, we consider a data generation model that contains two modalities of data, which are generated from signal and noise features. The signal feature correlates in different modalities, while there is no correlation between noise features among modalities. We then study the optimization of single-modal and multi-modal contrastive learning under gradient descent training. By studying the trajectories of signal learning and noise memorization, we establish the convergence conditions and further characterize the generalization ability in the downstream tasks. The results show that, through the cooperation between modalities, multi-modal contrastive learning can achieve better generalization in the downstream task. In contrast, without the help of the second modality, single-modal contrastive learning concentrates on learning noise from the data, and thus generalizes poorly on the downstream tasks. The main contributions of this work are summarized as follows:

- This work establishes the *first systematic comparative optimization analysis* for single-modal and multi-modal contrastive learning under gradient descent training in non-convex settings. We show that both single-modal and multi-modal can achieve near-zero training error under InfoMax contrastive loss after polynomial number of iterations, by overcoming the non-convex difficulty.

- By a trajectory-based analysis of the signal learning and noise memorization of the ReLU network from the data, we successfully characterize the difference in *generalization* between single-modal and multi-modal contrastive learning. The distinct SNRs of different modalities lead to a divergence in the generalization of downstream tasks for the two contrastive learning frameworks.

- Our theory suggests that the advantage of multi-modal over single-modal contrastive learning comes from the high quality of the second modality and the cooperation between the two modalities through contrastive learning. This divergence is ultimately reflected in the difference in feature learning and the final gap in downstream task generalization. Experimental results on both synthetic and real-world datasets confirm our theoretical findings and understanding.

## 2 Related Work

**Theoretical Understanding of Single-modal Contrastive Learning.** The seminal work [4] started theoretical research on single-modal contrastive learning. They assumed that different positive samples are independently drawn from the same latent class, making a connection to supervised learning. [55] identified two key properties related to the contrastive loss: alignment and uniformity. Alongside, [32] illustrated that predicting auxiliary prediction tasks helps in learning representations effective for downstream prediction tasks, and [52] provided a theoretical analysis of contrastive learning in the multi-view setting. Besides, [51] proposed a theoretical framework to understand contrastive self-supervised learning from an optimization perspective. [21] proposed a loss that performs spectral decomposition on the population augmentation graph and can be succinctly written as a contrastive learning objective on neural net representations. [46] pointed out the importance of inductive biases of the function class and training algorithm in understanding contrastive learning. The most related work to us is the work by [56]. Similar to them, this work studies ReLU networks and considers the signal-noise data model. However, we do not require the adjustable bias term in

the activation function, which plays a critical role in [56]. Furthermore, this work adopts a unified framework to compare with multi-modal contrastive learning, which is out of scope in [56].

**Understanding of Multi-modal Contrastive Learning.** As the multi-modal contrastive learning approaches such as CLIP received great success, recent works have been proposing explanations from empirical perspective. [36] empirically showed that high train-test similarity is insufficient to explain CLIP's OOD performance. [60] illustrated that CLIP behaves similarly to Bags-of-words in language-based image retrieval, i.e., the order of words in the input sentence does not largely affect CLIP to find the corresponding image. Besides, [16] demonstrated that training data diversity and the ability to leverage the diversity as supervised learning is the key to the effective robustness of CLIP. Theoretically, [13] proved that multi-modal contrastive learning can block-identify latent factors shared between modalities by the a generative data model. [44] analyzed the training dynamics of a simple multi-modal contrastive learning model and show that contrastive pairs are important for the model to efficiently balance the learned representations. Furthermore, [38] showed that each step of loss minimization by gradient descent can be seen as performing SVD on a contrastive cross-covariance matrix. Similar to us, [25] tried to answer why multi-modal learning is better than single model learning. However, they did not consider contrastive learning and thus cannot explain the success of multi-modal contrastive multi-modal learning.

**Data Quality Matters for Multi-modal Contrastive Learning.** Aligned with our theoretical results, there is a lot of empirical evidence showing that improving the alignment quality with more descriptive captions improves multi-modal contrastive learning. [16] show that the training distribution mostly determines the generalizability of CLIP. Furthermore, [47, 40, 18, 17] find filtering poorly aligned image-caption samples used for training leads to further improvements. Besides, [45, 39, 15] demonstrate that improving the descriptiveness of the captions could further boost the performance of CLIP. Besides, [34] demonstrated that the caused by a combination of model initialization and contrastive learning optimization. However, their results do not take neural network architecture into consideration, and do not provide an analysis of test errors either.

## 3 Problem Setting

**Notation.** We use bold-faced letters for vectors and matrices otherwise representing scalar. We use $\|\cdot\|_2$ to denote the Euclidean norm of a vector or the spectral norm of a matrix, while denoting $\|\cdot\|_F$ as the Frobenius norm of a matrix. For a neural network, we denote $\sigma(\cdot)$ as the activation function and we adopt ReLU activation where $\sigma(x) = \max\{0, x\}$ in this work. To simplify, we denote $[n] = \{1, 2, \ldots, n\}$.

**Data Model.** In this work, we consider the following data model, which consists of signal and noise. In the first modality, example $(\mathbf{x}, y) \sim \mathcal{D}$ is generated as follows:

$$\mathbf{x} = [\mathbf{x}^{(1)^\top}, \mathbf{x}^{(2)^\top}]^\top = [y\boldsymbol{\mu}^\top, \boldsymbol{\xi}^\top]^\top, \quad y \sim \text{unif}(\{-1, 1\}). \tag{1}$$

where $\mathbf{x} \in \mathbb{R}^{2d}$ is the input feature and $y \in \{-1, 1\}$ is the corresponding label generated from Rademacher distribution. In particular, $\mathbf{x}^{(1)} = y\boldsymbol{\mu} \in \mathbb{R}^d$ is the task-relevant signal vector, and $\mathbf{x}^{(2)} = \boldsymbol{\xi} \sim \mathcal{N}(\mathbf{0}, \sigma_\xi^2 \mathbf{I}) \in \mathbb{R}^d$ is the task-irrelevant noise vector. Intuitively, if a network learns primarily from signal, it can effectively generalize to unseen data and vice versa. Similar data models have been adopted in recent theoretical works on supervised learning [2, 26, 8, 23, 31, 63, 22, 11] and self-supervised learning [56, 50, 30].

Similarly for the second modality, a sample $(\widetilde{\mathbf{x}}, y) \sim \widetilde{\mathcal{D}}$ is generated as

$$\widetilde{\mathbf{x}} = [\widetilde{\mathbf{x}}^{(1)\top}, \widetilde{\mathbf{x}}^{(2)\top}]^\top = [y\widetilde{\boldsymbol{\mu}}^\top, \widetilde{\boldsymbol{\xi}}^\top]^\top, \quad y \sim \text{unif}(\{-1, 1\}), \tag{2}$$

where the input feature $\widetilde{\mathbf{x}} \in \mathbb{R}^{2\widetilde{d}}$ and the label $y$ is shared with the first modality. Besides, the signal is a given vector $\widetilde{\boldsymbol{\mu}} \in \mathbb{R}^{\widetilde{d}}$, and noise follows $\widetilde{\boldsymbol{\xi}} \sim \mathcal{N}(\mathbf{0}, \sigma_{\widetilde{\xi}}^2 \mathbf{I}) \in \mathbb{R}^{\widetilde{d}}$. The linear data models for multi-modal learning have also been studied in previous work [44]. To simplify the analysis, we set $d = \widetilde{d}, \sigma_\xi = \sigma_{\widetilde{\xi}}$. However, we highlight that extensions to deal with unmatched dimension and noise level is possible.

## 3.1 Single-modal Contrastive Learning

We use a single-layer neural network $\mathbf{h} : \mathbb{R}^{2d} \to \mathbb{R}^m$ with ReLU activation as our encoder, where $m$ is the number of neurons, which represents the embedding dimension. More precisely,

$$\mathbf{h}(\mathbf{x}) = [\bar{h}_1(\mathbf{x}), \ldots, \bar{h}_m(\mathbf{x})]^\top \in \mathbb{R}^m, \quad \text{where } \bar{h}_r(\mathbf{x}) = h_r(\mathbf{x}^{(1)}) + h_r(\mathbf{x}^{(2)}), \tag{3}$$

here we let $h_r(\mathbf{x}^{(i)}) = \sigma(\langle \mathbf{w}_r, \mathbf{x}^{(i)} \rangle)$ for $r \in [m]$, $i \in [2]$, and $\sigma(\cdot)$ is the ReLU activation function. We adopt a Gaussian to initialize the weights $\mathbf{w}_r^{(0)} \sim \mathcal{N}(\mathbf{0}, \sigma_0^2 \mathbf{I})$, where $\sigma_0$ severs as the strength.

Given a pair of positive data samples, the contrastive loss function is based on the similarity measure defined as the inner product between the representation of two samples $\mathbf{x}, \mathbf{x}' \in \mathbb{R}^{2d}$:

$$\text{Sim}_{\mathbf{h}}(\mathbf{x}, \mathbf{x}') = \frac{1}{m} \sum_{r=1}^{m} h_r(\mathbf{x}^{(1)}) \text{sg}(h_r(\mathbf{x}'^{(1)})) + \frac{1}{m} \sum_{r=1}^{m} h_r(\mathbf{x}^{(2)}) \text{sg}(h_r(\mathbf{x}'^{(2)})), \tag{4}$$

where the $\text{sg}(\cdot)$ is the stop-gradient operation, which is inspired by recent empirical works [19, 10] and theoretical work studying contrastive learning [56]. Here we define positive sample as

$$\widehat{\mathbf{x}} = [\widehat{\mathbf{x}}^{(1)\top}, \widehat{\mathbf{x}}^{(2)\top}]^\top = [y\boldsymbol{\mu}^\top, \boldsymbol{\xi}^\top + \boldsymbol{\epsilon}^\top]^\top, \quad \boldsymbol{\epsilon} \sim \mathcal{N}(\mathbf{0}, \sigma_\epsilon^2 \mathbf{I}). \tag{5}$$

In particular, we consider the form of augmentation where the signal stays invariant while the noise vector is corrupted with added independent noise. Similar setup has been considered in [57]. We consider the contrastive loss presented as follows:

$$L = -\frac{1}{n} \sum_{i=1}^{n} \log \left( \frac{e^{\text{Sim}_{\mathbf{h}}(\mathbf{x}_i, \widehat{\mathbf{x}}_i)/\tau}}{e^{\text{Sim}_{\mathbf{h}}(\mathbf{x}_i, \widehat{\mathbf{x}}_i)/\tau} + \sum_{j \neq i}^{M} e^{\text{Sim}_{\mathbf{h}}(\mathbf{x}_i, \mathbf{x}_j)/\tau}} \right), \tag{6}$$

where $\tau$ is the temperature parameter, $n$ is the number of training samples, and $M$ is the number of negative pairs. In this work, to efficiently optimize the loss to near zero, we require negative sample pairs do not share the same label, i.e., $y_j \neq y_i$ in (6). Note that this setting is aligned with supervised contrastive learning [29, 27].

We use gradient descent to optimize the contrastive learning loss, which leads to the gradient update:

$$\mathbf{w}_r^{(t+1)} = \mathbf{w}_r^{(t)} - \eta \nabla_{\mathbf{w}_r} L(\mathbf{W}^{(t)}) = \mathbf{w}_r^{(t)} + \frac{\eta}{nm\tau} \sum_{i=1}^{n} (1 - \ell_i'^{(t)}) h_r^{(t)}(\widehat{\mathbf{x}}_i^{(1)}) h'^{(t)}(\mathbf{x}_i^{(1)}) y_i \boldsymbol{\mu}$$

$$+ \frac{\eta}{nm\tau} \sum_{i=1}^{n} (1 - \ell_i'^{(t)}) h_r^{(t)}(\widehat{\mathbf{x}}_i^{(2)}) h_r'^{(t)}(\mathbf{x}_i^{(2)}) \boldsymbol{\xi}_i - \frac{\eta}{nm\tau} \sum_{i=1}^{n} \sum_{j \neq i}^{M} \ell_{i,j}'^{(t)} h_r^{(t)}(\mathbf{x}_j^{(1)}) h'^{(t)}(\mathbf{x}_i^{(1)}) y_i \boldsymbol{\mu}$$

$$- \frac{\eta}{nm\tau} \sum_{i=1}^{n} \sum_{j \neq i}^{M} \ell_{i,j}'^{(t)} h_r^{(t)}(\mathbf{x}_j^{(2)}) h_r'^{(t)}(\mathbf{x}_i^{(2)}) \boldsymbol{\xi}_i, \tag{7}$$

where we denote $\bar{h}_r^{(t)}(\mathbf{x}) = \sigma(\langle \mathbf{w}_r^{(t)}, \mathbf{x} \rangle)$, $\eta$ as the learning rate, and we define the loss derivatives as

$$\ell_i'^{(t)} \triangleq \frac{e^{\text{Sim}_{\mathbf{h}}(\mathbf{x}_i, \widehat{\mathbf{x}}_i)/\tau}}{e^{\text{Sim}_{\mathbf{h}}(\mathbf{x}_i, \widehat{\mathbf{x}}_i)/\tau} + \sum_{j \neq i}^{M} e^{\text{Sim}_{\mathbf{h}}(\mathbf{x}_i, \mathbf{x}_j)/\tau}}, \quad \ell_{i,j}'^{(t)} \triangleq \frac{e^{\text{Sim}_{\mathbf{h}}(\mathbf{x}_i, \mathbf{x}_j)/\tau}}{e^{\text{Sim}_{\mathbf{h}}(\mathbf{x}_i, \widehat{\mathbf{x}}_i)/\tau} + \sum_{j \neq i}^{M} e^{\text{Sim}_{\mathbf{h}}(\mathbf{x}_i, \mathbf{x}_j)/\tau}}. \tag{8}$$

Intuitively, when the similarity between positive pair is high, and the similarity between negative time is low, we can see $\ell_i'^{(t)} \approx 1$ and $\ell_{i,j}'^{(t)} \approx 0$, for $i \in [n]$ and $j \in [M]$. Therefore, the gradient descent in Eq. (7) is close to zero, indicating the near convergence result. Furthermore, from Eq. (7), we observe that the evolution direction of weight is composed of signal vector $\boldsymbol{\mu}$ and noise vectors $\boldsymbol{\xi}_i$ for $i \in [n]$. This observation plays a critical role in our following theoretical analysis.

## 3.2 Multi-modal Contrastive Learning

We use two neural networks $\mathbf{h} : \mathbb{R}^d \to \mathbb{R}^m$ and $\mathbf{g} : \mathbb{R}^{\tilde{d}} \to \mathbb{R}^m$ to encode two input modality $\mathbf{x}$ and $\widetilde{\mathbf{x}}$ respectively. Both neural networks use ReLU activation function. More precisely,

$$\mathbf{h}(\mathbf{x}) = [\bar{h}_1(\mathbf{x}), \ldots, \bar{h}_m(\mathbf{x})]^\top \in \mathbb{R}^m, \quad \text{where } \bar{h}_r(\mathbf{x}) = h_r(\mathbf{x}^{(1)}) + h_r(\mathbf{x}^{(2)})$$

$$\mathbf{g}(\widetilde{\mathbf{x}}) = [\bar{g}_1(\widetilde{\mathbf{x}}), \dots, \bar{g}_m(\widetilde{\mathbf{x}})]^\top \in \mathbb{R}^m, \quad \text{where } \bar{g}_r(\widetilde{\mathbf{x}}) = g_r(\widetilde{\mathbf{x}}^{(1)}) + g_r(\widetilde{\mathbf{x}}^{(2)})$$

we let $h_r(\mathbf{x}^{(i)}) = \sigma(\langle \mathbf{w}_r, \mathbf{x}^{(i)} \rangle)$ and $g_r(\widetilde{\mathbf{x}}^{(i)}) = \sigma(\langle \widetilde{\mathbf{w}}_r, \widetilde{\mathbf{x}}^{(i)} \rangle)$. Here $\sigma(\cdot)$ is the ReLU activation function, $\mathbf{w}_r \in \mathbb{R}^d$ and $\widetilde{\mathbf{w}}_r \in \mathbb{R}^{\widetilde{d}}$ for $r \in [m]$ are the weights in two networks. Given the embedding, the similarity function of the two modalities is defined as

$$\text{Sim}_{\mathbf{h},\mathbf{g}}(\mathbf{x}, \widetilde{\mathbf{x}}) = \frac{1}{m} \sum_{r=1}^m h_r(\mathbf{x}^{(1)}) \text{sg}(g_r(\widetilde{\mathbf{x}}^{(1)})) + \frac{1}{m} \sum_{r=1}^m h_r(\mathbf{x}^{(2)}) \text{sg}(g_r(\widetilde{\mathbf{x}}^{(2)})),$$

$$\text{Sim}_{\mathbf{g},\mathbf{h}}(\widetilde{\mathbf{x}}, \mathbf{x}) = \frac{1}{m} \sum_{r=1}^m g_r(\widetilde{\mathbf{x}}^{(1)}) \text{sg}(h_r(\mathbf{x}^{(1)})) + \frac{1}{m} \sum_{r=1}^m g_r(\widetilde{\mathbf{x}}^{(2)}) \text{sg}(h_r(\mathbf{x}^{(2)})).$$

The two similarity functions defined above are modality-centered with stop-gradient operation applied. The objective function of contrastive multi-modal learning can be expressed as

$$L = -\frac{1}{n} \sum_{i=1}^n \log\Big( \frac{e^{\text{Sim}_{\mathbf{h},\mathbf{g}}(\mathbf{x}_i, \widetilde{\mathbf{x}}_i)/\tau}}{e^{\text{Sim}_{\mathbf{h},\mathbf{g}}(\mathbf{x}_i, \widetilde{\mathbf{x}}_i)/\tau} + \sum_{j \neq i}^M e^{\text{Sim}_{\mathbf{h},\mathbf{g}}(\mathbf{x}_i, \widetilde{\mathbf{x}}_j)/\tau}} \Big)$$
$$-\frac{1}{n} \sum_{i=1}^n \log\Big( \frac{e^{\text{Sim}_{\mathbf{g},\mathbf{h}}(\widetilde{\mathbf{x}}_i, \mathbf{x}_i)/\tau}}{e^{\text{Sim}_{\mathbf{g},\mathbf{h}}(\widetilde{\mathbf{x}}_i, \mathbf{x}_i)/\tau} + \sum_{j \neq i}^M e^{\text{Sim}_{\mathbf{g},\mathbf{h}}(\widetilde{\mathbf{x}}_i, \mathbf{x}_j)/\tau}} \Big). \tag{9}$$

Same to the single-modal learning whose objective function is governed by Eq. (6), the objective function for multi-modal contrastive learning adopt one positive pair and $M$ negative pairs. Besides, we require the negative pairs do not share the same label. To optimize the objective function (9) for multi-modal learning, gradient descent is applied to train two encoders simultaneously. The gradient descent rule for the first modal network is governed by the following expression.

$$\mathbf{w}_r^{(t+1)} = \mathbf{w}_r^{(t)} - \eta \nabla_{\mathbf{w}_r} L(\mathbf{W}^{(t)}) = \mathbf{w}_r^{(t)} + \frac{\eta}{nm\tau} \sum_{i=1}^n (1 - \ell_i'^{(t)}) g_r^{(t)}(\widetilde{\mathbf{x}}_i^{(1)}) h_r'^{(t)}(\mathbf{x}^{(1)}) y_i \boldsymbol{\mu}$$

$$+ \frac{\eta}{nm\tau} \sum_{i=1}^n (1 - \ell_i'^{(t)}) g_r^{(t)}(\widetilde{\mathbf{x}}_i^{(2)}) h_r'^{(t)}(\mathbf{x}^{(2)}) \boldsymbol{\xi}_i - \frac{\eta}{nm\tau} \sum_{i=1}^n \sum_{j \neq i}^M \ell_{i,j}'^{(t)} g_r^{(t)}(\widetilde{\mathbf{x}}_j^{(1)}) h_r'^{(t)}(\mathbf{x}^{(1)}) y_i \boldsymbol{\mu}$$

$$- \frac{\eta}{nm\tau} \sum_{i=1}^n \sum_{j \neq i}^M \ell_{i,j}'^{(t)} g_r^{(t)}(\widetilde{\mathbf{x}}_j^{(2)}) h_r'^{(t)}(\mathbf{x}^{(2)}) \boldsymbol{\xi}_i. \tag{10}$$

Here with a slight abuse of notation, we use $\ell_i'^{(t)}, \ell_{i,j}'^{(t)}$ to represent the loss derivatives for both modalities. Compared to signal-modal learning, the main difference for the multi-modal learning is that the corresponding embedding is from another modality. The gradient update for the second modality can be derived similarly, which we omit here for clarity.

### 3.3 Downstream Task Evaluation

To evaluate the out-of-distribution generalization of single-modal and multi-modal contrastive learning for downstream task, we consider a test distribution $\mathcal{D}_{\text{test}}$, where a sample $\mathbf{x}_{\text{test}} = [y \cdot \boldsymbol{\nu}^\top, \boldsymbol{\zeta}^\top]^\top \sim \mathcal{D}_{\text{test}}$ is generated as follows. The test signal $\boldsymbol{\nu}$ satisfies $\langle \boldsymbol{\nu}, \boldsymbol{\mu} \rangle = O(\|\boldsymbol{\mu}\|_2^2 d^{-1/2})$ and the test noise follows $\boldsymbol{\zeta} \sim \mathcal{N}(\mathbf{0}, \sigma_\xi^2 \mathbf{I})$ and $y$ follows Rademacher distribution. After the training is complete, we introduce a linear head on top of the learned embedding $\mathbf{h}(\mathbf{x}_{\text{test}})$ for adapting to test distribution, i.e., $f(\mathbf{x}_{\text{test}}) = \langle \mathbf{w}, \mathbf{h}(\mathbf{x}_{\text{test}}) \rangle$. Specifically, we consider the task of classification and define the population 0-1 test error as $L_{\mathcal{D}_{\text{test}}} = \mathbb{P}_{\mathbf{x}_{\text{test}} \sim \mathcal{D}_{\text{test}}} \big[ y f(\mathbf{x}_{\text{test}}) < 0 \big]$.

## 4 Main Results

In this section, we introduce our key theoretical findings that elucidate the optimization and generalization result for both single-modal and multi-modal contrastive learning through the feature learning analysis. We use a trajectory-based analysis for the iterations induced by gradient descent, following a post-training analysis for the performance on the downstream test set. Below we provide the main assumption and main theorems.

**Assumption 4.1.** Let SNR $= \|\boldsymbol{\mu}\|_2/(\sigma_\xi \sqrt{d})$. Assume (1) $d \geq \widetilde{\Omega}(\max\{n^2, n\sigma_0^{-1}\sigma_\xi^{-1}, \sigma_0^{-2}\|\boldsymbol{\mu}\|_2^{-2}\})$. (2) $\eta \leq O(\min\{m\|\boldsymbol{\mu}\|_2^{-2}, nm\sigma_\xi^{-2}d^{-1}\})$. (3) $\sigma_0 \leq \widetilde{O}((\max\{\sigma_\xi\sqrt{d}, \|\boldsymbol{\mu}\|_2\})^{-1})$. (4) $m, n \geq \widetilde{\Omega}(1)$.

(5) $\sigma_\epsilon \leq \min\{\widetilde{\Theta}(\|\boldsymbol{\mu}\|_2), \sigma_\xi/\widetilde{\Omega}(1)\}$. (6) $n \cdot \mathrm{SNR}^2 = \Theta(1)$. (7) $C_\mu\|\boldsymbol{\mu}\|_2 = \|\widetilde{\boldsymbol{\mu}}\|_2$, where $C_\mu \geq 2.66$ is a constant.

(1) We adopt a high dimensional setting to ensure enough over-parameterization. (2,3) The learning rate and the strength of initialization are chosen to make sure the that gradient descent can effectively minimize the contrastive loss. (4) The choice of hidden size $m$ and number of training sample $n$ is to provide adequate concentration. (5) The strength of augmentation is set to keep the similarity between two positive samples. (6) The relation between number of sample and SNR is to distinguish the feature learning process between single-modal and multi-modal contrastive learning. (7) To differentiate single-modal and multi-modal contrastive learning, we introduce a constant $C_\mu$, which enables the cooperation between the two modalities in multi-modal contrastive learning.

**Theorem 4.2** (Single-Modal Contrastive Learning). *Under the single-modal learning setup, suppose Assumption 4.1 holds. Then after $T^* = \widetilde{\Theta}(\eta^{-1}mn\sigma_\xi^{-2}d^{-1} + \eta^{-1}mn\sigma_\xi^{-2}d^{-1}\epsilon^{-1})$, the with probability at least $1 - 1/d$, it holds that (1) Training error $L(T^*) \leq \epsilon$ and (2) Test error at down-stream task $L_{\mathcal{D}_{\mathrm{test}}}(T^*) = \Theta(1)$.*

Theorem 4.2 states that despite the small training error achieved by single-modal contrastive learning, the test error is large in the downstream task.

**Theorem 4.3** (Multi-Modal Contrastive Learning). *Under the single-modal learning setup, suppose Assumption 4.1 holds. Then after $T^* = \widetilde{\Theta}(\eta^{-1}mn\sigma_\xi^{-2}d^{-1} + \eta^{-1}mn\sigma_\xi^{-2}d^{-1}\epsilon^{-1})$, the with probability at least $1 - 1/d$, it holds that (1) Training error $L(T^*) \leq \epsilon$ and (2) Test error at down-stream task $L_{\mathcal{D}_{\mathrm{test}}}(T^*) = o(1)$.*

Theorem 4.3 demonstrates that trained multi-modal contrastive learning can achieve both small training error and downstream test error. Compared to Theorem 4.2, Theorem 4.3 shows that the generalization of multi-modal contrastive learning in downstream tasks is better than single-modal contrastive learning. The reason behind this difference is that the two modalities can cooperate with each other; the higher quality in one modality can boost the feature learning in the target modality, helping to generalize to the downstream task. On the contrary, augmentation often maintains the same SNR as the original data, so single-modal learning hardly benefits from the augmentation and can only memorize the noise from the data, which is not applicable to downstream tasks.

# 5 Proof Roadmap

## 5.1 Proof Sketch for Single Modal Contrastive Learning

The proof is constructed by a optimization analysis followed by a generlization analysis in the downstream task. Through the application of the gradient descent rule outlined in Eq. (7), we observe that the gradient descent iterate $\mathbf{w}_r^{(t)}$ is a linear combination of its random initialization $\mathbf{w}_r^{(0)}$, the signal vector $\boldsymbol{\mu}$ and the noise vectors in the training data $\boldsymbol{\xi}_i$ for $i \in [n]$. Consequently, for $r \in [m]$, the decomposition of weight vector iteration can be expressed:

$$\mathbf{w}_r^{(t)} = \mathbf{w}_r^{(0)} + \gamma_r^{(t)}\|\boldsymbol{\mu}\|_2^{-2}\boldsymbol{\mu} + \sum_{i=1}^{n}\rho_{r,i}^{(t)}\|\boldsymbol{\xi}_i\|_2^{-2}\boldsymbol{\xi}_i, \tag{11}$$

where $\gamma_r^{(t)}$ and $\rho_{r,i}^{(t)}$ serve as coefficients and represent signal learning and noise memorization respectively. Based on the the gradient descent update (7), the iteration of $\gamma_r^{(t)}$ and $\rho_{r,i}^{(t)}$ are given:

**Lemma 5.1** (Single-modal Contrastive Learning). *The coefficients $\gamma_r^{(t)}\rho_{r,i}^{(t)}$ in decomposition (11) satisfy the following equations:*

$$\gamma_r^{(t+1)} = \gamma_r^{(t)} + \frac{\eta}{nm\tau}\sum_{i=1}^{n}\left[(1 - \ell_i'^{(t)})h_r^{(t)}(\widehat{\mathbf{x}}_i^{(1)}) - \sum_{j\neq i}^{M}\ell_{i,j}'^{(t)}h_r^{(t)}(\mathbf{x}_j^{(1)})\right]h_r'^{(t)}(\mathbf{x}_i^{(1)})y_i\|\boldsymbol{\mu}\|_2^2, \tag{12}$$

$$\rho_{r,i}^{(t+1)} = \rho_{r,i}^{(t)} + \frac{\eta}{nm\tau}\left[(1 - \ell_i'^{(t)})h_r(\widehat{\mathbf{x}}_i^{(2)}) - \sum_{j\neq i}^{M}\ell_{i,j}'^{(t)}h_r(\mathbf{x}_j^{(2)})\right]h_r'^{(t)}(\mathbf{x}_i^{(2)})\|\boldsymbol{\xi}_i\|_2^2, \tag{13}$$

*where the initialization $\gamma_r^{(0)}, \rho_{r,i}^{(0)} = 0$.*

Lemma 5.1 tells how the coefficients evolve under gradient descent update. In the following, we introduce a two-stage dynamics to characterize the whole training process based on Eq 12 and Eq 13.

**First Stage: Exponential growth.** During the first stage, we show before $\gamma_r^{(t)}$ or $\rho_{r,i}^{(t)}$ grow to $\Theta(1)$, the embedding (3) is close to zero, suggesting the similarity is bounded by $1 \leq \mathrm{Sim}_{\mathbf{h}}(\mathbf{x}, \mathbf{x}') \leq C_\ell$ for some constant $C_\ell > 1$. The loss derivatives defined in (8) can thus be bounded within some constant range.

*Signal learning.* According to the update for signal learning in (12), we see the propagation can be simplified based on the hard-negative sampling strategy, i.e., the negative pairs do not share the same labels. This suggests the negative term is always zero as $\sum_{i=1}^{n} \sum_{j:y_j \neq y_i}^{M} \sigma(\langle \mathbf{w}_r^{(t)}, y_j \boldsymbol{\mu} \rangle) \sigma'(\langle \mathbf{w}_r^{(t)}, y_i \boldsymbol{\mu} \rangle) = 0$. The resulting update of $\gamma_r^{(t)}$ reduces to $\gamma_r^{(t+1)} = \gamma_r^{(t)} + \frac{\eta}{nm\tau} \sum_{i=1}^{n} (1 - \ell_i'^{(t)}) \sigma(\langle \mathbf{w}_r^{(t)}, y_i \boldsymbol{\mu} \rangle) \sigma'(\langle \mathbf{w}_r^{(t)}, y_i \boldsymbol{\mu} \rangle) y_i \|\boldsymbol{\mu}\|_2^2$. Examining the propagation of $\gamma_r^{(t)}$, we can divide the dynamics into two groups depending on the sign of weight initialization $\langle \mathbf{w}_r^{(0)}, \boldsymbol{\mu} \rangle$. Let $\mathcal{U}_+^{(t)} \triangleq \{r : \langle \mathbf{w}_r^{(t)}, \boldsymbol{\mu} \rangle > 0\}$ and $\mathcal{U}_-^{(t)} \triangleq \{r : \langle \mathbf{w}_r^{(t)}, \boldsymbol{\mu} \rangle < 0\}$. Then for $r \in \mathcal{U}_+^{(0)}$, we can show $\gamma_r^{(t)} \geq 0$ increases exponentially and thus the sign of inner produce stays invariant with $\mathcal{U}_+^{(t)} = \mathcal{U}_+^{(0)}$ for all $t \geq 0$. On the other hand, for $r \in \mathcal{U}_-^{(0)}$, we can show $\gamma_r^{(t)} \leq 0$ and decreases exponentially with $\mathcal{U}_-^{(t)} = \mathcal{U}_-^{(0)}$ for all $t \geq 0$.

*Noise memorization.* Compared to signal learning, the behaviour of noise memorization requires more detailed analysis. This is mainly because the negative pairs can not be eliminated simply based on label difference, as the noise patch $\boldsymbol{\xi}_i$ is generated independent of label $y_i$. In addition, the added noise $\boldsymbol{\epsilon}_i$ by augmentation can also contribute to the noise dynamics. We first show when the noise level $\sigma_\epsilon$ is much smaller compared to $\sigma_\xi$, the dynamics of noise memorization is largely remains unaffected. By the sign of $\langle \mathbf{w}_r^{(t)}, \boldsymbol{\xi}_i \rangle$, we partition the samples into two sets, i.e., $\mathcal{I}_{r,+}^{(t)} = \{i : \langle \mathbf{w}_r^{(t)}, \boldsymbol{\xi}_i \rangle > 0\}$, and $\mathcal{I}_{r,-}^{(t)} = \{i : \langle \mathbf{w}_r^{(t)}, \boldsymbol{\xi}_i \rangle < 0\}$. We can verify for $i \in \mathcal{I}_{r,-}^{(0)}$, the value of $\rho_{r,i}^{(t)}$ stays at zero based on the update (13) with an induction argument. For samples $i \in \mathcal{I}_{r,+}^{(0)}$ with positive initialization, we analyze the noise memorization based on the joint dynamics of samples with the same label. In particular, we define total noise memorization of positive and negative samples respectively as $B_{r,+}^{(t)} \triangleq \sum_{i:y_i=1} (\rho_{r,i}^{(t)} + \langle \mathbf{w}_r^{(0)}, \boldsymbol{\xi}_i \rangle) \mathbb{1}_{i \in \mathcal{I}_{r,+}^{(t)}}$ and $B_{r,-}^{(t)} \triangleq \sum_{i:y_i=-1} (\rho_{r,i}^{(t)} + \langle \mathbf{w}_r^{(0)}, \boldsymbol{\xi}_i \rangle) \mathbb{1}_{i \in \mathcal{I}_{r,+}}^{(t)}$. The update of $\rho_{r,i}^{(t)}$ in (13) then implies the dynamics of $B_{r,+}^{(t)}$ and $B_{r,-}^{(t)}$ as follows

$$B_{r,+}^{(t+1)} \approx B_{r,+}^{(t)} + \frac{\eta \sigma_\xi^2 d}{nm\tau} \Big(B_{r,+}^{(t)} - \frac{1}{2} B_{r,-}^{(t)}\Big), \quad B_{r,-}^{(t+1)} \approx B_{r,-}^{(t)} + \frac{\eta \sigma_\xi^2 d}{nm\tau} \Big(B_{r,-}^{(t)} - \frac{1}{2} B_{r,+}^{(t)}\Big),$$

where the coefficient of $1/2$ appears as a result of the randomness of the sign of initialization. This result suggests, individual $\rho_{r,i:y_i=1}^{(t)}$ cannot grow too slow compared to the $\rho_{r,i:y_i=-1}^{(t)}$. Following a similar induction argument, we are able to show $\rho_{r,i:y_i=1}^{(t)}$ has an exponential growth lower bound. On the other hand, for samples with $y_i = -1$ but with different neuron, we can use the same strategy to show an exponential growth lower bound for some neurons that satisfy the initialization conditions.

**Lemma 5.2.** *Under the Assumption 4.1, let $T_1 = \log\big(20/(\sigma_0 \sigma_\xi \sqrt{d})\big)/\log\big(1 + 0.96 \frac{\eta \sigma_\xi^2 d}{nm\tau}\big)$, we have $\gamma_r^{(t)} = \widetilde{O}(1/\sqrt{n})$ for all $r \in [m]$ and $0 \leq t \leq T_1$ and $\max_r \rho_{r,i}^{(T_1)} = \Omega(1)$ for all $i \in [n]$.*

**Second stage: convergence and scale difference.** At the end of first stage, the noise grows to a constant order while signal learning remains negligible. As a result, the loss derivatives are no longer bounded within some constant range. In the second stage, we aim to show the loss is able to converge to an arbitrarily small value $\epsilon$. Despite the unsupervised learning setup, we are still able to show loss convergence thanks to the hard negative samples. Let $F_0(\mathbf{W}, \mathbf{x}_i) = \mathrm{Sim}(\mathbf{x}_i, \widehat{\mathbf{x}}_i)$ be the similarity to the argumentation and $F_j(\mathbf{W}, \mathbf{x}_i) = \mathrm{Sim}(\mathbf{x}_i, \mathbf{x}_j)$ for $j = 1, ..., M$ be the similarity between the negative pairs. Then we can show there exists some $\mathbf{W}^*$ such that $\langle \nabla F_0(\mathbf{W}^{(t)}, \mathbf{x}_i), \mathbf{W}^* \rangle \geq 2 \log(2M/\epsilon)$ while $\langle \nabla F_j(\mathbf{W}^{(t)}, \mathbf{x}_i), \mathbf{W}^* \rangle \leq \log(2M/\epsilon)$ for all $j = 1, ..., M$. Then we can bound $\langle \nabla L_S(\mathbf{W}^{(t)}), \mathbf{W}^{(t)} - \mathbf{W}^* \rangle \geq \frac{1}{n} \sum_{i=1}^{n} L_i(\mathbf{W}^{(t)}) - \epsilon/2$. This as a result allows to show a monotonic decrease in the loss function as $L(\mathbf{W}^{(t)}) \leq \frac{1}{\eta} (\|\mathbf{W}^{(t)} - \mathbf{W}^*\|_F^2 - \|\mathbf{W}^{(t+1)} - \mathbf{W}^*\|_F^2) + \epsilon$ which guarantees convergence by telescoping over the inequality. Upon the convergence, we can also show

the scale difference obtained at the end of the first stage is maintained, i.e., $\gamma_r^{(t)} = \widetilde{O}(1/\sqrt{n})$ while $\max_r \rho_{r,i}^{(T_1)} = \Omega(1)$ for all $i \in [n]$. This suggests the non-linearly separability for the resulting embeddings and thus the downstream test error is non-vanishing. The formal convergence result is established in Lemma C.18. Combined with the generalization error demonstrated in Appendix C.3, this completes the proof of Theorem 4.2.

## 5.2 Proof Sketch for Multi-Modal Learning

Similar to single-modal contrastive learning, we decompose the decomposition of weight vector iteration for the network in the second modality and subsequently provide a two-stage analysis.

$$\widetilde{\mathbf{w}}_r^{(t)} = \widetilde{\mathbf{w}}_r^{(0)} + \widetilde{\gamma}_r^{(t)} \|\widetilde{\boldsymbol{\mu}}\|_2^{-2} \widetilde{\boldsymbol{\mu}} + \sum_i \widetilde{\rho}_{r,i}^{(t)} \|\widetilde{\boldsymbol{\xi}}_i\|_2^{-2} \widetilde{\boldsymbol{\xi}}_i, \tag{14}$$

where $\widetilde{\gamma}_r^{(t)}$ and $\widetilde{\rho}_{r,i}^{(t)}$ serve as coefficients for the weight decomposition in the text modal.

**Lemma 5.3** (Multi-Modal). *The coefficients* $\gamma_r^{(t)}, \rho_{r,i}^{(t)}, \widetilde{\gamma}_r^{(t)}, \widetilde{\rho}_{r,i}^{(t)}$ *in decomposition (11) satisfy*

$$\gamma_r^{(t+1)} = \gamma_r^{(t)} + \frac{\eta}{nm\tau} \sum_{i=1}^n [(1 - \ell_i'^{(t)})h_r'(y_i\boldsymbol{\mu}) - \sum_{j=1}^M \ell_{i,j}'^{(t)} h_r'(y_i\boldsymbol{\mu})] g_r(y_j\widetilde{\boldsymbol{\mu}}) y_i \|\boldsymbol{\mu}\|_2^2, \tag{15}$$

$$\rho_{r,i}^{(t+1)} = \rho_{r,i}^{(t)} + \frac{\eta}{nm\tau} (1 - \ell_i'^{(t)}) g_r(\widetilde{\boldsymbol{\xi}}_i) h_r'(\boldsymbol{\xi}_i) \|\boldsymbol{\xi}_i\|_2^2 - \frac{\eta}{nm\tau} \sum_{j=1}^M \ell_{i,j}'^{(t)} g_r(\widetilde{\boldsymbol{\xi}}_j) h_r'(\boldsymbol{\xi}_i) \|\boldsymbol{\xi}_i\|_2^2, \tag{16}$$

$$\widetilde{\gamma}_r^{(t+1)} = \widetilde{\gamma}_r^{(t)} + \frac{\eta}{nm\tau} \sum_{i=1}^n [(1 - \ell_i'^{(t)})h_r'(y_i\boldsymbol{\mu}) - \sum_{j=1}^M \ell_{i,j}'^{(t)} h_r'(y_i\boldsymbol{\mu})] g_r(y_j\widetilde{\boldsymbol{\mu}}) y_i \|\widetilde{\boldsymbol{\mu}}\|_2^2, \tag{17}$$

$$\widetilde{\rho}_{r,i}^{(t+1)} = \widetilde{\rho}_{r,i}^{(t)} + \frac{\eta}{nm\tau} (1 - \ell_i'^{(t)}) h_r(\boldsymbol{\xi}_i) g_r'(\widetilde{\boldsymbol{\xi}}_i) \|\widetilde{\boldsymbol{\xi}}_i\|_2^2 - \frac{\eta}{nm\tau} \sum_{j=1}^M \ell_{i,j}'^{(t)} h_r(\boldsymbol{\xi}_j) g_r'(\widetilde{\boldsymbol{\xi}}_i) \|\widetilde{\boldsymbol{\xi}}_i\|_2^2, \tag{18}$$

*where the initialization satisfy* $\gamma_r^{(0)}, \rho_{r,i}^{(0)}, \widetilde{\gamma}_r^{(0)}, \widetilde{\rho}_{r,i}^{(0)} = 0$

**First Stage: Exponential growth** The first stage of multi-modal learning shares similar characteristics as single-modal learning.

*Signal learning.* For signal learning, we analyze the trajectories for both $\gamma_r^{(t)}$ and $\widetilde{\gamma}_r^{(t)}$. To this end, we partition the neurons depending on their initialization status. Apart from $\mathcal{U}_+^{(t)}$ and $\mathcal{U}_-^{(t)}$ defined in the single-modal learning, we additionally define $\widetilde{\mathcal{U}}_+^{(t)} \triangleq \{r : \langle \widetilde{\mathbf{w}}_r^{(t)}, \widetilde{\boldsymbol{\mu}} \rangle > 0\}, \widetilde{\mathcal{U}}_-^{(t)} \triangleq \{r : \langle \widetilde{\mathbf{w}}_r^{(t)}, \widetilde{\boldsymbol{\mu}} \rangle < 0\}$ for the other modality. Then for $r \in \mathcal{U}_+^{(0)} \cap \widetilde{\mathcal{U}}_+^{(0)}$, we can show $\gamma_r^{(t)} \geq 0$, $\phi_r^{(t)} \geq 0$ and are increasing. For neurons $r \in \mathcal{U}_-^{(t)} \cap \widetilde{\mathcal{U}}_-^{(t)}$, we can show $\gamma_r^{(t)} \leq 0, \widetilde{\gamma}_r^{(t)} \leq 0$ and are decreasing. Furthermore, the sign of inner product stays invariant, i.e., $\mathcal{U}_+^{(t)} \cap \widetilde{\mathcal{U}}_+^{(t)} = \mathcal{U}_+^{(0)} \cap \widetilde{\mathcal{U}}_+^{(0)}$ and $\mathcal{U}_-^{(t)} \cap \widetilde{\mathcal{U}}_-^{(t)} = \mathcal{U}_-^{(0)} \cap \widetilde{\mathcal{U}}_-^{(0)}$. For neurons with only one of the modalities activated initially, i.e., $r \in \mathcal{U}_+^{(0)} \cap \widetilde{\mathcal{U}}_-^{(0)}$ or $r \in \mathcal{U}_-^{(0)} \cap \widetilde{\mathcal{U}}_+^{(0)}$, we can show there exists some time $t' \geq 0$ such that the neurons are either positively or negatively activated with $r \in \mathcal{U}_+^{(t')} \cap \widetilde{\mathcal{U}}_+^{(t')}$ and $r \in \mathcal{U}_-^{(t')} \cap \widetilde{\mathcal{U}}_-^{(t')}$.

This shows synchronization of the signal learning patterns of two modalities. The pre-synchronization phase reduces the speed of learning and thus we can only focus on neurons with the same sign for the initialization in order to decide the lower and upper bound.

*Noise memorization.* For noise memorization, we partition the samples into two sets for both modalities according to the initialization, namely, $\mathcal{I}_{r,+}^{(t)} = \{i : \langle \mathbf{w}_r^{(t)}, \boldsymbol{\xi}_i \rangle > 0\}, \mathcal{I}_{r,-}^{(t)} = \{i : \langle \mathbf{w}_r^{(t)}, \boldsymbol{\xi}_i \rangle < 0\}$, and similarly for the second modality $\widetilde{\mathcal{I}}_{r,+}^{(t)}, \widetilde{\mathcal{I}}_{r,-}^{(t)}$. Because the noise memorization are correlated in the two modalities, we separately analyze the samples as follows.

(1) For $i \in \mathcal{I}_{r,-}^{(0)} \cap \widetilde{\mathcal{I}}_{r,-}^{(0)}$, we can show $\rho_{r,i}^{(t)} = 0, \mathcal{I}_{r,-}^{(t)} = \mathcal{I}_{r,-}^{(0)}$ and $\widetilde{\rho}_{r,i}^{(t)} = 0, \widetilde{\mathcal{I}}_{r,-}^{(t)} = \widetilde{\mathcal{I}}_{r,-}^{(0)}$.

(2) For $i \in \mathcal{I}_{r,-}^{(0)} \cap \widetilde{\mathcal{I}}_{r,+}^{(0)}$, we can show $\rho_{r,i}^{(t)} = 0, \mathcal{I}_{r,-}^{(t)} = \mathcal{I}_{r,-}^{(0)}$ and $\widetilde{\rho}_{r,i}^{(t)} \leq 0$.

(3) For $i \in \mathcal{I}_{r,+}^{(0)} \cap \widetilde{\mathcal{I}}_{r,-}^{(0)}$, we can show $\widetilde{\rho}_{r,i}^{(t)} = 0, \widetilde{\mathcal{I}}_{r,-}^{(t)} = \widetilde{\mathcal{I}}_{r,-}^{(0)}$ and $\rho_{r,i}^{(t)} \leq 0$.

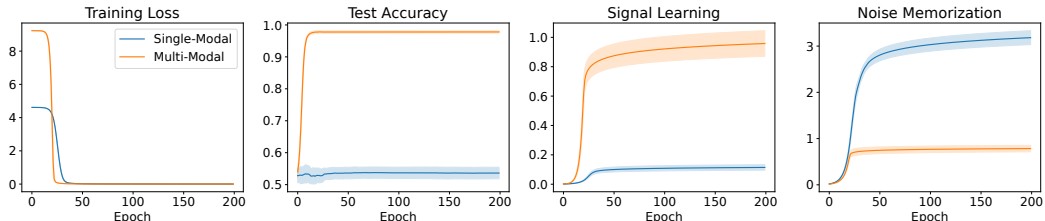

Figure 1: Training loss, test accuracy, signal learning and noise memorization of single-modal and multi-modal contrastive learning.

(4) For $i \in \mathcal{I}_{r,+}^{(0)} \cap \widetilde{\mathcal{I}}_{r,+}^{(0)}$, without loss of generality, we consider upper bounding noise memorization for the first modality and $y_i = 1$. To this end, we first define the individual and joint noise memorization for the first modality as $\Psi_{r,i}^{(t)} \triangleq \rho_{r,i}^{(t)} + \langle \mathbf{w}_r^{(0)}, \boldsymbol{\xi}_i \rangle$, and $B_{r,+,+}^{(t)} \triangleq \sum_{i:y_i=1}(\rho_{r,i}^{(t)} + \langle \mathbf{w}_r^{(0)}, \boldsymbol{\xi}_i \rangle)\mathbb{1}_{i \in \mathcal{I}_{r,+}^{(t)} \cap \widetilde{\mathcal{I}}_{r,+}^{(t)}}$, $B_{r,+,-}^{(t)} \triangleq \sum_{i:y_i=1}(\rho_{r,i}^{(t)} + \langle \mathbf{w}_r^{(0)}, \boldsymbol{\xi}_i \rangle)\mathbb{1}_{i \in \mathcal{I}_{r,+}^{(t)} \cap \widetilde{\mathcal{I}}_{r,-}^{(t)}}$. Similar definitions exist for the other modality. The joint dynamics of noise memorization can be first upper bounded by the other modality as $\widetilde{\Psi}_{r,i}^{(t)} \geq \frac{101C_\ell}{M+1}(B_{r,+,+}^{(t)} + B_{r,+,-}^{(t)})$. Then we can upper bound the individual noise memorization by

$$\Psi_{r,i}^{(t)} \leq (1 + \frac{1.06\eta\sigma_\xi^2 d}{nm\tau})^t(\Psi_{r,i}^{(0)} + \widetilde{\Psi}_{r,i}^{(0)}).$$

We show the combined dynamics of $\gamma_r^{(t)}$ and $\rho_r^{(t)}$ exhibits exponential growth while the magnitude of their difference shrinks exponentially. The results are summarized as follows

**Lemma 5.4.** *Under the Assumption 4.1, let* $T_1 = \log\left(20/(\sigma_0\|\boldsymbol{\mu}\|_2)\right)/\log\left(1 + 0.48C_\mu\frac{\eta\|\boldsymbol{\mu}\|_2^2}{m\tau}\right)$, *we have* $\rho_{r,i}^{(T_1)} = \widetilde{O}(1/\sqrt{n})$ *for all* $r \in [m]$, $i \in [n]$, *and* $0 \leq t \leq T_1$ *and* $\max_r \gamma_r^{(T_1)} = \Omega(1)$.

**Second Stage: Convergence and scale difference.** The second stage presents similar patterns compared to single-modal learning. Thanks to the correlation between the two modality during gradient descent training, the two neural network converge at the same time, minimizing the training loss. Besides, The scale difference at the end of the first stage is carried over throughout the second stage until convergence. Therefore, it allows to show a monotonic decrease in the loss function as $L(\mathbf{W}^{(t)}, \widetilde{\mathbf{W}}^{(t)}) \leq \|\mathbf{W}^{(t)} - \mathbf{W}^*\|_F^2 + \|\widetilde{\mathbf{W}}^{(t)} - \widetilde{\mathbf{W}}^*\|_F^2 - \|\mathbf{W}^{(t+1)} - \mathbf{W}^*\|_F^2 - \|\widetilde{\mathbf{W}}^{(t+1)} - \widetilde{\mathbf{W}}^*\|_F^2 + 2\epsilon$, which guarantees convergence by telescoping over the inequality. At the same time, until convergence, we can show the scale difference obtained at the end of the first stage is maintained, namely $\max_{r,i} \rho_{r,i}^{(t)} = \widetilde{O}(1/\sqrt{n})$ and $\max_r \gamma_r^{(t)} = \Omega(1)$. This suggests the signal learning dominates the noise memorization and thus the resulting embeddings are linearly separable, which guarantees a small test error for downstream tasks. The formal convergence result is established in Lemma D.15. Combined with the generalization error demonstrated in Appendix D.3, this completes the proof of Theorem 4.3.

## 6 Experiments

**Synthetic experiments** We conduct synthetic experiments to verify the theoretical results obtained in the previous sections. We generate samples following the theoretical setups, where we set the data dimension $d = 2000$, number of training samples $n = 100$, number of test samples $n_{\text{test}} = 200$, and the hidden size of all encoders as $m = 50$. We adopt gradient descent with a learning rate of 0.01 as the optimizer to train the model by 200 epochs. In the single-modal setting, the $\boldsymbol{\mu}$ is set to be $[5, 0, ..., 0]^T$ and the $\boldsymbol{\xi} \sim \mathcal{N}(\mathbf{0}, \mathbf{I})$ for the in-distribution data, and the augmentation vector $\boldsymbol{\epsilon} \sim \mathcal{N}(\mathbf{0}, 0.01 * \mathbf{I})$. For the multi-modal setting, $\tilde{\boldsymbol{\mu}} = [0, 15, 0, ..., 0]^T$. In addition, for the OOD test data $\mathbf{x}_{\text{test}} = [\boldsymbol{\nu}^\top, \boldsymbol{\zeta}^\top]^\top$, we set $\boldsymbol{\nu} = [2, 0, ..., 0]$ and $\boldsymbol{\zeta} \sim \mathcal{N}(\mathbf{0}, \mathbf{I})$. We perform logistic regression based on the learned features $\mathbf{h}(\mathbf{x}_{\text{test}})$ and apply the learned classifier head to evaluate OOD generalization error in terms of prediction accuracy.

**Results.** In Figure 1, we see the training loss of both single-modal and multi-modal learning converges rapidly. At the same time, OOD test accuracy of multi-modal learning converges to nearly 1.0 while

that of single-modal learning stagnates around 0.5. This is primarily because under the setup where the other modality $\tilde{\mu}$ has a higher SNR, signal learning of $\mu$ is lifted. This can be verified from the third plot of Figure 1, where the signal learning of multi-modal framework is significantly higher than single-modal. Further, it can be observed that single-modal contrastive learning exhibits more severe noise memorization, which suppresses signal learning. In contrast, multi-modal contrastive learning exhibits less severe noise memorization which would further encourage signal learning. These phenomena again support and align with our theoretical results.

**Real-world experiments**    We now extend the comparison of single-modal and multi-modal learning to realistic image data, ColoredMNIST [3, 54], which is a typical benchmark studying the generalization capability under distribution shifts. The ColoredMNIST dataset is a variation of the standard MNIST dataset, where each digit is assigned a specific color based on its label. The two modalities are image, and text that describes the images. The task is a 10-class classification that recognizes the number of the colored MNIST images. Specifically, we have 10 colors to color 10 digits, and introduce spurious correlations via label noises following the literature:

- For the *training* set, 10% of labels will be clipped to a random class. For images with class '0' (or '1'), they will be colored as red (or green) with a probability of 77.5%, and as another random color with a probability of 22.5%. The coloring scheme introduces a spurious correlation.

- For the *test* set, 10% of labels will be clipped to a random class. For images with class '0' (or '1'), they will be colored as green (or red) with a probability of 77.5%, and as another random color with a probability of 22.5%. The coloring scheme can be considered as reversing the training spurious correlations. Therefore, the evaluation on test set can reflect to what extent the model learns to use the spurious features, i.e., colors, to classify images.

We implement the multi-modal learning following the practice in [54], where we consider an ideal language encoder that successfully encodes the caption of the images into one-hot labels of colors and digits. For single-modal learning, we follow the implementation of the SimCLR [9] to construct a set of augmentations to learn the representations.

**Results.** Under the distribution shift, we verify that multi-modal learning archives an out-of-distribution test accuracy of 82.13%, which outperforms that of single-modal learning 12.68%. As a result, we can claim that the effective SNR of invariant features (the shape of the digit) will be degraded under the impact of the injected

Table 1: Performance comparison for single and multi-modal contrastive learning.

| Model | Train Accuracy | Test Accuracy |
|---|---|---|
| Single-modal | 88.43% | 12.68% |
| Multi-modal | 87.77% | 82.13% |

color. Therefore, the performance of single-modal may be suboptimal as it cannot effectively utilize the information of the digit's shape. On the other hand, multi-modal demonstrates a better capacity for handling this scenario.

## 7    Conclusions

In this work, we have established a comprehensive comparison of the optimization differences during the pre-training stage and the generalization gap between single-modal and multi-modal contrastive learning for downstream tasks. With the cooperation between modalities, multi-modal contrastive learning can achieve better feature learning and generalization on downstream tasks compared to single-modal learning. On the other hand, data augmentation alone can hardly improve data quality and thus cannot boost the performance of single-modal contrastive learning. Together, these results quantitatively demonstrate the superiority of multi-modal learning over single-modal learning and emphasize the importance of data quality in multi-modal contrastive learning.

## Acknowledgments and Disclosure of Funding

We thank the anonymous reviewers for their insightful comments to improve the paper. Wei Huang is supported by JSPS KAKENHI Grant Number 24K20848. Yuan Cao is supported by NSFC 12301657 and HK RGC-ECS 27308624. Taiji Suzuki is partially supported by JSPS KAKENHI (24K02905) and JST CREST (JPMJCR2015).

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

# Appendix

## Contents

# A  Limitations and broader impact

While our theoretical analysis is novel in terms of optimization and generalization, the data model can be further modified to be more practical. Our theoretical analysis may be further used for empirical and theoretical studies of contrastive learning, especially multi-modal contrastive learning. However, we do not foresee a direct social impact from our theory.

# B  Preliminary Lemmas

Before the proof, we introduce lemmas that are useful in proving our main theorem.

**Lemma B.1.** *Let $x \sim \mathcal{N}(0, \sigma^2)$. Then $\mathbb{P}(|x| \leq c) = 2\mathrm{erf}\left(\frac{c}{\sqrt{2}\sigma}\right) \leq 2\sqrt{1 - \exp(-\frac{2c^2}{\sigma^2 \pi})}$.*

*Proof.* The probability density function for $x$ is given by

$$f(x) = \frac{1}{\sqrt{2\pi}\sigma} \exp\left(-\frac{x^2}{2\sigma^2}\right).$$

Then we know that

$$\mathbb{P}(|x| \leq c) = \frac{1}{\sqrt{2\pi}\sigma} \int_{-c}^{c} \exp\left(-\frac{x^2}{2\sigma^2}\right) dx.$$

By the definition of $\mathrm{erf}$ function

$$\mathrm{erf}(c) = \frac{2}{\sqrt{\pi}} \int_0^c \exp(-x^2) dx,$$

and variable substitution yields

$$\mathrm{erf}\left(\frac{c}{\sqrt{2}\sigma}\right) = \frac{1}{\sqrt{2\pi}\sigma} \int_0^c \exp\left(-\frac{x^2}{2\sigma^2}\right) dx.$$

Therefore, we first conclude $\mathbb{P}(|x| \leq c) = 2\mathrm{erf}\left(\frac{c}{\sqrt{2}\sigma}\right)$.

Next, by the inequality $\mathrm{erf}(x) \leq \sqrt{1 - \exp(-4x^2/\pi)}$, we finally obtain

$$\mathbb{P}(|x| \leq c) \leq 2\sqrt{1 - \exp\left(-\frac{2c^2}{\sigma^2 \pi}\right)}.$$

$\square$

Lemma B.1 introduces an anti-concentration result. In later sections, this lemma will be used to show that with a relatively large initialization for the weight vector, some initial properties hold.

**Lemma B.2.** *Under condition that $d \geq \frac{400n}{\sigma_0 \sigma_\xi} \sqrt{\frac{\log(6n/\delta)}{-\pi \log(1 - \delta^2/(4m^2))}}$, and $\widetilde{d} \geq \frac{400n}{\sigma_0 \sigma_{\widetilde{\xi}}} \sqrt{\frac{\log(6n/\delta)}{-\pi \log(1 - \delta^2/(4m^2))}}$, then with probability at least $1 - \delta$, we can show for all $r \in [m]$,*

$$|\langle \mathbf{w}_r^{(0)}, \boldsymbol{\mu} \rangle| \geq 100 \cdot \mathrm{SNR} \sqrt{\frac{8 \log(6n/\delta)}{d}} n,$$

$$|\langle \widetilde{\mathbf{w}}_r^{(0)}, \widetilde{\boldsymbol{\mu}} \rangle| \geq 100 \cdot \mathrm{SNR} \sqrt{\frac{8 \log(6n/\delta)}{\widetilde{d}}} n.$$

*Proof of Lemma B.2.* By Lemma B.1, because $\langle \mathbf{w}_r^{(0)}, \boldsymbol{\mu} \rangle \sim \mathcal{N}(0, \sigma_0^2 \|\boldsymbol{\mu}\|_2^2)$, we can show

$$\mathbb{P}\big(|\langle \mathbf{w}_r^{(0)}, \boldsymbol{\mu} \rangle| \leq c\big) \leq 2\sqrt{1 - \exp\left(-\frac{2c^2}{\sigma_0^2 \|\boldsymbol{\mu}\|_2^2 \pi}\right)}.$$

Let $c = 100 \cdot \text{SNR}\sqrt{\frac{8\log(6n/\delta)}{d}}n = 100n\|\boldsymbol{\mu}\|_2\sigma_\xi^{-1}d^{-1}\sqrt{8\log(6n/\delta)}$ and plug it into the RHS of the above inequality, which becomes:

$$\text{RHS} = 2\sqrt{1 - \exp\left(-\frac{160000\log(6n/\delta)n^2}{\sigma_0^2\sigma_\xi^2 d^2\pi}\right)}.$$

Then we can verify that when $d$ satisfies that $d \geq \frac{400n}{\sigma_0\sigma_\xi}\sqrt{\frac{\log(6n/\delta)}{-\pi\log(1-\delta^2/(4m^2))}}$, it holds that RHS $\leq$ $\delta/m$. This suggests for a single neuron $r \in [m]$, we have $\mathbb{P}(|\langle \mathbf{w}_r^{(0)}, \boldsymbol{\mu}\rangle| \leq c) \leq \delta/m$. Applying union bound, we can show the desired result.

Similarly, with the same procedure, we can prove the result for the other modality.

$$|\langle \widetilde{\mathbf{w}}_r^{(0)}, \widetilde{\boldsymbol{\mu}}\rangle| \geq 100 \cdot \text{SNR}\sqrt{\frac{8\log(6n/\delta)}{\widetilde{d}}}n.$$

$\square$

**Lemma B.3** ([31]). *Let $\mathcal{S}_1 = \{i \in [n] : y_i = 1\}$ and $\mathcal{S}_{-1} = \{i \in [n] : y_i = -1\}$. Then with probability at least $1 - \delta$,*

$$|\mathcal{S}_1|, |\mathcal{S}_{-1}| \in \left[\frac{n}{2} - \sqrt{\frac{n}{2}\log(4/\delta)}, \frac{n}{2} + \sqrt{\frac{n}{2}\log(4/\delta)}\right].$$

Lemma B.3 states that when the label is randomly sampled, the number of positive samples and negative samples is close to $\frac{n}{2}$, adequately.

**Lemma B.4** ([8]). *Suppose that $d \geq \Omega(\log(mn/\delta))$, $m = \Omega(\log(1/\delta))$. Then with probability at least $1 - \delta$, it satisfies that for all $r \in [m], i \in [n]$,*

$$|\langle \mathbf{w}_r^{(0)}, \boldsymbol{\mu}\rangle| \leq \sqrt{2\log(8m/\delta)}\sigma_0\|\boldsymbol{\mu}\|_2$$
$$|\langle \mathbf{w}_r^{(0)}, \boldsymbol{\xi}_i\rangle| \leq 2\sqrt{\log(8mn/\delta)}\sigma_0\sigma_\xi\sqrt{d}$$
$$|\langle \mathbf{w}_r^0, \boldsymbol{\epsilon}_i\rangle| \leq 2\sqrt{\log(8mn/\delta)}\sigma_0\sigma_\epsilon\sqrt{d}.$$

*and for all $i \in [n]$*

$$\sigma_0\|\boldsymbol{\mu}\|_2/2 \leq \max_{r\in[m]}\langle \mathbf{w}_r^{(0)}, \boldsymbol{\mu}\rangle \leq \sqrt{2\log(8m/\delta)}\sigma_0\|\boldsymbol{\mu}\|_2$$
$$\sigma_0\sigma_\xi\sqrt{d}/4 \leq \max_{r\in[m]}\langle \mathbf{w}_r^{(0)}, \boldsymbol{\xi}_i\rangle \leq 2\sqrt{\log(8mn/\delta)}\sigma_0\sigma_\xi\sqrt{d}.$$

**Lemma B.5** ([8]). *Suppose that $\delta > 0$ and $d = \Omega(\log(6n/\delta)))$. Then with probability $1 - \delta$,*

$$\sigma_\xi^2 d/2 \leq \|\boldsymbol{\xi}_i\|_2^2 \leq 3\sigma_\xi^2 d/2,$$
$$|\langle \boldsymbol{\xi}_i, \boldsymbol{\xi}_{i'}\rangle| \leq 2\sigma_\xi^2\sqrt{d\log(6n^2/\delta)}$$
$$|\langle \boldsymbol{\xi}_i, \boldsymbol{\mu}\rangle| \leq \|\boldsymbol{\mu}\|_2\sigma_\xi\sqrt{2\log(6n/\delta)}$$
$$\sigma_\epsilon^2 d/2 \leq \|\boldsymbol{\epsilon}_i\|_2^2 \leq 3\sigma_\epsilon^2 d/2,$$
$$|\langle \boldsymbol{\epsilon}_i, \boldsymbol{\xi}_{i'}\rangle| \leq 2\sigma_\epsilon\sigma_\xi\sqrt{d\log(6n^2/\delta)}$$
$$|\langle \boldsymbol{\epsilon}_i, \boldsymbol{\mu}\rangle| \leq \|\boldsymbol{\mu}\|_2\sigma_\epsilon\sqrt{2\log(6n/\delta)}$$

*for all $i, i' \in [n]$.*

## C  Single-modal Contrastive Learning: Proof of Theorem 4.2

In this section, we provide the proof for Theorem 4.2, which states main results of single modal learning. The training dynamics are based on the coefficient iterations presented in Lemma 5.1. Below, we provide a proof for this lemma:

*Proof of Lemma 5.1.* Recall that the weight decomposition is expressed as

$$\mathbf{w}_r^{(t)} = \mathbf{w}_r^{(0)} + \gamma_r^{(t)}\|\boldsymbol{\mu}\|_2^{-2}\boldsymbol{\mu} + \sum_{i=1}^{n}\rho_{r,i}^{(t)}\|\boldsymbol{\xi}_i\|_2^{-2}\boldsymbol{\xi}_i.$$

We plug it into the gradient descent update as described by Equation 10 yields

$$\mathbf{w}_r^{(t+1)} = \mathbf{w}_r^{(0)} + \gamma_r^{(t+1)}\|\boldsymbol{\mu}\|_2^{-2}\boldsymbol{\mu} + \sum_{i=1}^{n}\rho_{r,i}^{(t+1)}\|\boldsymbol{\xi}_i\|_2^{-2}\boldsymbol{\xi}_i$$

$$= \mathbf{w}_r^{(0)} + \gamma_r^{(t)}\|\boldsymbol{\mu}\|_2^{-2}\boldsymbol{\mu} + \sum_{i=1}^{n}\rho_{r,i}^{(t)}\|\boldsymbol{\xi}_i\|_2^{-2}\boldsymbol{\xi}_i + \frac{\eta}{nm\tau}\sum_{i=1}^{n}(1-\ell_i'^{(t)})h_r^{(t)}(\widehat{\mathbf{x}}_i^{(1)})h'^{(t)}(\mathbf{x}_i^{(1)})y_i\boldsymbol{\mu}$$

$$+ \frac{\eta}{nm\tau}\sum_{i=1}^{n}(1-\ell_i'^{(t)})h_r^{(t)}(\widehat{\mathbf{x}}_i^{(2)})h'^{(t)}(\mathbf{x}_i^{(2)})\boldsymbol{\xi}_i - \frac{\eta}{nm\tau}\sum_{i=1}^{n}\sum_{j\neq i}^{M}\ell_{i,j}'^{(t)}h_r^{(t)}(\mathbf{x}_j^{(1)})h'^{(t)}(\mathbf{x}_i^{(1)})y_i\boldsymbol{\mu}$$

$$- \frac{\eta}{nm\tau}\sum_{i=1}^{n}\sum_{j\neq i}^{M}\ell_{i,j}'^{(t)}h_r^{(t)}(\mathbf{x}_j^{(2)})h'^{(t)}(\mathbf{x}_i^{(2)})\boldsymbol{\xi}_i.$$

By comparing the coefficients in front of $\boldsymbol{\mu}$ and $\boldsymbol{\xi}_i$ on both sides of the equation, we can obtain

$$\gamma_r^{(t+1)} = \gamma_r^{(t)} + \frac{\eta}{nm\tau}\sum_{i=1}^{n}\Big[(1-\ell_i'^{(t)})h_r^{(t)}(\widehat{\mathbf{x}}_i^{(1)}) - \sum_{j\neq i}^{M}\ell_{i,j}'^{(t)}h_r^{(t)}(\mathbf{x}_j^{(1)})\Big]h'^{(t)}(\mathbf{x}_i^{(1)})y_i\|\boldsymbol{\mu}\|_2^2,$$

$$\rho_{r,i}^{(t+1)} = \rho_{r,i}^{(t)} + \frac{\eta}{nm\tau}\Big[(1-\ell_i'^{(t)})h_r(\widehat{\mathbf{x}}_i^{(2)}) - \sum_{j\neq i}^{M}\ell_{i,j}'^{(t)}h_r(\mathbf{x}_j^{(2)})\Big]h'^{(t)}(\mathbf{x}_i^{(2)})\|\boldsymbol{\xi}_i\|_2^2,$$

which completes the proof. $\qquad\square$

According to the behavior of the defined loss derivative (8), we split the entire training dynamics into two phases. In the first stage, the loss derivative remains close to its initial value as the similarity is small from initialization. Later, as the similarity grows to a constant value, the loss derivative is no longer close to the initial value, and the dynamics transition to the second stage. In this stage, the similarity increases logarithmically, and the empirical loss converges.

## C.1 First Stage

In the first stage, the derivative of the loss is close to its initial value because the similarity is small. Below, we provide a useful lemma for establishing such a result.

**Lemma C.1.** *Suppose that $\gamma_r^{(t)} = O(1)$ and $\rho_{r,i}^{(t)} = O(1)$ for all $r \in [m]$ and $i \in [n]$. Under Assumption 4.1, then for any $\delta > 0$, with probability at least $1 - \delta$*

$$|\langle\mathbf{w}_r^{(t)} - \mathbf{w}_r^{(0)}, \boldsymbol{\xi}_i\rangle - \rho_{r,i}^{(t)}| \leq 5\sqrt{\frac{\log(6n^2/\delta)}{d}}n$$

$$|\langle\mathbf{w}_r^{(t)} - \mathbf{w}_r^{(0)}, \boldsymbol{\mu}\rangle - \gamma_r^{(t)}| \leq \mathrm{SNR}\sqrt{\frac{8\log(6n/\delta)}{d}}n$$

*for all $r \in [m]$, $i \in [n]$.*

*Proof of Lemma C.1.* From the signal-noise decomposition of $\mathbf{w}_r^{(t)}$, we derive

$$|\langle\mathbf{w}_r^{(t)} - \mathbf{w}_r^{(0)}, \boldsymbol{\xi}_i\rangle - \rho_{r,i}^{(t)}| \overset{(a)}{=} |\gamma_r^{(t)}\langle\boldsymbol{\mu}, \boldsymbol{\xi}_i\rangle\|\boldsymbol{\mu}\|_2^{-2} + \sum_{i'=1}^{n}\rho_{r,i}^{(t)}\langle\boldsymbol{\xi}_{i'}, \boldsymbol{\xi}_i\rangle\|\boldsymbol{\xi}_{i'}\|_2^{-2}|$$

$$\overset{(b)}{\leq} \|\boldsymbol{\mu}\|_2^{-1}\sigma_\xi\sqrt{2\log(6n/\delta)} + 4\sqrt{\frac{\log(6n^2/\delta)}{d}}n$$

$$\overset{(c)}{\leq} 5\sqrt{\frac{\log(6n^2/\delta)}{d}}n.$$

Equation (a) results from the weight decomposition (see Equation 11). In the first stage, we used the upper bounds for $|\gamma_r^{(t)}|$ and $|\rho_r^{(t)}|$, and applied Lemma B.5 in inequality (b). Finally, inequality (c) follows from the condition $n\text{SNR}^2 = \Theta(1)$.

Further,

$$|\langle \mathbf{w}_r^{(t)} - \mathbf{w}_r^{(0)}, \boldsymbol{\mu}\rangle - \gamma_r^{(t)}| = |\sum_{i=1}^n \rho_{r,i}^{(t)}\|\boldsymbol{\xi}_i\|_2^{-2}\langle \boldsymbol{\xi}_i, \boldsymbol{\mu}\rangle| \leq 2n \cdot \text{SNR}\sqrt{\frac{2\log(6n/\delta)}{d}},$$

where we have used Lemma B.5. $\qquad\square$

Now, we proceed to the lemma concerning the derivative of the loss as follows:

**Lemma C.2.** *If* $\max\{\gamma_r^{(t)}, \rho_{r,i}^{(t)}\} = O(1)$ *and under Assumption 4.1, there exists a constant* $C_\ell > 1$ *such that*

$$\frac{1}{C_\ell(1+M)} \leq \ell_i'^{(t)} \leq \frac{C_\ell}{1+M}, \quad \frac{1}{C_\ell(M+1)} \leq \ell_{i,j}'^{(t)} \leq \frac{C_\ell}{1+M},$$

*for all* $i \in [n]$.

*Proof of Lemma C.2.* From the update of $\mathbf{w}_r^{(t)}$, we have

$$|\langle \mathbf{w}_r^{(t)}, \boldsymbol{\xi}_i\rangle| \overset{(a)}{\leq} |\langle \mathbf{w}_r^{(0)}, \boldsymbol{\xi}_i\rangle| + \rho_{r,i}^{(t)} + 5\sqrt{\frac{\log(6n^2/\delta)}{d}}n$$

$$\overset{(b)}{\leq} 2\sqrt{\log(8mn/\delta)}\sigma_0\sigma_\xi\sqrt{d} + \rho_{r,i}^{(t)} + 5\sqrt{\frac{\log(6n^2/\delta)}{d}}n$$

$$\overset{(c)}{=} O(1),$$

where (a) is by Lemma C.1, and (b) is by Lemma B.4. Finally, in inequality (c) we have used the condition that $\sigma_0 \leq \frac{1}{2\sqrt{\log(8mn/\delta)}\sigma_\xi\sqrt{d}}$ and $d > n^2\log(6n^2/\delta)$ according to Assumption 4.1, and $\max\{\gamma_r^{(t)}, \rho_{r,i}^{(t)}\} = O(1)$. At the same time,

$$|\langle \mathbf{w}_r^{(t)}, \boldsymbol{\mu}\rangle| \overset{(a)}{\leq} |\langle \mathbf{w}_r^{(0)}, \boldsymbol{\mu}\rangle| + \gamma_r^{(t)} + \text{SNR}\sqrt{\frac{8\log(6n/\delta)}{d}}n$$

$$\overset{(b)}{\leq} \sqrt{2\log(8m/\delta)}\sigma_0\|\boldsymbol{\mu}\|_2 + \text{SNR}\sqrt{\frac{8\log(6n/\delta)}{d}}n$$

$$\overset{(c)}{=} O(1),$$

where (a) is by Lemma C.1, (b) is by Lemma B.4, and (c) is by $\sigma_0 \leq \frac{1}{2\sqrt{\log(8m/\delta)}\|\boldsymbol{\mu}\|_2}$ and $d > \text{SNR}^2 n^2\log(6n^2/\delta)$ according to Assumption 4.1, and $\max\{\gamma_r^{(t)}, \rho_{r,i}^{(t)}\} = O(1)$. Besides,

$$|\langle \mathbf{w}_r^{(t)}, \boldsymbol{\epsilon}_i\rangle| = |\langle \mathbf{w}_r^{(0)}, \boldsymbol{\epsilon}_i\rangle + \gamma_r^{(t)}\|\boldsymbol{\mu}\|_2^{-2}\langle \boldsymbol{\mu}, \boldsymbol{\epsilon}_i\rangle + \sum_{i=i}^n \rho_{r,i}^{(t)}\|\boldsymbol{\xi}_i\|_2^{-2}\langle \boldsymbol{\xi}_i, \boldsymbol{\epsilon}_i\rangle|$$

$$\overset{(a)}{\leq} |\langle \mathbf{w}_r^{(0)}, \boldsymbol{\epsilon}_i\rangle| + \|\boldsymbol{\mu}\|_2^{-1}\sigma_\epsilon\sqrt{2\log(6n/\delta)} + 4\sqrt{\frac{\log(6n^2/\delta)}{d}}\sigma_\epsilon\sigma_\xi^{-1}n$$

$$\overset{(b)}{\leq} 2\sqrt{\log(8mn/\delta)}\sigma_0\sigma_\epsilon\sqrt{d} + \|\boldsymbol{\mu}\|_2^{-1}\sigma_\epsilon\sqrt{2\log(6n/\delta)} + 4\sqrt{\frac{\log(6n^2/\delta)}{d}}\sigma_\epsilon\sigma_\xi^{-1}n$$

$$\overset{(c)}{=} O(1),$$

where (a) follows from Lemma B.5, (b) from Lemma B.4, and (c) from the conditions $\sigma_0 \leq \frac{1}{2\sqrt{\log(8mn/\delta)}\sigma_\epsilon\sqrt{d}}$, $\sigma_\epsilon \leq \frac{\|\boldsymbol{\mu}\|_2}{\sqrt{2\log(6n/\delta)}}$, $d > n^2\log(6n^2/\delta)$, and $\sigma_\epsilon < \sigma_\xi$.

Next, we calculate the upper bound of the similarity measure. First, we examine the negative pair. For any $i, j \in [n]$, we have

$$\text{Sim}_{\mathbf{h}}(\mathbf{x}_i, \mathbf{x}_j) = \frac{1}{m} \langle \mathbf{h}(\mathbf{x}_i^{(1)}), \text{sg}(\mathbf{h}(\mathbf{x}_j^{(1)})) \rangle + \frac{1}{m} \langle \mathbf{h}(\mathbf{x}_i^{(2)}), \text{sg}(\mathbf{h}(\mathbf{x}_j^{(2)})) \rangle$$

$$= \frac{1}{m} \sum_{r=1}^{m} \sigma(\langle \mathbf{w}_r^{(t)}, \boldsymbol{\xi}_i \rangle) \sigma(\langle \mathbf{w}_r^{(t)}, \boldsymbol{\xi}_j \rangle) + \frac{1}{m} \sum_{r=1}^{m} \sigma(\langle \mathbf{w}_r^{(t)}, y_i \boldsymbol{\mu} \rangle) \sigma(\langle \mathbf{w}_r^{(t)}, y_j \boldsymbol{\mu} \rangle)$$

$$\leq \max\{|\langle \mathbf{w}_r^{(t)}, y_i \boldsymbol{\mu} \rangle \langle \mathbf{w}_r^{(t)}, y_j \boldsymbol{\mu} \rangle|, |\langle \mathbf{w}_r^{(t)}, \boldsymbol{\xi}_i \rangle \langle \mathbf{w}_r^{(t)}, \boldsymbol{\xi}_j \rangle|\} = O(1).$$

Similarly, for positive pair,

$$\text{Sim}_{\mathbf{h}}(\mathbf{x}_i, \widehat{\mathbf{x}}_j) = \frac{1}{m} \sum_{r=1}^{m} \sigma(\langle \mathbf{w}_r^{(t)}, \boldsymbol{\xi}_i \rangle) \sigma(\langle \mathbf{w}_r^{(t)}, \boldsymbol{\xi}_i + \boldsymbol{\epsilon}_i \rangle) + \frac{1}{m} \sum_{r=1}^{m} \sigma(\langle \mathbf{w}_r^{(t)}, y_i \boldsymbol{\mu} \rangle) \sigma(\langle \mathbf{w}_r^{(t)}, y_i \boldsymbol{\mu} \rangle)$$

$$\leq \max\{|\langle \mathbf{w}_r^{(t)}, y_i \boldsymbol{\mu} \rangle \langle \mathbf{w}_r^{(t)}, y_j \boldsymbol{\mu} \rangle|, |\langle \mathbf{w}_r^{(t)}, \boldsymbol{\xi}_i \rangle \langle \mathbf{w}_r^{(t)}, \boldsymbol{\xi}_i + \boldsymbol{\epsilon}_i \rangle|\} = O(1).$$

According to the above result, we can say that $1 \leq e^{\text{Sim}_{\mathbf{h}}(\mathbf{x}, \mathbf{x}')} \leq C_\ell$, where $C_\ell$ is a positive constant. Then we can provide the upper bound for $\ell'_i$ and $\ell'_{i,j}$

$$\ell_i'^{(t)} = \frac{e^{\text{Sim}_{\mathbf{h}}(\mathbf{x}_i, \widehat{\mathbf{x}}_i)/\tau}}{e^{\text{Sim}_{\mathbf{h}}(\mathbf{x}_i, \widehat{\mathbf{x}}_i)/\tau} + \sum_{j \neq i}^{M} e^{\text{Sim}_{\mathbf{h}}(\mathbf{x}_i, \mathbf{x}_j)/\tau}} \leq \frac{C_\ell}{1 + M},$$

$$\ell_i'^{(t)} = \frac{e^{\text{Sim}_{\mathbf{h}}(\mathbf{x}_i, \widehat{\mathbf{x}}_i)/\tau}}{e^{\text{Sim}_{\mathbf{h}}(\mathbf{x}_i, \widehat{\mathbf{x}}_i)/\tau} + \sum_{j \neq i}^{M} e^{\text{Sim}_{\mathbf{h}}(\mathbf{x}_i, \mathbf{x}_j)/\tau}} \geq \frac{1}{C_\ell(1 + M)},$$

$$\ell_{i,j}'^{(t)} = \frac{e^{\text{Sim}_{\mathbf{h}}(\mathbf{x}_i, \mathbf{x}_j)/\tau}}{e^{\text{Sim}_{\mathbf{h}}(\mathbf{x}_i, \widehat{\mathbf{x}}_i)/\tau} + \sum_{j \neq i}^{M} e^{\text{Sim}_{\mathbf{h}}(\mathbf{x}_i, \mathbf{x}_j)/\tau}} \geq \frac{1}{C_\ell(M + 1)},$$

$$\ell_{i,j}'^{(t)} = \frac{e^{\text{Sim}_{\mathbf{h}}(\mathbf{x}_i, \mathbf{x}_j)/\tau}}{e^{\text{Sim}_{\mathbf{h}}(\mathbf{x}_i, \widehat{\mathbf{x}}_i)/\tau} + \sum_{j \neq i}^{M} e^{\text{Sim}_{\mathbf{h}}(\mathbf{x}_i, \mathbf{x}_j)/\tau}} \leq \frac{C_\ell}{M + 1}.$$

This completes the proof. □

### C.1.1 Dynamics of Signal Learning: Upper Bound

In the first stage, the growth rate of signal learning is exponential. We establish an upper bound for the growth of signal learning.

Then we consider the growth of signal learning coefficient $\gamma_r^{(t)}$. Depending on the initialization, we define $\mathcal{U}_+^{(t)} = \{r \in [m] : \langle \mathbf{w}_r^{(t)}, \boldsymbol{\mu} \rangle > 0\}$ and $\mathcal{U}_-^{(t)} = \{r \in [m] : \langle \mathbf{w}_r^{(t)}, \boldsymbol{\mu} \rangle < 0\}$.

**Lemma C.3.** *Under the condition* $d \geq \frac{400n}{\sigma_0 \sigma_\xi} \sqrt{\frac{\log(6n/\delta)}{-\pi \log(1 - \delta^2/(4m^2))}}$ *and Assumption 4.1, for all* $t \geq 0$, *we have* $\mathcal{U}_+^{(t)} = \mathcal{U}_+^{(0)}$, $\mathcal{U}_-^{(t)} = \mathcal{U}_-^{(0)}$ *and* $\gamma_r^{(t)} > 0$ *is an increasing sequence for all* $r \in \mathcal{U}_+^{(0)}$ *and* $\gamma_r^{(t)} \leq 0$ *and is a decreasing sequence for all* $r \in \mathcal{U}_-^{(0)}$.

*Proof of Lemma C.3.* We prove the claims by induction. To better understand the dynamics, we first derive the propagation for signal learning from the first step. For $r \in \mathcal{U}_+^{(0)}$, i.e., $\langle \mathbf{w}_r^{(0)}, \boldsymbol{\mu} \rangle > 0$, we can see

$$\gamma_r^{(1)} = \gamma_r^{(0)} + \frac{\eta}{nm\tau} \sum_{i=1}^{n} (1 - \ell_i'^{(0)}) \sigma(\langle \mathbf{w}_r^{(0)}, y_i \boldsymbol{\mu} \rangle) \sigma'(\langle \mathbf{w}_r^{(0)}, y_i \boldsymbol{\mu} \rangle) y_i \|\boldsymbol{\mu}\|_2^2$$

$$- \frac{\eta}{nm\tau} \sum_{i=1}^{n} \sum_{j \neq i}^{M} \ell_{i,j}'^{(0)} \sigma(\langle \mathbf{w}_r^{(0)}, y_j \boldsymbol{\mu} \rangle) \sigma'(\langle \mathbf{w}_r^{(0)}, y_i \boldsymbol{\mu} \rangle) y_i \|\boldsymbol{\mu}\|_2^2$$

$$= \gamma_r^{(0)} + \frac{\eta}{nm\tau} \sum_{i:y_i=1}^{n} (1 - \ell_i'^{(0)}) \sigma(\langle \mathbf{w}_r^{(0)}, y_i \boldsymbol{\mu} \rangle) \sigma'(\langle \mathbf{w}_r^{(0)}, y_i \boldsymbol{\mu} \rangle) y_i \|\boldsymbol{\mu}\|_2^2$$

$$-\frac{\eta}{nm\tau}\sum_{i:y_i=1}^{n}\sum_{j:y_j=-1}^{M}\ell_{i,j}'^{(0)}\sigma(\langle \mathbf{w}_r^{(0)},y_j\boldsymbol{\mu}\rangle)\sigma'(\langle \mathbf{w}_r^{(0)},y_i\boldsymbol{\mu}\rangle)y_i\|\boldsymbol{\mu}\|_2^2$$

$$=\frac{\eta}{nm\tau}\sum_{i:y_i=1}^{n}(1-\ell_i'^{(0)})\langle \mathbf{w}_r^{(0)},\boldsymbol{\mu}\rangle\|\boldsymbol{\mu}\|_2^2>0,$$

where last equality is by $\langle \mathbf{w}_r^{(0)},\boldsymbol{\mu}\rangle>0$, $\gamma_r^{(0)}=0$, and the fact that negative samples satisfy $y_j\neq y_i$. Thus, we verify that the sign of $\gamma_r^{(1)}$ follow its initialization and $\gamma_r^{(1)}>0$.

Next, we show the propagation of inner product at $t=1$, for $r\in\mathcal{U}_+^{(0)}$, we have

$$\langle \mathbf{w}_r^{(1)},\boldsymbol{\mu}\rangle\overset{(a)}{\geq}\langle \mathbf{w}_r^{(0)},\boldsymbol{\mu}\rangle+\gamma_r^{(1)}-\text{SNR}\sqrt{\frac{8\log(6n/\delta)}{d}}n$$

$$\overset{(b)}{\geq}0.99\langle \mathbf{w}_r^{(0)},\boldsymbol{\mu}\rangle+\gamma_r^{(1)}>0,$$

where inequality (a) is by Lemma C.1, the second inequality (b) is by Lemma B.2, and the last by $\gamma_r^{(1)}>0$. Hence we verify $\mathcal{U}_+^{(1)}=\mathcal{U}_+^{(0)}$.

Now suppose at iteration $t$, the claims are satisfied, namely $\langle \mathbf{w}_r^{(t)},\boldsymbol{\mu}\rangle>0$ and $\gamma_r^{(t)}\geq\gamma_r^{(t-1)}\geq0$ for $r\in\mathcal{U}_+^{(0)}$. Then following similar argument, for $r\in\mathcal{U}_+^{(0)}$

$$\gamma_r^{(t+1)}=\gamma_r^{(t)}+\frac{\eta}{nm\tau}\sum_{i:y_i=1}^{n}(1-\ell_i'^{(t)})\langle \mathbf{w}_r^{(t)},\boldsymbol{\mu}\rangle\|\boldsymbol{\mu}\|_2^2\geq\gamma_r^{(t)}\geq0,$$

where we use the induction condition that $\langle \mathbf{w}_r^{(t)},\boldsymbol{\mu}\rangle>0$. Further by Lemma C.1 and B.2

$$\langle \mathbf{w}_r^{(t+1)},\boldsymbol{\mu}\rangle\geq0.99\langle \mathbf{w}_r^{(0)},\boldsymbol{\mu}\rangle+\gamma_r^{(t+1)}>0,$$

where we use $\gamma_r^{(t+1)}\geq0$. This completes the induction for $r\in\mathcal{U}_+^{(0)}$.

Similarly, for those neuron $r$ that satisfies $\langle \mathbf{w}_r^0,\boldsymbol{\mu}\rangle<0$, i.e., $r\in\mathcal{U}_-^{(0)}$, we have

$$\gamma_r^{(1)}=\gamma_r^{(0)}+\frac{\eta}{nm\tau}\sum_{i=1}^{n}(1-\ell_i'^{(0)})\sigma(\langle \mathbf{w}_r^{(0)},y_i\boldsymbol{\mu}\rangle)\sigma'(\langle \mathbf{w}_r^{(0)},y_i\boldsymbol{\mu}\rangle)y_i\|\boldsymbol{\mu}\|_2^2$$

$$-\frac{\eta}{nm\tau}\sum_{i=1}^{n}\sum_{j\neq i}^{M}\ell_{i,j}'^{(0)}\sigma(\langle \mathbf{w}_r^{(0)},y_j\boldsymbol{\mu}\rangle)\sigma'(\langle \mathbf{w}_r^{(0)},y_i\boldsymbol{\mu}\rangle)y_i\|\boldsymbol{\mu}\|_2^2$$

$$=-\frac{\eta}{nm\tau}\sum_{i:y_i=-1}^{n}(1-\ell_i'^{(0)})\langle \mathbf{w}_r^{(0)},y_i\boldsymbol{\mu}\rangle\|\boldsymbol{\mu}\|_2^2<0,$$

where the last equality is by $\langle \mathbf{w}_r^{(0)},\boldsymbol{\mu}\rangle<0$, $y_i=-1$, the property of ReLU activation, and the fact that $y_j\neq y_i$ in the negative pair term. Hence we see $\gamma_r^{(1)}\leq\gamma_r^{(0)}=0$.

Similarly, for the inner product at $t=1$,

$$\langle \mathbf{w}_r^{(1)},\boldsymbol{\mu}\rangle=\langle \mathbf{w}_r^{(0)},\boldsymbol{\mu}\rangle+\gamma_r^{(1)}+\sum_{i=1}^{n}\rho_{r,i}^{(t)}\|\boldsymbol{\xi}_i\|_2^{-2}\langle \boldsymbol{\xi}_i,\boldsymbol{\mu}\rangle$$

$$\leq\langle \mathbf{w}_r^{(0)},\boldsymbol{\mu}\rangle+\gamma_r^{(1)}+\text{SNR}\sqrt{\frac{8\log(6n/\delta)}{d}}n$$

$$\leq-0.99|\langle \mathbf{w}_r^{(0)},\boldsymbol{\mu}\rangle|+\gamma_r^{(1)}<0,$$

where the first inequity is by Lemma C.1, the second inequality follows from Lemma B.2, and the last inequality follows from $\gamma_r^{(1)}\leq0$.

Now suppose at iteration $t$, the claims are satisfied, namely $\langle \mathbf{w}_r^{(t)}, \boldsymbol{\mu} \rangle < 0$ and $\gamma_r^{(t)} \leq \gamma_r^{(t-1)} \leq 0$ for $r \in \mathcal{U}_+^{(0)}$. Then following similar argument, for $r \in \mathcal{U}_-^{(0)}$

$$\gamma_r^{(t+1)} = \gamma_r^{(t)} - \frac{\eta}{nm\tau} \sum_{i:y_i=-1}^{n} (1 - \ell_i'^{(t)}) \langle \mathbf{w}_r^{(t)}, \boldsymbol{\mu} \rangle \|\boldsymbol{\mu}\|_2^2 \leq \gamma_r^{(t)} \leq 0,$$

where we use the induction condition that $\langle \mathbf{w}_r^{(t)}, \boldsymbol{\mu} \rangle < 0$. Further

$$\langle \mathbf{w}_r^{(t+1)}, \boldsymbol{\mu} \rangle = \langle \mathbf{w}_r^{(0)}, \boldsymbol{\mu} \rangle + \gamma_r^{(t+1)} + \sum_{i=1}^{n} \rho_{r,i}^{(t)} \|\boldsymbol{\xi}_i\|_2^{-2} \langle \boldsymbol{\xi}_i, \boldsymbol{\mu} \rangle$$

$$\overset{(a)}{\leq} \langle \mathbf{w}_r^{(0)}, \boldsymbol{\mu} \rangle + \gamma_r^{(t+1)} + \text{SNR} \sqrt{\frac{8 \log(6n/\delta)}{d}} n \overset{(b)}{<} 0,$$

where inequality (a) follows from Lemma C.1; we use $\gamma_r^{(t+1)} \leq 0$ and Lemma B.2 in deriving inequality (b). This completes the induction for $r \in \mathcal{U}_-^{(0)}$.

$\square$

With Lemma C.3 at hand, we are ready to demonstrate the upper bound of the growth rate for signal learning.

**Lemma C.4.** *With the same condition as in Lemma C.2 and Lemma C.3 and $n \geq 2500 \log(4/\delta)$, define $A_r^{(t)} = \gamma_r^{(t)} + \langle \mathbf{w}_r^{(0)}, \boldsymbol{\mu} \rangle$ for $r \in \mathcal{U}_+^{(0)}$; and $A_r^{(t)} = -\gamma_r^{(t)} - \langle \mathbf{w}_r^{(0)}, \boldsymbol{\mu} \rangle$ for $r \in \mathcal{U}_-^{(0)}$. With probability at least $1 - \delta$, we have*

$$A_r^{(t)} \leq \left(1 + \frac{0.52\eta\|\boldsymbol{\mu}\|_2^2}{m\tau}\right) A_r^{(0)}.$$

*Proof.* Lemma C.3 suggests that for $r \in [m]$, we can upper bound for $|\gamma_r^{(t)}|$. Without loss of generality, we consider $r \in \mathcal{U}_+^{(0)}$. By Lemma C.1, we can see

$$\langle \mathbf{w}_r^{(t)}, \boldsymbol{\mu} \rangle \leq \gamma_r^{(t)} + \langle \mathbf{w}_r^{(0)}, \boldsymbol{\mu} \rangle + \text{SNR} \sqrt{\frac{8 \log(6n/\delta)}{d}} n$$

$$\leq 1.01 \left(\gamma_r^{(t)} + \langle \mathbf{w}_r^{(0)}, \boldsymbol{\mu} \rangle\right), \tag{19}$$

$$\langle \mathbf{w}_r^{(t)}, \boldsymbol{\mu} \rangle \geq \gamma_r^{(t)} + \langle \mathbf{w}_r^{(0)}, \boldsymbol{\mu} \rangle - \text{SNR} \sqrt{\frac{8 \log(6n/\delta)}{d}} n$$

$$\geq 0.99 \left(\gamma_r^{(t)} + \langle \mathbf{w}_r^{(0)}, \boldsymbol{\mu} \rangle\right), \tag{20}$$

where we use Lemma B.2 and $\gamma_r^{(t)} \geq 0$.

Then, the update equation for $\gamma_r^{(t)}$ follows

$$\gamma_r^{(t+1)} = \gamma_r^{(t)} + \frac{\eta}{nm\tau} \sum_{i:y_i=1}^{n} (1 - \ell_i'^{(t)}) \langle \mathbf{w}_r^{(t)}, \boldsymbol{\mu} \rangle \|\boldsymbol{\mu}\|_2^2.$$

Then we find that

$$A_r^{(t+1)} = A_r^{(t)} + \frac{\eta}{nm\tau} \sum_{i:y_i=1} (1 - \ell_i'^{(t)}) \langle \mathbf{w}_r^{(t)}, \boldsymbol{\mu} \rangle \|\boldsymbol{\mu}\|_2^2$$

$$\overset{(a)}{\leq} A_r^{(t)} + \frac{\eta}{nm\tau} \sum_{i:y_i=1} \left(1 - \frac{1}{C_\ell(M+1)}\right) \|\boldsymbol{\mu}\|_2^2 1.01 A_r^{(t)}$$

$$\overset{(b)}{\leq} A_r^{(t)} + \frac{\eta\|\boldsymbol{\mu}\|_2^2}{m\tau} \left(\frac{1}{2} + \sqrt{\frac{1}{2n} \log(4/\delta)}\right) 1.01 A_r^{(t)}$$

$$\overset{(c)}{\leq} \left(1 + \frac{0.52\eta\|\boldsymbol{\mu}\|_2^2}{m\tau}\right) A_r^{(t)}, \tag{21}$$

where the first inequality (a) is by (19) and Lemma C.2. The second inequality (b) is by Lemma B.3 and $1 - \frac{1}{C_\ell(M+1)} < 1$. The last inequality (c) is by the condition $n \geq 2500 \log(4/\delta)$.

$\square$

### C.1.2 Dynamics of Noise Memorization: Lower Bound

To establish the lower bound of noise memorization in the first stage, we require that the added noise level $\sigma_\epsilon \leq \sigma_\xi/C'$ for sufficiently large, with $C' \geq \widetilde{\Omega}(1)$. We prove the following result that upper bound the scale of $\langle \mathbf{w}_r^{(t)}, \boldsymbol{\epsilon}_i \rangle$.

**Lemma C.5.** *Under the same condition as Lemma C.1 and* $d = \widetilde{\Omega}(\max\{\sigma_0^{-2}\|\boldsymbol{\mu}\|_2^{-2}, n\sigma_0^{-1}\sigma_\xi^{-1}\})$, *there exists a sufficiently large constant* $C_\xi > 0$ *such that*

$$|\langle \mathbf{w}_r^{(0)}, \boldsymbol{\xi}_i \rangle| \geq C_\xi |\langle \mathbf{w}_r^{(t)}, \boldsymbol{\epsilon}_i \rangle|$$

*for all* $i \in [n]$.

*Proof of Lemma C.5.* According to the decomposition of $\mathbf{w}_r^{(t)}$, we can show

$$|\langle \mathbf{w}_r^{(t)}, \boldsymbol{\epsilon}_i \rangle| = |\langle \mathbf{w}_r^{(0)}, \boldsymbol{\epsilon}_i \rangle + \gamma_r^{(t)}\|\boldsymbol{\mu}\|_2^{-2}\langle \boldsymbol{\mu}, \boldsymbol{\epsilon}_i \rangle + \sum_{i'=1}^n \rho_{r,i'}^{(t)}\|\boldsymbol{\xi}_{i'}\|_2^{-2}\langle \boldsymbol{\xi}_{i'}, \boldsymbol{\epsilon}_i \rangle|$$

$$\leq |\langle \mathbf{w}_r^{(0)}, \boldsymbol{\epsilon}_i \rangle| + \gamma_r^{(t)}\|\boldsymbol{\mu}\|_2^{-2}|\langle \boldsymbol{\mu}, \boldsymbol{\epsilon}_i \rangle| + \sum_{i'=1}^n |\rho_{r,i}^{(t)}|\|\boldsymbol{\xi}_{i'}\|_2^{-2}|\langle \boldsymbol{\xi}_{i'}, \boldsymbol{\epsilon}_i \rangle|$$

$$\overset{(a)}{\leq} 2\sqrt{\log(8mn)/\delta}\sigma_0\sigma_\epsilon\sqrt{d} + \|\boldsymbol{\mu}\|_2^{-1}\sigma_\epsilon\sqrt{2\log(6n/\delta)} + 4\sigma_\epsilon\sigma_\xi^{-1}\sqrt{\frac{\log(6n^2/\delta)}{d}}n$$

$$\leq 1/C|\langle \mathbf{w}_r^{(0)}, \boldsymbol{\xi}_i \rangle|,$$

where the second inequality (a) follows from Lemma B.5 and the last inequality is by the following anti-concentration result.

Because $\langle \mathbf{w}_r^{(0)}, \boldsymbol{\xi}_i \rangle$ is a Gaussian random variable with mean zero and variance $\sigma_0\sigma_\xi\sqrt{d}$, by Lemma B.1, we compute

$$\mathbb{P}\big(|\langle \mathbf{w}_r^{(0)}, \boldsymbol{\xi}_i \rangle| \leq c\big) \leq 2\sqrt{1 - \exp\left(-\frac{2c^2}{\pi\sigma_0^2\sigma_\xi^2 d}\right)},$$

When $d \geq \frac{2c^2}{-\log(1-\delta^2/(4n^2))\pi\sigma_0^2\sigma_\xi^2}$, we have $\mathbb{P}\big(|\langle \mathbf{w}_r^{(0)}, \boldsymbol{\xi}_i \rangle| \leq c\big) \leq \delta/n$ and by union bound, we have with probability at least $1-\delta$, it holds $|\langle \mathbf{w}_r^{(0)}, \boldsymbol{\xi}_i \rangle| > c$. Here we let $c = C\big(2\sqrt{\log(8mn)/\delta}\sigma_0\sigma_\epsilon\sqrt{d} + \|\boldsymbol{\mu}\|_2^{-1}\sigma_\epsilon\sqrt{2\log(6n/\delta)} + 4\sigma_\epsilon\sigma_\xi^{-1}\sqrt{\frac{\log(6n^2/\delta)}{d}}n\big) \geq C|\langle \mathbf{w}_r^{(t)}, \boldsymbol{\epsilon}_i \rangle|$. Then we can show the desired result with the condition $d = \widetilde{\Omega}(\max\{\sigma_0^{-2}\|\boldsymbol{\mu}\|_2^{-2}, n\sigma_0^{-1}\sigma_\xi^{-1}\})$.

$\square$

Define that $\mathcal{I}_{r,+}^{(t)} = \{i : \langle \mathbf{w}_r^{(t)}, \boldsymbol{\xi}_i \rangle > 0\}$ and $\mathcal{I}_{r,-}^{(t)} = \{i : \langle \mathbf{w}_r^{(t)}, \boldsymbol{\xi}_i \rangle < 0\}$. To show the result regarding $\mathcal{I}_{r,+}^{(t)}$ and $\mathcal{I}_{r,-}^{(t)}$, we prepare the following anti-concentration result:

**Lemma C.6.** *Under the condition* $d \geq \sqrt{\frac{300\log(6n^2/\delta)}{-\log(1-\delta^2/4n^2)\pi\sigma_0^2\sigma_\xi^2}}$, *then with probability at least* $1 - \delta$, *it satisfies*

$$|\langle \mathbf{w}_r^{(0)}, \boldsymbol{\xi}_i \rangle| > 150\sqrt{\log(6n^2/\delta)/d}.$$

*Proof of Lemma C.6.* Here we want to show $|\langle \mathbf{w}_r^{(0)}, \boldsymbol{\xi}_i \rangle| \geq 150\sqrt{\log(6n^2/\delta)/dn} \overset{\triangle}{=} c$ with high probability. To see this, because $\langle \mathbf{w}_r^{(0)}, \boldsymbol{\xi}_i \rangle$ is a Gaussian random variable with mean zero and

variance $\sigma_0^2 \sigma_\xi^2 d$, by Lemma B.1, we compute

$$\mathbb{P}\big(|\langle \mathbf{w}_r^{(0)}, \boldsymbol{\xi}_i\rangle| \leq c\big) \leq 2\sqrt{1 - \exp\left(-\frac{2c^2}{\pi \sigma_0^2 \sigma_\xi^2 d}\right)},$$

Thus when $d \geq \sqrt{\frac{300 \log(6n^2/\delta)}{-\log(1-\delta^2/4n^2)\pi \sigma_0^2 \sigma_\xi^2}}$, we have $\mathbb{P}\big(|\langle \mathbf{w}_r^{(0)}, \boldsymbol{\xi}_i\rangle| \leq c\big) \leq \delta/n$ and by union bound, we have with probability at least $1 - \delta$, it holds $|\langle \mathbf{w}_r^{(0)}, \boldsymbol{\xi}_i\rangle| > 150\sqrt{\log(6n^2/\delta)/d}$. $\qquad\square$

We then show that neurons with negative inner products with the noise at initialization would stay negative and the corresponding $\rho$ stays zero.

**Lemma C.7.** *Under the same condition as Lemma C.1 and Lemma C.6, for all $t > 0$, we have $\mathcal{I}_{r,-}^{(t)} = \mathcal{I}_{r,-}^{(0)}$ and $\rho_{r,i}^{(t)} = 0$ for all $i \in \mathcal{I}_{r,-}^{(0)}$.*

*Proof of Lemma C.7.* The proof is by induction. We first consider $i \in \mathcal{I}_{r,-}^{(0)}$. At $t = 1$,

$$\rho_{r,i}^{(1)} = \rho_{r,i}^{(0)} + \frac{\eta}{nm\tau}(1 - \ell_i'^{(0)})\sigma(\langle \mathbf{w}_r^{(0)}, \boldsymbol{\xi}_i + \boldsymbol{\epsilon}_i\rangle)\sigma'(\langle \mathbf{w}_r^{(0)}, \boldsymbol{\xi}_i\rangle)\|\boldsymbol{\xi}_i\|_2^2$$

$$- \frac{\eta}{nm\tau}\sum_{j\neq i}^{M} \ell_{i,j}'^{(0)}\sigma(\langle \mathbf{w}_r^{(0)}, \boldsymbol{\xi}_j\rangle)\sigma'(\langle \mathbf{w}_r^{(0)}, \boldsymbol{\xi}_i\rangle)\|\boldsymbol{\xi}_i\|_2^2 = 0,$$

where we have used the condition that $\langle \mathbf{w}_r^{(0)}, \boldsymbol{\xi}_i\rangle < 0$ and the property of ReLU activation.

Next we consider

$$\langle \mathbf{w}_r^{(1)}, \boldsymbol{\xi}_i\rangle = \langle \mathbf{w}_r^{(0)} + \gamma_r^{(1)}\boldsymbol{\mu}\|\boldsymbol{\mu}\|_2^{-2} + \sum_{i=1}^{n} \rho_{r,i}^{(1)}\|\boldsymbol{\xi}_i\|_2^{-2}\boldsymbol{\xi}_i, \boldsymbol{\xi}_i\rangle$$

$$= \langle \mathbf{w}_r^{(0)}, \boldsymbol{\xi}_i\rangle + \gamma_r^{(1)}\langle \boldsymbol{\mu}, \boldsymbol{\xi}_i\rangle\|\boldsymbol{\mu}\|_2^{-2} + \rho_{r,i}^{(1)} + \sum_{i'\neq i}^{n} \rho_{r,i'}^{(1)}\|\boldsymbol{\xi}_{i'}\|_2^{-2}\langle \boldsymbol{\xi}_{i'}, \boldsymbol{\xi}_i\rangle$$

$$\overset{(a)}{\leq} \langle \mathbf{w}_r^{(0)}, \boldsymbol{\xi}_i\rangle + 5\sqrt{\frac{\log(6n^2/\delta)}{d}}n \overset{(b)}{<} 0,$$

where inequality (a) is by Lemma C.1 and $\rho_{r,i}^{(1)} = 0$, and inequality (b) is by Lemma C.6.

Suppose at iteration $t$, the claim is satisfied, i.e., $\langle \mathbf{w}_r^{(t)}, \boldsymbol{\xi}_i\rangle < 0$, $\rho_{r,i}^{(t)} = 0$, for all $i \in \mathcal{I}_{r,-}^{(0)}$. Then

$$\rho_{r,i}^{(t+1)} = \rho_{r,i}^{(t)} + \frac{\eta}{nm\tau}(1 - \ell_i'^{(t)})\sigma(\langle \mathbf{w}_r^{(t)}, \boldsymbol{\xi}_i + \boldsymbol{\epsilon}_i\rangle)\sigma'(\langle \mathbf{w}_r^{(t)}, \boldsymbol{\xi}_i\rangle)\|\boldsymbol{\xi}_i\|_2^2$$

$$- \frac{\eta}{nm\tau}\sum_{j\neq i}^{M} \ell_{i,j}'^{(t)}\sigma(\langle \mathbf{w}_r^{(t)}, \boldsymbol{\xi}_j\rangle)\sigma'(\langle \mathbf{w}_r^{(t)}, \boldsymbol{\xi}_i\rangle)\|\boldsymbol{\xi}_i\|_2^2 < 0,$$

Next we consider the update of inner product as

$$\langle \mathbf{w}_r^{(t+1)}, \boldsymbol{\xi}_i\rangle = \langle \mathbf{w}_r^{(0)} + \gamma_r^{(t+1)}\boldsymbol{\mu}\|\boldsymbol{\mu}\|_2^{-2} + \sum_{i=1}^{n} \rho_{r,i}^{(t+1)}\|\boldsymbol{\xi}_i\|_2^{-2}\boldsymbol{\xi}_i, \boldsymbol{\xi}_i\rangle$$

$$= \langle \mathbf{w}_r^{(0)}, \boldsymbol{\xi}_i\rangle + \gamma_r^{(t+1)}\langle \boldsymbol{\mu}, \boldsymbol{\xi}_i\rangle\|\boldsymbol{\mu}\|_2^{-2} + \rho_{r,i}^{(t+1)} + \sum_{i'\neq i}^{n} \rho_{r,i'}^{(t+1)}\|\boldsymbol{\xi}_{i'}\|_2^{-2}\langle \boldsymbol{\xi}_{i'}, \boldsymbol{\xi}_i\rangle$$

$$\leq \langle \mathbf{w}_r^{(0)}, \boldsymbol{\xi}_i\rangle + \gamma_r^{(t+1)}|\langle \boldsymbol{\mu}, \boldsymbol{\xi}_i\rangle|\|\boldsymbol{\mu}\|_2^{-2} + \sum_{i'\neq i}^{n} |\rho_{r,i'}^{(t+1)}|\|\boldsymbol{\xi}_{i'}\|_2^{-2}|\langle \boldsymbol{\xi}_{i'}, \boldsymbol{\xi}_i\rangle|$$

$$\leq \langle \mathbf{w}_r^{(0)}, \boldsymbol{\xi}_i\rangle + 5\sqrt{\frac{\log(6n^2/\delta)}{d}}n < 0,$$

where the last inequality is by a anti-concentration analysis shown in Lemma C.6. This completes the induction for $i \in \mathcal{I}_{r,-}^{(t)}$. $\qquad\square$

Before formally stating the main lemma on the lower bound of noise memorization, we prepare several lemmas that will be useful.

**Lemma C.8.** *Suppose that $\delta > 0$. Then with probability $1 - \delta$, for all $r \in [m]$, we have:*

$$\left| \sum_{i:y_i=1} \mathbb{1}_{i \in \mathcal{I}_{r,+}^{(0)}} - \frac{n}{4} \right| \leq \sqrt{\frac{n}{2} \log(4/\delta)}.$$

*Proof.* The proof is by Hoeffding's inequality, for arbitrary $t > 0$, we have that

$$\mathbb{P}\left( \left| \sum_{i:y_i=1} \mathbb{1}_{i \in \mathcal{I}_{r,+}} - \mathbb{E}\left[ \sum_{i:y_i=1} \mathbb{1}_{i \in \mathcal{I}_{r,+}} \right] \right| \leq t \right) \leq 2 \exp\left( -\frac{2t^2}{n} \right).$$

By the randomness of initialization and Rademacher distribution, we have:

$$\mathbb{E}\left[ \sum_{i:y_i=1} \mathbb{1}_{i \in \mathcal{I}_{r,+}} \right] = \frac{n}{4}$$

Setting $t = \sqrt{n/2 \log(4/\delta)}$ and taking a union bound over $r \in [m]$, we conclude with high probability at least $1 - \delta$, it holds:

$$\left| \sum_{i:y_i=1} \mathbb{1}_{i \in \mathcal{I}_{r,+}} - \frac{n}{4} \right| \leq \sqrt{\frac{n}{2} \log(4/\delta)}.$$

$\square$

To establish the lower bound for noise memorization we define

$$B_{r,+}^{(t)} \triangleq \sum_{i:y_i=+1} (\rho_{r,i}^{(t)} + \langle \mathbf{w}_r^{(0)}, \boldsymbol{\xi}_i \rangle) \mathbb{1}_{i \in \mathcal{I}_{r,+}^{(t)}}, \quad B_{r,-}^{(t)} \triangleq \sum_{i:y_i=-1} (\rho_{r,i}^{(t)} + \langle \mathbf{w}_r^{(0)}, \boldsymbol{\xi}_i \rangle) \mathbb{1}_{i \in \mathcal{I}_{r,+}^{(t)}}.$$

**Lemma C.9.** *Suppose $\delta > 0$, the with probability at least $1 - \delta$, we have*

$$B_{r,+}^{(0)} \leq \sigma_0 \sigma_\xi \sqrt{dn}(1 + \sqrt{\log(1/\delta)/2}), \quad B_{r,-}^{(0)} \leq \sigma_0 \sigma_\xi \sqrt{dn}(1 + \sqrt{\log(1/\delta)/2}).$$

*Proof.* By Bernstein's inequality, for arbitrary $t > 0$, we have

$$P(|B_{r,+}^{(0)} - \sigma_0 \sigma_\xi \sqrt{nd}| > t) \leq \exp(-\frac{t^2}{2n/4\sigma_0^2 \sigma_\xi^2 d}).$$

Setting $t = \sigma_0 \sigma_\xi \sqrt{nd \log(1/\delta)/2}$, we further have

$$B_{r,+}^{(0)} \leq \sigma_0 \sigma_\xi \sqrt{dn}(1 + \sqrt{\log(1/\delta)/2}).$$

Similarly, the same result holds for $B_{r,-}^{(0)}$. $\square$

**Lemma C.10.** *For arbitrary constant $C > 0$, under the condition $n \geq \frac{8C^2 \log(1/\delta)}{\log(1/(1-(\delta/m)^2))}$, then with probability at least $1 - \delta$, it satisfies*

$$|\langle \mathbf{w}_r^{(0)}, \boldsymbol{\xi}_i \rangle| > \frac{CB_{r,+}^{(0)}}{n}, \quad |\langle \mathbf{w}_r^{(0)}, \boldsymbol{\xi}_i \rangle| > \frac{CB_{r,-}^{(0)}}{n}.$$

*Proof of Lemma C.10.* Here we want to show $|\langle \mathbf{w}_r^{(0)}, \boldsymbol{\xi}_i \rangle| > \frac{B_{r,+}^{(0)}}{n}$ with high probability. To see this, because $\langle \mathbf{w}_r^{(0)}, \boldsymbol{\xi}_i \rangle$ is a random variable with positive mean and variance $\sigma_0^2 \sigma_\xi^2 d$. Besides, by Lemma C.9, we have

$$\frac{B_{r,+}^{(0)}}{n} \leq \frac{\sigma_0 \sigma_\xi \sqrt{dn}(1 + \sqrt{\log(1/\delta)/2})}{n}.$$

By Lemma B.1, we recall

$$\mathbb{P}\big(|\langle \mathbf{w}_r^{(0)}, \boldsymbol{\xi}_i\rangle| \le t\big) \le \sqrt{1 - \exp\left(-\frac{8t^2}{\pi\sigma_0^2\sigma_\xi^2 d}\right)},$$

Thus when $n \ge \frac{8C^2\log(1/\delta)}{\log(1/(1-(\delta/m)^2))}$, we have

$$\mathbb{P}\big(|\langle \mathbf{w}_r^{(0)}, \boldsymbol{\xi}_i\rangle| \le \frac{C\sigma_0\sigma_\xi\sqrt{dn}(1+\sqrt{\log(1/\delta)/2})}{n}\big) \le \delta/m$$

and by union bound, we have with probability at least $1 - \delta$, it holds $|\langle \mathbf{w}_r^{(0)}, \boldsymbol{\xi}_i\rangle| > \frac{B_{r,+}^{(0)}}{n}$ and $|\langle \mathbf{w}_r^{(0)}, \boldsymbol{\xi}_i\rangle| > \frac{B_{r,-}^{(0)}}{n}$. $\qquad\square$

Besides, we define

$$\Psi_{r,i}^{(t)} \triangleq \rho_{r,i}^{(t)} + \langle \mathbf{w}_r^{(0)}, \boldsymbol{\xi}_i\rangle, \text{ with } y_i = 1, i \in \mathcal{I}_{r,+}^{(t)}$$
$$\Phi_{r,i}^{(t)} \triangleq \rho_{r,i}^{(t)} + \langle \mathbf{w}_r^{(0)}, \boldsymbol{\xi}_i\rangle, \text{ with } y_i = -1, i \in \mathcal{I}_{r,+}^{(t)}$$

With all the results (lemmas) and definitions outlined above at hand, we are ready to state the lemmas that provide the lower bound for noise memorization as follows.

**Lemma C.11.** *Under the same condition as Theorem 4.2, then with probability at least $1 - \delta$,*

$$\Psi_{r,i}^{(t)} \ge (1 + \frac{0.96\eta\sigma_\xi^2 d}{nm\tau})^t \Psi_{r,i}^{(0)}, \tag{22}$$

$$\Psi_{r,i}^{(t)} \ge \frac{101C_\ell}{M+1} B_{r,-}^{(t)}, \tag{23}$$

$$\rho_{r,i:y_i=1}^{(t)} \mathbb{1}_{i\in\mathcal{I}_{r,+}^{(t)}} \ge 0. \tag{24}$$

*Proof.* The proof is by induction. We can check

$$\rho_{r,i:y_i=1}^{(0)} \mathbb{1}_{i\in\mathcal{I}_{r,+}} = 0, \quad \Psi_{r,i}^{(0)} \ge (1 + \frac{0.96\eta\sigma_\xi^2 d}{nm\tau})^0 \Psi_{r,i}^{(0)}.$$

Besides, by Lemma C.10 and $M \in \frac{n}{2}(1 \pm o(n^{-1/2}))$, it holds that

$$\Psi_{r,i}^{(0)} = \langle \mathbf{w}_r^{(0)}, \boldsymbol{\xi}_i\rangle \ge 101\frac{C_\ell}{M+1} B_{r,-}^{(0)}.$$

Assuming that the results hold at $t$, we continue the proof by calculating the propagation of $B_{r,+}^{(t)}$ and $B_{r,-}^{(t)}$ respectively. First, we can see by Lemma C.1 and by induction, for $y_i = 1$ and $i \in \mathcal{I}_{r,+}$,

$$\langle \mathbf{w}_r^{(t)}, \boldsymbol{\xi}_i + \boldsymbol{\epsilon}_i\rangle \ge \langle \mathbf{w}_r^{(0)}, \boldsymbol{\xi}_i\rangle + \rho_{r,i}^{(t)} - |\langle \mathbf{w}_r^{(t)}, \boldsymbol{\epsilon}_i\rangle| - 6\sqrt{\frac{\log(6n^2/\delta)}{d}}n$$
$$\ge (1 - o(1))\langle \mathbf{w}_r^{(0)}, \boldsymbol{\xi}_i\rangle + \rho_{r,i}^{(t)} \ge 0.$$

We can show that

$$B_{r,-}^{(t+1)} = B_{r,-}^{(t)} + \frac{\eta}{nm\tau}\sum_{i:y_i=-1}\left[(1-\ell_i'^{(t)})\langle \mathbf{w}_r^{(t)}, \boldsymbol{\xi}_i + \boldsymbol{\epsilon}_i\rangle - \sum_{j\ne i}\ell_{i,j}'^{(t)}\langle \mathbf{w}_r^{(t)}, \boldsymbol{\xi}_j\rangle\mathbb{1}_{i\in\mathcal{I}_{r,+}}\right]\mathbb{1}_{i\in\mathcal{I}_{r,+}}\|\boldsymbol{\xi}_i\|_2^2$$

$$\le B_{r,-}^{(t)} + \frac{\eta}{nm\tau}\sum_{i:y_i=-1}\left[\frac{C_\ell(M+1)-1}{C_\ell(M+1)}(\langle \mathbf{w}_r^{(0)}, \boldsymbol{\xi}_i\rangle + \rho_{r,i}^{(t)} + |\langle \mathbf{w}_r^{(t)}, \boldsymbol{\epsilon}_i\rangle| + 6\sqrt{\frac{\log(6n^2/\delta)}{d}}n)\right.$$

$$\left. - \sum_{j\ne i}\mathbb{1}_{j\in\mathcal{I}_{r,+}}\frac{1}{C_\ell(M+1)}(\langle \mathbf{w}_r^{(0)}, \boldsymbol{\xi}_j\rangle + \rho_{r,j}^{(t)} - 6\sqrt{\frac{\log(6n^2/\delta)}{d}}n)\right](\sigma_\xi^2 d + \sigma_\xi^2\sqrt{d\log(6n^2/\delta)})\mathbb{1}_{i\in\mathcal{I}_{r,+}}$$

$$
\leq B_{r,-}^{(t)} + \frac{\eta}{n\tau m}\left[\frac{C_\ell(M+1)-1}{C_\ell(M+1)}1.01B_{r,-}^{(t)} - \sum_{i:y_i=-1}\mathbb{1}_{i\in\mathcal{I}_{r,+}}\frac{1}{C_\ell(M+1)}0.99B_{r,-}^{(t)}\right](\sigma_\xi^2 d + \sigma_\xi^2\sqrt{d\log(6n^2/\delta)})
$$

$$
- \sum_{i:y_i=-1}\mathbb{1}_{i\in\mathcal{I}_{r,+}}\frac{1}{C_\ell(M+1)}(B_{r,-}^{(t)} - \sum_{j\neq i}6\sqrt{\frac{\log(6n^2/\delta)}{d}}n)\Bigg](\sigma_\xi^2 d + \sigma_\xi^2\sqrt{d\log(6n^2/\delta)})
$$

$$
\leq B_{r,-}^{(t)} + \frac{\eta}{nm\tau}\left[1.01B_{r,-}^{(t)} - \frac{0.99^2}{2C_l}B_{r,+}^{(t)}\right]1.035\sigma_\xi^2 d
$$

$$
\leq B_{r,-}^{(t)} + \frac{\eta}{nm\tau}\left[1.05B_{r,-}^{(t)} - 0.5B_{r,+}^{(t)}\right]\sigma_\xi^2 d
$$

$$
\leq B_{r,-}^{(t)} + \frac{1.05\eta\sigma_\xi^2 d}{nm\tau}B_{r,-}^{(t)},
$$

where the first inequality is by Lemma C.2, Lemma C.1 and Lemma B.5. Furthermore, the second inequality is by Lemma C.5, i.e., $|\langle \mathbf{w}_r^{(t)}, \boldsymbol{\epsilon}_i\rangle| \leq 1/C_\xi|\langle \mathbf{w}_r^{(0)}, \boldsymbol{\xi}_i\rangle|$ (for $C_\xi > 200$), $|0.01\langle \mathbf{w}_r^{(0)}, \boldsymbol{\xi}_i\rangle| \geq 6\sqrt{\log(6n^2/\delta)d^{-1}}n$, the definition of $B_{r,+}^{(t)}$ and $B_{r,-}^{(t)}$ and the induction $B_{r,+}^{(t)} \geq B_{r,+}^{(0)}$, $B_{r,-}^{(t)} \geq B_{r,-}^{(0)}$. The third inequality is by $M \geq 100C_\ell - 1$, $M \in \frac{n}{2}(1 \pm o(n^{-1/2}))$, $d > 10000\log(6n^2/\delta)$ and Lemma C.8. The last inequality is by $B_{r,+}^{(t)} \geq 0$.

As a result, we conclude that

$$
B_{r,-}^{(t)} \leq (1 + \frac{1.05\eta\sigma_\xi^2 d}{nm\tau})^t B_{r,-}^{(0)}.
$$

Finally, the induction step for $\Psi_{r,i}^{(t)}$ can be calculated as follows:

$$
\Psi_{r,i}^{(t+1)} = \Psi_{r,i}^{(t)} + \frac{\eta}{nm\tau}\left[(1 - \ell_i'^{(t)})\langle \mathbf{w}_r^{(t)}, \boldsymbol{\xi}_i + \boldsymbol{\epsilon}_i\rangle - \sum_{j\neq i}\mathbb{1}_{j\in\mathcal{I}_{r,+}}\ell_{i,j}'^{(t)}\langle \mathbf{w}_r^{(t)}, \boldsymbol{\xi}_j\rangle\right]\mathbb{1}_{i\in\mathcal{I}_{r,+}}\|\boldsymbol{\xi}_i\|_2^2
$$

$$
\geq \Psi_{r,i}^{(t)} + \frac{\eta}{nm\tau}\left[\frac{(M+1)-C_\ell}{(M+1)}(\langle \mathbf{w}_r^{(0)}, \boldsymbol{\xi}_i\rangle + \rho_{r,i}^{(t)} - |\langle \mathbf{w}_r^{(t)}, \boldsymbol{\epsilon}_i\rangle| - 6\sqrt{\frac{\log(6n^2/\delta)}{d}}n)\right.
$$

$$
\left. - \sum_{j\neq i}\mathbb{1}_{j\in\mathcal{I}_{r,+}}\frac{C_l}{M+1}(\langle \mathbf{w}_r^{(0)}, \boldsymbol{\xi}_j\rangle + \rho_{r,j}^{(t)} + 6\sqrt{\frac{\log(6n^2/\delta)}{d}}n)\right](\sigma_\xi^2 d - \sigma_\xi^2\sqrt{d\log(6n^2/\delta)})\mathbb{1}_{i\in\mathcal{I}_{r,+}}
$$

$$
\geq \Psi_{r,i}^{(t)} + \frac{\eta}{nm\tau}\left[\frac{(M+1)-C_\ell}{M+1}0.99\Psi_{r,i}^{(t)} - \frac{C_\ell}{M+1}1.01B_{r,-}^{(t)}\right](\sigma_\xi^2 d - \sigma_\xi^2\sqrt{d\log(6n^2/\delta)})
$$

$$
\geq \Psi_{r,i}^{(t)} + \frac{\eta}{nm\tau}\left[0.99^2\Psi_{r,i}^{(t)} - 1.01\frac{C_l}{M+1}B_{r,-}^{(t)}\right]0.99\sigma_\xi^2 d
$$

$$
\geq \Psi_{r,i}^{(t)} + \frac{\eta\sigma_\xi^2 d}{nm\tau}\left[0.96\Psi_{r,i}^{(t)}\right],
$$

where the first inequality is by Lemma C.2, Lemma C.1 and Lemma B.5. Furthermore, the second inequality is by Lemma C.5, i.e., $|\langle \mathbf{w}_r^{(t)}, \boldsymbol{\epsilon}_i\rangle| \leq 1/C_\xi|\langle \mathbf{w}_r^{(0)}, \boldsymbol{\xi}_i\rangle|$ (for $C_\xi > 200$), $|0.01\langle \mathbf{w}_r^{(0)}, \boldsymbol{\xi}_i\rangle| \geq 6\sqrt{\log(6n^2/\delta)d^{-1}}n$, the definition of $B_{r,+}^{(t)}$ and $B_{r,-}^{(t)}$. The third inequality is by $M \geq 100C_\ell - 1$. The last inequality is by induction (23).

Then we check induction induction (23) through following inequalities:

$$
B_{r,-}^{(t)} \leq (1 + \frac{1.05\eta\sigma_\xi^2 d}{nm\tau})^t B_{r,-}^{(0)},
$$

$$
\Psi_{r,i}^{(t)} \geq (1 + \frac{0.96\eta\sigma_\xi^2 d}{nm\tau})^t \Psi_{r,i}^{(0)}.
$$

Together it confirms that $\Psi_{r,i}^{(t)} \geq \frac{101C_\ell}{M+1}B_{r,-}^{(t)}$.

Finally, we check the sign of $\rho_{r,i}^{(t)}$ for $i : y_i = 1$ and $i \in \mathcal{I}_{r,+}^{(t)}$:

$$\rho_{r,i}^{(t)} = \Psi_{r,i}^{(t)} - \langle \mathbf{w}_r^{(0)}, \boldsymbol{\xi} \rangle \geq (1 + \frac{\eta \sigma_\xi^2 d}{2nm\tau})^t \Psi_{r,i}^{(t)} - \langle \mathbf{w}_r^{(0)}, \boldsymbol{\xi} \rangle > \frac{\eta \sigma_\xi^2 d}{2nm\tau} \Psi_{r,i}^{(t)} > 0.$$

This completes the induction proof. $\qquad\square$

### C.1.3 Noise Memorization: Proof of Lemma 5.2

Before proving Lemma 5.2, we require a lower bound for the initialization. Define that $\mathcal{U}_+^{(0)} = \{r : \langle \mathbf{w}_r^{(0)}, \boldsymbol{\xi}_i \rangle > 0\}$ for $y_i = 1$, and $\mathcal{U}_-^{(0)} = \{r : \langle \mathbf{w}_r^{(0)}, \boldsymbol{\xi}_i \rangle < 0\}$ for $y_i = -1$.

**Lemma C.12.** *Suppose that $\delta > 0$ and $m \geq \widetilde{\Omega}(1)$. Then with probability at least $1 - \delta$, we have*

$$\frac{1}{m} \sum_{r \in \mathcal{U}_+^{(0)}} \Psi_{r,i}^{(0)} \geq 0.2\sigma_0\sigma_\xi\sqrt{d}, \quad \frac{1}{m} \sum_{r \in \mathcal{U}_-^{(0)}} \Phi_{r,i}^{(0)} \geq 0.2\sigma_0\sigma_\xi\sqrt{d}.$$

*Proof of Lemma C.12.* Consider $y_i = 1$. Note that $\langle \mathbf{w}_r^{(0)}, \boldsymbol{\xi}_i \rangle \sim \mathcal{N}(0, \sigma_0^2\|\boldsymbol{\xi}_i\|_2^2)$. We define that the event $\mathcal{A} = \{r \in [m], \langle \mathbf{w}_r^{(0)}, \boldsymbol{\xi}_i \rangle > 0\}$. Then we can compute that $\langle \mathbf{w}_r^{(0)}, \boldsymbol{\xi}_i \rangle \mathbb{1}(\mathcal{A})$ becomes a half-normal distribution with the expectation

$$\mathbb{E}[\langle \mathbf{w}_r^{(0)}, \boldsymbol{\xi}_i \rangle \mathbb{1}(\mathcal{A})] = \frac{\sqrt{2}\sigma_0\|\boldsymbol{\xi}_i\|_2}{\sqrt{\pi}}.$$

We then apply the sub-Gaussian concentration inequality that with probability at least $1 - \delta$

$$\left| \sum_{r \in \mathcal{U}_+^{(0)}} \Psi_{r,i}^{(0)} - \frac{m\sigma_0\|\boldsymbol{\xi}_i\|_2}{\sqrt{2\pi}} \right| \leq \widetilde{O}(m^{-1/2}).$$

Then we have

$$\frac{1}{m} \sum_{r \in \mathcal{U}_+^{(0)}} \Psi_{r,i}^{(0)} \geq 0.4\sigma_0\|\boldsymbol{\xi}_i\|_2 \geq \sigma_0\sigma_\xi\sqrt{d},$$

where we have used Lemma B.5. Similarly, we can show that for $y_i = -1$, the same result holds.

$\qquad\square$

*Proof of Lemma 5.2.* From the upper bound on (21), we take the maximum over $r \in \mathcal{U}_+^{(0)}$, which gives

$$\max_r A_r^{(t)} \leq \left(1 + 0.52\frac{\eta\|\boldsymbol{\mu}\|_2^2}{m\tau}\right)^t \max_r A_r^{(0)}$$

$$\leq \left(1 + 0.52\frac{\eta\|\boldsymbol{\mu}\|_2^2}{m\tau}\right)^t \sigma_0\|\boldsymbol{\mu}\|_2\sqrt{2\log(8m/\delta)}.$$

Under the SNR condition $n \cdot \text{SNR}^2 \leq 1.8$, we can see there exists a scale difference between $\max_{r,i} \Psi_{r,i}^{(t)}$ and $\max_{r,i} A_r^{(t)}$ at the end of first stage.

At the same time, for noise memorization, from the lower bound established in Lemma C.11, we have that

$$\frac{1}{m} \sum_{r \in \mathcal{U}_+^{(0)}} \Psi_{r,i}^{(t)} \geq (1 + \frac{0.96\eta\sigma_\xi^2 d}{nm\tau})^t \frac{1}{m} \sum_{r \in \mathcal{U}_+^{(0)}} \Psi_{r,i}^{(0)}$$

$$\geq (1 + \frac{0.96\eta\sigma_\xi^2 d}{nm\tau})^t 0.2\sigma_0\sigma_\xi\sqrt{d},$$

where the second inequality is due to Lemma C.12.

Let
$$T_1 = \log\left(20/(\sigma_0\sigma_\xi\sqrt{d})\right)/\log\left(1 + 0.96\frac{\eta\sigma_\xi^2 d}{nm\tau}\right).$$

Then we have $\frac{1}{m}\sum_{r\in\mathcal{U}_+^{(0)}}\Psi_{r,i}^{(t)}$ reach 3 within $T_1$ iterations by (22). Similarly, we can also show that $\frac{1}{m}\sum_{r\in\mathcal{U}_-^{(0)}}\Phi_{r,i}^{(t)}$ reach 3 within $T_1$ iterations.

On the other hand, we compute the scale of $\max_r A_r^{(T_1)}$ as

$$
\begin{aligned}
\max_r A_r^{(T_1)} &\le \left(1 + 0.52\frac{\eta\|\boldsymbol{\mu}\|_2^2}{m\tau}\right)^{T_1}\sigma_0\|\boldsymbol{\mu}\|_2\sqrt{2\log(8m/\delta)}\\
&= \exp\left(\frac{\log(1 + 0.52\frac{\eta\|\boldsymbol{\mu}\|_2^2}{m\tau})}{\log(1 + 0.96\frac{\eta\sigma_\xi^2 d}{nm\tau})}\log(12/(\sigma_0\sigma_\xi\sqrt{d}))\right)\sigma_0\|\boldsymbol{\mu}\|_2\sqrt{2\log(8m/\delta)}\\
&\le \exp\left((0.55\cdot n\cdot\mathrm{SNR}^2 + O((\frac{\eta\|\boldsymbol{\mu}\|_2^2}{m\tau})^2)\log(20/(\sigma_0\sigma_\xi\sqrt{d}))\right)\sigma_0\|\boldsymbol{\mu}\|_2\sqrt{2\log(8m/\delta)}\\
&\le \exp\left((0.55\cdot n\cdot\mathrm{SNR}^2 + 0.01)\log(30/(\sigma_0\sigma_\xi\sqrt{d}))\right)\sigma_0\|\boldsymbol{\mu}\|_2\sqrt{2\log(8m/\delta)}\\
&\le \exp\left(\log(30/(\sigma_0\sigma_\xi\sqrt{d}))\right)\sigma_0\|\boldsymbol{\mu}\|_2\sqrt{2\log(8m/\delta)}\\
&= 20\sqrt{2\log(8m/\delta)}\mathrm{SNR}\\
&= O(\sqrt{\log(m/\delta)/n})\\
&= \widetilde{O}(n^{-1/2})
\end{aligned}
$$

where we choose $\eta$ sufficiently small for the third inequality. The last inequality is by the SNR condition. Because we can choose $n \ge C\log(m/\delta)$ for sufficiently large constant $C$, $\max_r A_r^{(T_1)} = o(1)$.

$\square$

## C.2  Second Stage

**Proposition C.13.** *Let $T^*$ be the maximum admissible iteration and let $\alpha = \log(3MT^*)$. Then we can show*

$$|\gamma_r^{(t)}| \le \alpha, \quad |\rho_{r,i}^{(t)}| \le \alpha.$$

*Proof of Proposition C.13.* We need to show $\rho_{r,i}^{(t)} \le \alpha$. We prove the claim by induction. It is clear when $t = 0$, $\rho_{r,i}^{(t)} = 0 \le \alpha$. Suppose for all $0 \le t \le \widetilde{T} - 1$, we have $\rho_{r,i}^{(t)} \le \alpha$. We aim to show the claim holds for $\widetilde{T}$.

By the update of $\rho_{r,i}^{(t)}$,

$$\rho_{r,i}^{(t+1)} \le \rho_{r,i}^{(t)} + \frac{\eta}{nm\tau}(1 - \ell_i'^{(t)})\sigma(\langle\mathbf{w}_r^{(t)}, \boldsymbol{\xi}_i + \boldsymbol{\epsilon}_i\rangle)\|\boldsymbol{\xi}_i\|_2^2.$$

Here without loss of generality, we consider the $r, i$ pairs such that $i \in \mathcal{I}_{r,+}^{(0)}$, i.e., $\langle\mathbf{w}_r^{(0)}, \boldsymbol{\xi}_i\rangle > 0$ with $y_i = 1$. Let $t_{r,i}$ be the last time $t$ such that $\rho_{r,i}^{(t)} \le 0.5\alpha$. Then we have

$$
\begin{aligned}
\rho_{r,i}^{(\widetilde{T})} &\le \rho_{r,i}^{(t_r)} + \frac{\eta}{nm\tau}(1 - \ell_i'^{(t_{r,i})})\sigma(\langle\mathbf{w}_r^{(t_{r,i})}, \boldsymbol{\xi}_i + \boldsymbol{\epsilon}_i\rangle)\|\boldsymbol{\xi}_i\|_2^2\\
&\quad + \sum_{t_{r,i}\le t\le\widetilde{T}}\frac{\eta}{nm\tau}(1 - \ell_i'^{(t)})\sigma(\langle\mathbf{w}_r^{(t)}, \boldsymbol{\xi}_i + \boldsymbol{\epsilon}_i\rangle)\|\boldsymbol{\xi}_i\|_2^2.
\end{aligned}
\tag{25}
$$

The second term can be bounded as

$$\frac{\eta}{nm\tau}(1 - \ell_i'^{(t_{r,i})})\sigma(\langle\mathbf{w}_r^{(t_{r,i})}, \boldsymbol{\xi}_i + \boldsymbol{\epsilon}_i\rangle)\|\boldsymbol{\xi}_i\|_2^2 \le \frac{\eta}{nm\tau}\left(\frac{3}{2}\langle\mathbf{w}_r^{(0)}, \boldsymbol{\xi}_i\rangle + \rho_{r,i}^{(t_{r,i})}\right)\|\boldsymbol{\xi}_i\|_2^2$$

$$\leq \frac{3\eta\sigma_\xi^2 d}{2nm\tau}(0.3\alpha + 0.5\alpha)$$
$$\leq 0.25\alpha, \tag{26}$$

where the first inequality is by $1 - \ell_i'^{(t_r)} \leq 1$ and modified Lemma C.1 with $\rho_{r,i}^{(t)} = O(\alpha)$. The second inequality is by $\langle \mathbf{w}_r^{(0)}, \boldsymbol{\xi}_i \rangle \leq 0.2\alpha$ and $\eta \leq \frac{5}{24}\frac{nm\tau}{\sigma_\xi^2 d}$.

For notation convenience, we let

$$F_0(\mathbf{W}, \mathbf{x}_i) = \mathrm{Sim}_\mathbf{h}(\mathbf{x}_i, \widehat{\mathbf{x}}_i)/\tau$$
$$= \frac{1}{m\tau}\sum_{r=1}^m \sigma(\langle \mathbf{w}_r, y_i\boldsymbol{\mu}\rangle)\mathrm{sg}(\sigma(\langle \mathbf{w}_r, y_i\boldsymbol{\mu}\rangle)) + \frac{1}{m\tau}\sum_{r=1}^m \sigma(\langle \mathbf{w}_r, \boldsymbol{\xi}_i\rangle)\mathrm{sg}(\sigma(\langle \mathbf{w}_r, \boldsymbol{\xi}_i + \boldsymbol{\epsilon}_i\rangle))$$
$$F_j(\mathbf{W}, \mathbf{x}_i) = \mathrm{Sim}_\mathbf{h}(\mathbf{x}_i, \widehat{\mathbf{x}}_j)/\tau$$
$$= \frac{1}{m\tau}\sum_{r=1}^m \sigma(\langle \mathbf{w}_r, y_i\boldsymbol{\mu}\rangle)\mathrm{sg}(\sigma(\langle \mathbf{w}_r, y_j\boldsymbol{\mu}\rangle)) + \frac{1}{m\tau}\sum_{r=1}^m \sigma(\langle \mathbf{w}_r, \boldsymbol{\xi}_i\rangle)\mathrm{sg}(\sigma(\langle \mathbf{w}_r, \boldsymbol{\xi}_j\rangle)), \text{ for } j = 1, ..., M$$

Next, we show the bound for $F_0(\mathbf{W}^{(t)}, \mathbf{x}_i)$ and $F_j(\mathbf{W}^{(t)}, \mathbf{x}_i)$ for $j = 1, ..., M$ as

$$F_0(\mathbf{W}^{(t)}, \mathbf{x}_i) \geq \frac{1}{m\tau}\sum_{r=1}^m \sigma(\langle \mathbf{w}_r^{(t)}, \boldsymbol{\xi}_i\rangle)\sigma(\langle \mathbf{w}_r^{(t)}, \boldsymbol{\xi}_i + \boldsymbol{\epsilon}_i\rangle).$$

Further, we have for $j = 1, ..., M$

$$F_j(\mathbf{W}^{(t)}, \mathbf{x}_i) = \frac{1}{m\tau}\sum_{r=1}^m \sigma(\langle \mathbf{w}_r^{(t)}, \boldsymbol{\xi}_i\rangle)\sigma(\langle \mathbf{w}_r^{(t)}, \boldsymbol{\xi}_j\rangle).$$

Then we can bound the difference between $F_0(\mathbf{W}^{(t)}, \mathbf{x}_i)$ and $F_j(\mathbf{W}^{(t)}, \mathbf{x}_i)$ as

$$F_0(\mathbf{W}^{(t)}, \mathbf{x}_i) - F_j(\mathbf{W}^{(t)}, \mathbf{x}_i) \geq \frac{1}{m\tau}\sum_{r=1}^m \sigma(\langle \mathbf{w}_r^{(t)}, \boldsymbol{\xi}_i\rangle)\sigma(\langle \mathbf{w}_r^{(t)}, \boldsymbol{\xi}_i + \boldsymbol{\epsilon}_i\rangle) - \frac{1}{m\tau}\sum_{r=1}^m \sigma(\langle \mathbf{w}_r^{(t)}, \boldsymbol{\xi}_i\rangle)\sigma(\langle \mathbf{w}_r^{(t)}, \boldsymbol{\xi}_j\rangle)$$
$$= \frac{1}{m\tau}\sum_{r=1}^m \sigma(\langle \mathbf{w}_r^{(t)}, \boldsymbol{\xi}_i\rangle)\left(\sigma(\langle \mathbf{w}_r^{(t)}, \boldsymbol{\xi}_i + \boldsymbol{\epsilon}_i\rangle) - \sigma(\langle \mathbf{w}_r^{(t)}, \boldsymbol{\xi}_j\rangle)\right)$$
$$\geq 0.5\alpha(\frac{1}{2}\langle \mathbf{w}_r^{(0)}, \boldsymbol{\xi}_i\rangle + \rho_{r,i}^{(t)} - \frac{3}{2}\langle \mathbf{w}_r^{(0)}, \boldsymbol{\xi}_j\rangle - \rho_{r,j}^{(t)})/\tau$$
$$\geq 0.5\alpha(\frac{1}{2}\langle \mathbf{w}_r^{(0)}, \boldsymbol{\xi}_i\rangle + \frac{1}{2}\rho_{r,i}^{(t)} - \frac{3}{2}\langle \mathbf{w}_r^{(0)}, \boldsymbol{\xi}_j\rangle)/\tau$$
$$\geq 0.5\alpha \cdot 0.1\alpha/\tau$$
$$\geq \alpha, \tag{27}$$

where the second inequality is by $\sigma(\langle \mathbf{w}_r^{(t)}, \boldsymbol{\xi}_i\rangle) \geq \frac{3}{4}\langle \mathbf{w}_r^{(0)}, \boldsymbol{\xi}_i\rangle + \rho_{r,i}^{(t)} \geq \rho_{r,i}^{(t)} \geq 0.5\alpha$ for $i \in \mathcal{I}_{r,+}^{(0)}$. Further $\sigma(\langle \mathbf{w}_r^{(t)}, \boldsymbol{\xi}_i + \boldsymbol{\epsilon}_i\rangle) \geq \frac{1}{2}\langle \mathbf{w}_r^{(0)}, \boldsymbol{\xi}_i\rangle + \rho_{r,i}^{(t)}$, $\sigma(\langle \mathbf{w}_r^{(t)}, \boldsymbol{\xi}_j\rangle) \leq \frac{3}{2}\langle \mathbf{w}_r^{(0)}, \boldsymbol{\xi}_j\rangle + \rho_{r,j}^{(t)}$. The fourth inequality is by induction and $\frac{1}{2}\langle \mathbf{w}_r^{(0)}, \boldsymbol{\xi}_i\rangle - \frac{3}{2}\langle \mathbf{w}_r^{(0)}, \boldsymbol{\xi}_j\rangle \geq -0.15\alpha$. The last inequality is by the condition that $\alpha \geq 20\tau$.

Further, we bound the loss derivative as

$$1 - \ell_i'^{(t)} = 1 - \frac{1}{1 + \sum_{j=1}^M \exp(F_j(\mathbf{W}^{(t)}, \mathbf{x}_i) - F_0(\mathbf{W}^{(t)}, \mathbf{x}_i))}$$
$$\leq 1 - \frac{1}{1 + M\exp(-\alpha)}$$
$$\leq 1 - \frac{T^*}{1 + T^*}$$

$$= \frac{1}{T^* + 1}, \tag{28}$$

where the second inequality is by (27).

Finally, we bound

$$\sum_{t_{r,i} \leq t \leq \widetilde{T}} \frac{\eta}{nm\tau} (1 - \ell_i'^{(t)}) \sigma(\langle \mathbf{w}_r^{(t)}, \boldsymbol{\xi}_i + \boldsymbol{\epsilon}_i \rangle) \|\boldsymbol{\xi}_i\|_2^2 \leq \frac{3\eta\sigma_\xi^2 d}{2nm\tau} \cdot 2\alpha \cdot \sum_{t_{r,i} \leq t \leq \widetilde{T}} (1 - \ell_i'^{(t)})$$

$$\leq \frac{3\eta\sigma_\xi^2 d}{nm\tau} \cdot \alpha \cdot \frac{T^*}{T^* + 1}$$

$$\leq 0.25\alpha \tag{29}$$

where the first inequality is by Lemma B.5 and $\sigma(\langle \mathbf{w}_r^{(t)}, \boldsymbol{\xi}_i + \boldsymbol{\epsilon}_i \rangle) \leq \frac{3}{2} \langle \mathbf{w}_r^{(0)}, \boldsymbol{\xi}_i \rangle + \rho_{r,i}^{(t)} \leq 2\alpha$ by induction. The last inequality is by the condition $\eta \leq \frac{1}{12} \frac{nm\tau}{\sigma_\xi^2 d}$.

Combining (26) and (29) with (25) gives

$$\rho_{r,i}^{(\widetilde{T})} \leq 0.5\alpha + 0.25\alpha + 0.25\alpha = \alpha,$$

which completes the induction. $\qquad \square$

**Lemma C.14.** *Under Assumption 4.1, for $0 \leq t \leq T^*$, we have*

$$\|\nabla L_S(\mathbf{W}^{(t)})\|_F^2 \leq O(\max\{\|\boldsymbol{\mu}\|_2^2, \sigma_\xi^2 d\}) L_S(\mathbf{W}^{(t)}).$$

*Proof of Lemma C.14.* First, we can write the gradient of $\nabla_{\mathbf{w}_r} L_i(\mathbf{W})$ as

$$\nabla_{\mathbf{w}_r} L_i(\mathbf{W}) = \sum_{j=0}^{M} \frac{\partial L_i(\mathbf{W})}{\partial F_j(\mathbf{W}, \mathbf{x}_i)} \nabla_{\mathbf{w}_r} F_j(\mathbf{W}, \mathbf{x}_i),$$

where

$$\frac{\partial L_i(\mathbf{W})}{\partial F_0} = -1 + \frac{e^{F_0(\mathbf{W}, \mathbf{x}_i)}}{e^{F_0(\mathbf{W}, \mathbf{x}_i)} + \sum_{j=1}^{M} e^{F_j(\mathbf{W}, \mathbf{x}_i)}}$$

$$\frac{\partial L_i(\mathbf{W})}{\partial F_j} = \frac{e^{F_j(\mathbf{W}, \mathbf{x}_i)}}{e^{F_0(\mathbf{W}, \mathbf{x}_i)} + \sum_{j=1}^{M} e^{F_j(\mathbf{W}, \mathbf{x}_i)}}, \text{ for } j = 1, ..., M$$

By the derivation of the gradient,

$$\|\nabla L_S(\mathbf{W}^{(t)})\|_F^2 \leq \left( \frac{1}{n} \sum_{i=1}^{n} \|\nabla L_i(\mathbf{W})\|_F \right)^2$$

$$= \left( \frac{1}{n} \sum_{i=1}^{n} \sqrt{\sum_{r=1}^{m} \|\nabla_{\mathbf{w}_r} L_i(\mathbf{W}^{(t)})\|_2^2} \right)^2$$

$$\leq \left( \frac{1}{n} \sum_{i=1}^{n} \sum_{r=1}^{m} \|\nabla_{\mathbf{w}_r} L_i(\mathbf{W}^{(t)})\|_2 \right)^2$$

$$= \left( \frac{1}{n} \sum_{i=1}^{n} \sum_{r=1}^{m} \left\| \sum_{j=0}^{M} \frac{\partial L_i(\mathbf{W})}{\partial F_j(\mathbf{W}, \mathbf{x}_i)} \nabla_{\mathbf{w}_r} F_j(\mathbf{W}, \mathbf{x}_i) \right\|_2 \right)^2$$

$$\leq \left( \frac{1}{n} \sum_{i=1}^{n} \sum_{r=1}^{m} \sum_{j=0}^{M} \left| \frac{\partial L_i(\mathbf{W})}{\partial F_j(\mathbf{W}, \mathbf{x}_i)} \right| \|\nabla_{\mathbf{w}_r} F_j(\mathbf{W}, \mathbf{x}_i)\|_2 \right)^2, \tag{30}$$

where the first inequality is by triangle inequality and second inequality uses $\sum_i a_i^2 \leq (\sum_i a_i)^2$ for $a_i \geq 0$ and the last inequality is by triangle inequality.

Now we upper bound $\|\nabla_{\mathbf{w}_r} F_j(\mathbf{W}, \mathbf{x}_i)\|_2$ as

$$\|\nabla_{\mathbf{w}_r} F_0(\mathbf{W}, \mathbf{x}_i)\|_2 = \frac{1}{m\tau} \|\sigma'(\langle \mathbf{w}_r, y_i\boldsymbol{\mu}\rangle)\sigma(\langle \mathbf{w}_r, y_i\boldsymbol{\mu}\rangle)y_i\boldsymbol{\mu} + \sigma'(\langle \mathbf{w}_r, \boldsymbol{\xi}_i\rangle)\sigma(\langle \mathbf{w}_r, \boldsymbol{\xi}_i + \boldsymbol{\epsilon}_i\rangle)\boldsymbol{\xi}_i\|_2$$

$$\leq \frac{1}{m\tau}\big(\sigma(\langle \mathbf{w}_r, y_i\boldsymbol{\mu}\rangle)\|\boldsymbol{\mu}\|_2 + \sigma(\langle \mathbf{w}_r, \boldsymbol{\xi}_i + \boldsymbol{\epsilon}_i\rangle)\|\boldsymbol{\xi}_i\|_2\big)$$

$$\leq \frac{1}{m\tau}\big(\sigma(\langle \mathbf{w}_r, y_i\boldsymbol{\mu}\rangle) + \sigma(\langle \mathbf{w}_r, \boldsymbol{\xi}_i + \boldsymbol{\epsilon}_i\rangle)\big)O\big(\max\{\|\boldsymbol{\mu}\|_2, \sigma_\xi\sqrt{d}\}\big),$$

where the first inequality is by triangle inequality and the second inequality is by Jensen's inequality. Similarly, we can obtain for $j = 1, ..., M$

$$\|\nabla_{\mathbf{w}_r} F_j \mathbf{W}, \mathbf{x}_i)\|_2 \leq \frac{1}{m\tau}\big(\sigma(\langle \mathbf{w}_r, y_j\boldsymbol{\mu}\rangle) + \sigma(\langle \mathbf{w}_r, \boldsymbol{\xi}_j\rangle)\big)O\big(\max\{\|\boldsymbol{\mu}\|_2, \sigma_\xi\sqrt{d}\}\big).$$

For clarity let

$$z_{r,0} = \sigma(\langle \mathbf{w}_r, y_i\boldsymbol{\mu}\rangle) + \sigma(\langle \mathbf{w}_r, \boldsymbol{\xi}_i + \boldsymbol{\epsilon}_i\rangle)$$
$$z_{r,j} = \sigma(\langle \mathbf{w}_r, y_j\boldsymbol{\mu}\rangle) + \sigma(\langle \mathbf{w}_r, \boldsymbol{\xi}_j\rangle), \text{ for } j = 1, ..., M$$

Substituting the above results into (30) gives

$$\|\nabla L_S(\mathbf{W}^{(t)})\|_F^2 \leq \Big(\frac{1}{nm\tau}\sum_{i=1}^{n}\sum_{r=1}^{m}\sum_{j=0}^{M}\Big|\frac{\partial L_i(\mathbf{W})}{\partial F_j(\mathbf{W}, \mathbf{x}_i)}\Big|z_{r,j}\Big)^2 O\big(\max\{\|\boldsymbol{\mu}\|_2^2, \sigma_\xi^2 d\}\big)$$

$$\leq \Big(\frac{1}{n\tau}\sum_{i=1}^{n}\big(\sum_{j=0}^{M}\Big|\frac{\partial L_i(\mathbf{W})}{\partial F_j(\mathbf{W}, \mathbf{x}_i)}\Big|\big)\big(\frac{1}{m}\sum_{r=1}^{m}\sum_{j=0}^{M}z_{r,j}\big)\Big)^2 O\big(\max\{\|\boldsymbol{\mu}\|_2^2, \sigma_\xi^2 d\}\big),$$

where the second inequality is by $\sum_i a_i b_i \leq (\sum_i a_i)(\sum_i b_i)$ for $a_i, b_i \geq 0$.

Next we can verify that

$$\sum_{j=0}^{M}\Big|\frac{\partial L_i(\mathbf{W})}{\partial F_j(\mathbf{W}, \mathbf{x}_i)}\Big| = 1 - \frac{e^{F_0(\mathbf{W}, \mathbf{x}_i)}}{e^{F_0(\mathbf{W}, \mathbf{x}_i)} + \sum_{j=1}^{M} e^{F_j(\mathbf{W}, \mathbf{x}_i)}} + \sum_{j=1}^{M} \frac{e^{F_j(\mathbf{W}, \mathbf{x}_i)}}{e^{F_0(\mathbf{W}, \mathbf{x}_i)} + \sum_{j=1}^{M} e^{F_j(\mathbf{W}, \mathbf{x}_i)}}$$

$$= 2\big(1 - \frac{e^{F_0(\mathbf{W}, \mathbf{x}_i)}}{e^{F_0(\mathbf{W}, \mathbf{x}_i)} + \sum_{j=1}^{M} e^{F_j(\mathbf{W}, \mathbf{x}_i)}}\big)$$

$$\leq -6\log\big(\frac{e^{F_0(\mathbf{W}, \mathbf{x}_i)}}{e^{F_0(\mathbf{W}, \mathbf{x}_i)} + \sum_{j=1}^{M} e^{F_j(\mathbf{W}, \mathbf{x}_i)}}\big) = 6L_i(\mathbf{W}), \tag{31}$$

where the last inequality is by $1 - x \leq -3\log(x)$ for $x \in [0, 1]$. Furthermore, we can show

$$2\big(1 - \frac{e^{F_0(\mathbf{W}, \mathbf{x}_i)}}{e^{F_0(\mathbf{W}, \mathbf{x}_i)} + \sum_{j=1}^{M} e^{F_j(\mathbf{W}, \mathbf{x}_i)}}\big)\big(\frac{1}{m}\sum_{r=1}^{m}\sum_{j=0}^{M}z_{r,j}\big)^2 = O(1),$$

which leads to

$$\|\nabla L_S(\mathbf{W}^{(t)})\|_F^2 \leq \Big(\frac{1}{n\tau}\sum_{i=1}^{n}\sqrt{\big(\sum_{j=0}^{M}\Big|\frac{\partial L_i(\mathbf{W}^{(t)})}{\partial F_j(\mathbf{W}^{(t)}, \mathbf{x}_i)}\Big|\big)^2\big(\frac{1}{m}\sum_{r=1}^{m}\sum_{j=0}^{M}z_{r,j})^2\big)}\Big)^2 O\big(\max\{\|\boldsymbol{\mu}\|_2^2, \sigma_\xi^2 d\}\big)$$

$$\leq \Big(\frac{1}{n\tau}\sum_{i=1}^{n}\sqrt{\sum_{j=0}^{M}\Big|\frac{\partial L_i(\mathbf{W}^{(t)})}{\partial F_j(\mathbf{W}^{(t)}, \mathbf{x}_i)}\Big|}\Big)^2 O\big(\max\{\|\boldsymbol{\mu}\|_2^2, \sigma_\xi^2 d\}\big)$$

$$\leq O\big(\max\{\|\boldsymbol{\mu}\|_2^2, \sigma_\xi^2 d\}\big)\frac{1}{n}\sum_{i=1}^{n}L_i(\mathbf{W}^{(t)})$$

$$\leq O\big(\max\{\|\boldsymbol{\mu}\|_2^2, \sigma_\xi^2 d\}\big)L_S(\mathbf{W}^{(t)}),$$

where the third inequality is by (31) and Cauchy-Schwartz inequality. $\qquad\square$

We define
$$\mathbf{w}_r^* = \mathbf{w}_r^{(0)} + 2\tau \log(2M/\epsilon) \sum_{i=1}^n \frac{\boldsymbol{\xi}_i}{\|\boldsymbol{\xi}_i\|_2^2}.$$

Recall in the first stage
$$\mathbf{w}_r^{(T_1)} = \mathbf{w}_r^{(0)} + \gamma_r^{(T_1)} \|\boldsymbol{\mu}\|_2^{-2} \boldsymbol{\mu} + \sum_{i=1}^n \rho_{r,i}^{(T_1)} \|\boldsymbol{\xi}_i\|_2^{-2} \boldsymbol{\xi}_i,$$

and we have

- $\max_r |\gamma_r^{(t)}| = \widetilde{O}(n^{-1/2})$ for all $0 \le t \le T_1$.
- $\max_{r,i} \rho_{r,i}^{(T_1)} \ge 2$.

**Lemma C.15.** *Under Assumption 4.1, we have* $\|\mathbf{W}^{(T_1)} - \mathbf{W}^*\|_F \le \widetilde{O}(m^{1/2} n^{1/2} \sigma_\xi^{-1} d^{-1/2})$.

*Proof of Lemma C.15.* We have

$$
\begin{aligned}
\|\mathbf{W}^{(T_1)} - \mathbf{W}^*\|_F &\le \|\mathbf{W}^{(T_1)} - \mathbf{W}^{(0)}\|_F + \|\mathbf{W}^{(0)} - \mathbf{W}^*\|_F \\
&\le \sum_r \frac{\gamma_r^{(T_1)}}{\|\boldsymbol{\mu}\|_2} + O(\sqrt{m}) \max_r \Big\| \sum_{i=1}^n \rho_{r,i}^{(T_1)} \frac{\boldsymbol{\xi}_i}{\|\boldsymbol{\xi}_i\|_2^2} \Big\|_2 + O(m^{1/2} n^{1/2} \log(1/\epsilon) \sigma_\xi^{-1} d^{-1/2}) \\
&\le \widetilde{O}(m^{1/2} n^{1/2} \sigma_\xi^{-1} d^{-1/2}).
\end{aligned}
$$

$\square$

**Lemma C.16.** *Under Assumption 4.1, we have for all* $t \in [T_1, T^*]$
$$\langle \nabla F_0(\mathbf{W}^{(t)}, \mathbf{x}_i), \mathbf{W}^* \rangle \ge 2 \log(2M/\epsilon),$$
$$\langle \nabla F_j(\mathbf{W}^{(t)}, \mathbf{x}_i), \mathbf{W}^* \rangle \le \log(2M/\epsilon).$$

*Proof of Lemma C.16.* Based on the definition of $\mathbf{W}^*$ and $F_j(\mathbf{W}^{(t)}, \mathbf{x}_i)$, we can derive for $j = 0$,

$$\langle \nabla F_0(\mathbf{W}^{(t)}, \mathbf{x}_i), \mathbf{W}^* \rangle$$

$$= \sum_{r=1}^m \langle \nabla_{\mathbf{w}_r} F_0(\mathbf{W}^{(t)}, \mathbf{x}_i), \mathbf{w}_r^* \rangle$$

$$= \frac{1}{m\tau} \sum_{r=1}^m \sigma'(\langle \mathbf{w}_r^{(t)}, y_i \boldsymbol{\mu} \rangle) \sigma(\langle \mathbf{w}_r^{(t)}, y_i \boldsymbol{\mu} \rangle) \langle \mathbf{w}_r^*, y_i \boldsymbol{\mu} \rangle + \frac{1}{m\tau} \sum_{r=1}^m \sigma'(\langle \mathbf{w}_r^{(t)}, \boldsymbol{\xi}_i \rangle) \sigma(\langle \mathbf{w}_r^{(t)}, \boldsymbol{\xi}_i + \boldsymbol{\epsilon}_i \rangle) \langle \mathbf{w}_r^*, \boldsymbol{\xi}_i \rangle$$

$$= \frac{1}{m\tau} \sum_{r=1}^m \sigma'(\langle \mathbf{w}_r^{(t)}, y_i \boldsymbol{\mu} \rangle) \sigma(\langle \mathbf{w}_r^{(t)}, y_i \boldsymbol{\mu} \rangle) \Big( \langle \mathbf{w}_r^{(0)}, y_i \boldsymbol{\mu} \rangle + 2\tau \log(2M/\epsilon) \sum_{i=1}^n \langle \boldsymbol{\xi}_i, y_i \boldsymbol{\mu} \rangle \|\boldsymbol{\xi}_i\|_2^{-2} \Big)$$

$$+ \frac{1}{m\tau} \sum_{r=1}^m \sigma'(\langle \mathbf{w}_r^{(t)}, \boldsymbol{\xi}_i \rangle) \sigma(\langle \mathbf{w}_r^{(t)}, \boldsymbol{\xi}_i + \boldsymbol{\epsilon}_i \rangle) \Big( \langle \mathbf{w}_r^{(0)}, \boldsymbol{\xi}_i \rangle + 2\tau \log(2M/\epsilon) + 2\tau \log(2M/\epsilon) \sum_{i' \ne i} \langle \boldsymbol{\xi}_i, \boldsymbol{\xi}_{i'} \rangle \|\boldsymbol{\xi}_{i'}\|_2^{-2} \Big)$$

$$\ge \underbrace{\frac{1}{m\tau} \sum_{r=1}^m \sigma'(\langle \mathbf{w}_r^{(t)}, \boldsymbol{\xi}_i \rangle) \sigma(\langle \mathbf{w}_r^{(t)}, \boldsymbol{\xi}_i + \boldsymbol{\epsilon}_i \rangle) 2\tau \log(2M/\epsilon)}_{I_1} - \underbrace{\frac{1}{m\tau} \sum_{r=1}^m \sigma(\langle \mathbf{w}_r^{(t)}, y_i \boldsymbol{\mu} \rangle) \widetilde{O}(\sigma_0 \|\boldsymbol{\mu}\|_2)}_{I_2}$$

$$- \underbrace{\frac{1}{m\tau} \sum_{r=1}^m \sigma(\langle \mathbf{w}_r^{(t)}, y_i \boldsymbol{\mu} \rangle) 2\tau \log(2M/\epsilon) \widetilde{O}(n \|\boldsymbol{\mu}\|_2 \sigma_\xi^{-1} d^{-1})}_{I_3} - \underbrace{\frac{1}{m\tau} \sum_{r=1}^m \sigma(\langle \mathbf{w}_r^{(t)}, \boldsymbol{\xi}_i + \boldsymbol{\epsilon}_i \rangle) \widetilde{O}(\sigma_0 \sigma_\xi \sqrt{d})}_{I_4}$$

$$- \underbrace{\frac{1}{m\tau} \sum_{r=1}^m \sigma(\langle \mathbf{w}_r^{(t)}, \boldsymbol{\xi}_i + \boldsymbol{\epsilon}_i \rangle) 2\tau \log(2M/\epsilon) \widetilde{O}(n d^{-1/2})}_{I_5}$$

where the inequality is by Lemma B.5.

Next, we bound $I_1, I_2, I_3, I_4, I_5$ separately. For $I_1$, we take maximum over $r$, which results in

$$\frac{1}{m}\sum_{r=1}^{m}\sigma(\langle\mathbf{w}_r^{(t)},\boldsymbol{\xi}_i+\boldsymbol{\epsilon}_i\rangle) \geq \frac{1}{m}\sum_{r=1}^{m}\sigma(\frac{1}{2}\langle\mathbf{w}_r^{(0)},\boldsymbol{\xi}_i\rangle+\rho_{r,i}^{(t)}) \geq 2.$$

Thus we obtain

$$I_1 \geq 4\log(2M/\epsilon).$$

In addition, we can show by upper bound on $\rho_{r,i}^{(t)}$ and $\gamma_r^{(t)}$,

$$\langle\mathbf{w}_r^{(t)},\boldsymbol{\xi}_i+\boldsymbol{\epsilon}_i\rangle \leq \frac{3}{2}|\langle\mathbf{w}_r^{(0)},\boldsymbol{\xi}_i\rangle|+|\rho_{r,i}^{(t)}| \leq \widetilde{O}(1),$$

$$\langle\mathbf{w}_r^{(t)},\boldsymbol{\mu}\rangle \leq \frac{3}{2}|\langle\mathbf{w}_r^{(0)},\boldsymbol{\mu}\rangle|+|\gamma_r^{(t)}| \leq \widetilde{O}(1).$$

This implies

$$I_2 \leq \widetilde{O}(\sigma_0\|\boldsymbol{\mu}\|_2),\ I_3 \leq \log(2M/\epsilon)\widetilde{O}(nm\|\boldsymbol{\mu}\|_2\sigma_\xi^{-1}d^{-1}),\ I_4 \leq \widetilde{O}(\sigma_0\sigma_\xi\sqrt{d}),\ I_5 \leq \widetilde{O}(nmd^{-1/2}).$$

Based on the conditions on $\sigma_0, d$, we can show

$$\langle\nabla F_0(\mathbf{W}^{(t)},\mathbf{x}_i),\mathbf{W}^*\rangle \geq 4\log(2M/\epsilon) - I_2 - I_3 - I_4 - I_5 \geq 2\log(2M/\epsilon).$$

Now we prove for the claim for $F_j(\mathbf{W}^{(t)},\mathbf{W}^*)$ for $j = 1, ..., M$ as follows.

$$\langle\nabla F_j(\mathbf{W}^{(t)},\mathbf{x}_i),\mathbf{W}^*\rangle$$

$$= \frac{1}{m\tau}\sum_{r=1}^{m}\sigma'(\langle\mathbf{w}_r^{(t)},y_i\boldsymbol{\mu}\rangle)\sigma(\langle\mathbf{w}_r^{(t)},y_j\boldsymbol{\mu}\rangle)\langle\mathbf{w}_r^*,y_i\boldsymbol{\mu}\rangle + \frac{1}{m\tau}\sum_{r=1}^{m}\sigma'(\langle\mathbf{w}_r^{(t)},\boldsymbol{\xi}_i\rangle)\sigma(\langle\mathbf{w}_r^{(t)},\boldsymbol{\xi}_j\rangle)\langle\mathbf{w}_r^*,\boldsymbol{\xi}_i\rangle$$

$$= \frac{1}{m\tau}\sum_{r=1}^{m}\sigma'(\langle\mathbf{w}_r^{(t)},\boldsymbol{\xi}_i\rangle)\sigma(\langle\mathbf{w}_r^{(t)},\boldsymbol{\xi}_j\rangle)\langle\mathbf{w}_r^*,\boldsymbol{\xi}_i\rangle$$

$$\leq I_3 + I_4 + I_5 \leq \log(2M/\epsilon),$$

where the second equality is by $y_i \neq y_j$. $\qquad\qquad\square$

**Lemma C.17.** *Under Assumption 4.1, we have*

$$\|\mathbf{W}^{(t)}-\mathbf{W}^*\|_F^2 - \|\mathbf{W}^{(t+1)}-\mathbf{W}^*\| \geq \eta L_S(\mathbf{W}^{(t)}) - \eta\epsilon.$$

*Proof of Lemma C.17.* First, we verify that for $j = 0$,

$$\langle\nabla F_0(\mathbf{W}^{(t)},\mathbf{x}_i),\mathbf{W}^{(t)}\rangle = \sum_{r=1}^{m}\langle\nabla_{\mathbf{w}_r}F_0(\mathbf{W}^{(t)},\mathbf{x}_i),\mathbf{w}_r^{(t)}\rangle$$

$$= \frac{1}{m\tau}\sum_{r=1}^{m}\langle\sigma'(\langle\mathbf{w}_r,y_i\boldsymbol{\mu}\rangle)\mathrm{sg}(\sigma(\langle\mathbf{w}_r,y_i\boldsymbol{\mu}\rangle))y_i\boldsymbol{\mu},\mathbf{w}_r^{(t)}\rangle$$

$$+ \frac{1}{m\tau}\sum_{r=1}^{m}\langle\sigma'(\langle\mathbf{w}_r,\boldsymbol{\xi}_i\rangle)\mathrm{sg}(\sigma(\langle\mathbf{w}_r,\boldsymbol{\xi}_i+\boldsymbol{\epsilon}_i\rangle))\boldsymbol{\xi}_i,\mathbf{w}_r^{(t)}\rangle$$

$$= \frac{1}{m\tau}\sum_{r=1}^{m}\sigma(\langle\mathbf{w}_r,y_i\boldsymbol{\mu}\rangle)\mathrm{sg}(\sigma(\langle\mathbf{w}_r,y_i\boldsymbol{\mu}\rangle))$$

$$+ \frac{1}{m\tau}\sum_{r=1}^{m}\sigma(\langle\mathbf{w}_r,\boldsymbol{\xi}_i\rangle)\mathrm{sg}(\sigma(\langle\mathbf{w}_r,\boldsymbol{\xi}_i+\boldsymbol{\epsilon}_i\rangle))$$

$$= F_0(\mathbf{W}^{(t)},\mathbf{x}_i). \qquad\qquad (32)$$

Similarly, we can show for $j = 1, ..., M$, it satisfies that

$$\langle \nabla F_j(\mathbf{W}^{(t)}, \mathbf{x}_i), \mathbf{W}^{(t)} \rangle = F_j(\mathbf{W}^{(t)}, \mathbf{x}_i). \tag{33}$$

By the update of $\mathbf{W}^{(t)}$, we have

$$\|\mathbf{W}^{(t)} - \mathbf{W}^*\|_F^2 - \|\mathbf{W}^{(t+1)} - \mathbf{W}^*\|_F^2$$
$$= 2\eta \langle \nabla L_S(\mathbf{W}^{(t)}), \mathbf{W}^{(t)} - \mathbf{W}^* \rangle - \eta^2 \|\nabla L_S(\mathbf{W}^{(t)})\|_F^2$$
$$= \frac{2\eta}{n} \sum_{i=1}^n \sum_{j=0}^M \frac{\partial L_i(\mathbf{W}^{(t)})}{\partial F_j(\mathbf{W}^{(t)}, \mathbf{x}_i)} \langle \nabla F_j(\mathbf{W}^{(t)}, \mathbf{x}_i), \mathbf{W}^{(t)} - \mathbf{W}^* \rangle - \eta^2 \|\nabla L_S(\mathbf{W}^{(t)})\|_F^2$$
$$= \frac{2\eta}{n} \sum_{i=1}^n \sum_{j=0}^M \frac{\partial L_i(\mathbf{W}^{(t)})}{\partial F_j(\mathbf{W}^{(t)}, \mathbf{x}_i)} \left( F_j(\mathbf{W}^{(t)}, \mathbf{x}_i) - \langle \nabla F_j(\mathbf{W}^{(t)}, \mathbf{x}_i), \mathbf{W}^* \rangle \right) - \eta^2 \|\nabla L_S(\mathbf{W}^{(t)})\|_F^2$$
$$\geq \frac{2\eta}{n} \sum_{i=1}^n \left( \frac{\partial L_i(\mathbf{W}^{(t)})}{\partial F_0(\mathbf{W}^{(t)}, \mathbf{x}_i)} \left( F_0(\mathbf{W}^{(t)}, \mathbf{x}_i) - 2\log(2M/\epsilon) \right) + \sum_{j=1}^M \frac{\partial L_i(\mathbf{W}^{(t)})}{\partial F_j(\mathbf{W}^{(t)}, \mathbf{x}_i)} \left( F_j(\mathbf{W}^{(t)}, \mathbf{x}_i) - \log(2M/\epsilon) \right) \right)$$
$$\quad - \eta^2 \|\nabla L_S(\mathbf{W}^{(t)})\|_F^2$$
$$\geq \frac{2\eta}{n} \sum_{i=1}^n \left( L_i(\mathbf{W}^{(t)}) + \log\left( \frac{e^{2\log(2M/\epsilon)}}{e^{2\log(2M/\epsilon)} + Me^{\log(2M/\epsilon)}} \right) \right) - \eta^2 \|\nabla L_S(\mathbf{W}^{(t)})\|_F^2$$
$$= \frac{2\eta}{n} \sum_{i=1}^n \left( L_i(\mathbf{W}^{(t)}) - \log(1 + \frac{\epsilon}{2}) \right) - \eta^2 \|\nabla L_S(\mathbf{W}^{(t)})\|_F^2$$
$$\geq \eta L_S(\mathbf{W}^{(t)}) - \eta \epsilon,$$

where the third equality is by (32) and (33). The first inequality is by Lemma C.16. The second inequality is due to the convexity of negative log-Softmax function. The last inequality is by Lemma C.14 (and the conditions on $\eta$) and $\log(1 + x) \leq x$ for $x \geq 0$. $\qquad \square$

**Lemma C.18.** *Under Assumption 4.1, let $T = T_1 + \lfloor \frac{\|\mathbf{W}^{(T_1)} - \mathbf{W}^*\|_F^2}{\eta \epsilon} \rfloor = T_1 + \widetilde{O}(mn\sigma_\xi^{-2}d^{-1}\eta^{-1}\epsilon^{-1})$. Then we have $\max_r |\gamma_r^{(t)}| \leq \widetilde{O}(1/\sqrt{n})$ for all $T_1 \leq t \leq T$. In addition, we have*

$$\frac{1}{t - T_1 + 1} \sum_{s=T_1}^t L_S(\mathbf{W}^{(s)}) \leq \frac{\|\mathbf{W}^{(T_1)} - \mathbf{W}^*\|_F^2}{\eta(t - T_1 + 1)} + \epsilon$$

*for all $T_1 \leq t \leq T$. Thus there exists an iterate $\mathbf{W}^{(s)}$ for $s \in [T_1, T]$ with training loss smaller than $2\epsilon$.*

*Proof of Lemma C.18.* By Lemma C.17, for $t \in [T_1, T]$,

$$\|\mathbf{W}^{(s)} - \mathbf{W}^*\|_F^2 - \|\mathbf{W}^{(s+1)} - \mathbf{W}^*\| \geq \eta L_S(\mathbf{W}^{(t)}) - \eta \epsilon$$

for all $s \leq t$. Summing over the inequality and dividing both sides by $t - T_1 + 1$ yields

$$\frac{1}{t - T_1 + 1} \sum_{s=T_1}^t L_S(\mathbf{W}^{(s)}) \leq \frac{\|\mathbf{W}^{(T_1)} - \mathbf{W}^*\|_F^2 + \eta \epsilon(t - T_1 + 1)}{\eta(t - T_1 + 1)} = \frac{\|\mathbf{W}^{(T_1)} - \mathbf{W}^*\|_F^2}{\eta(t - T_1 + 1)} + \epsilon \leq 2\epsilon,$$

where the last inequality is by the definition of $T$, for all $T_1 \leq t \leq T$. In addition, we have

$$\sum_{t=T_1}^T L_S(\mathbf{W}^{(t)}) \leq \frac{2\|\mathbf{W}^{(T_1)} - \mathbf{W}^*\|_F^2}{\eta} = \widetilde{O}(\eta^{-1}mnd^{-1}\sigma_\xi^{-2}). \tag{34}$$

Next we prove the claim that $\max_r |\gamma_r^{(t)}| \leq 3\beta$, where $\beta = |\max_r \gamma_r^{(T_1)}| = \widetilde{O}(1/\sqrt{n})$ for all $T_1 \leq t \leq T$. Without loss of generality, we only consider $r \in \mathcal{U}_+^{(0)}$. First it is evident that at $t = T_1$,

we have $\max_r \gamma_r^{(t)} = \beta \leq 3\beta$. Next suppose there exists $\widetilde{T} \in [T_1, T]$ such that $\max_r \gamma_r^{(t)} \leq 3\beta$ for all $t \in [T_1, \widetilde{T} - 1]$. Then we let $\phi^{(t)} = \max_r \gamma_r^{(t)}$ and we have by the update of $\gamma_r^{(t)}$,

$$\phi^{(t+1)} \leq \phi^{(t)} + \frac{\eta}{nm\tau} \sum_{i:y_i=1} (1 - \ell_i'^{(t)}) \max_r (\frac{3}{2}\langle \mathbf{w}_r^{(0)}, \boldsymbol{\mu}\rangle + \phi^{(t)})\|\boldsymbol{\mu}\|_2^2$$

$$\leq \phi^{(t)} + \frac{\eta}{nm\tau} \sum_{i:y_i=1} (1 - \ell_i'^{(t)})(\frac{3}{2}\sqrt{2\log(8m/\delta)}\sigma_0\|\boldsymbol{\mu}\|_2 + \phi^{(t)})\|\boldsymbol{\mu}\|_2^2$$

$$\leq \phi^{(t)} + \frac{\eta}{m\tau} L_S(\mathbf{W}^{(t)})(\frac{3}{2}\sqrt{2\log(8m/\delta)}\sigma_0\|\boldsymbol{\mu}\|_2 + \phi^{(t)})\|\boldsymbol{\mu}\|_2^2,$$

where the third inequality is by (31).

Now summing over $t = T_1, ..., \widetilde{T} - 1$, we have

$$\phi^{(\widetilde{T})} \leq \phi^{(T_1)} + \sum_{t=T_1}^{\widetilde{T}-1} \frac{\eta}{m\tau} L_S(\mathbf{W}^{(t)})(\frac{3}{2}\sqrt{2\log(8m/\delta)}\sigma_0\|\boldsymbol{\mu}\|_2 + \phi^{(t)})\|\boldsymbol{\mu}\|_2^2$$

$$\leq \phi^{(T_1)} + O(\frac{\eta\|\boldsymbol{\mu}\|_2^2}{m\tau})\beta \sum_{t=T_1}^{\widetilde{T}-1} L_S(\mathbf{W}^{(t)})$$

$$\leq \phi^{(T_1)} + O(n\|\boldsymbol{\mu}\|_2^2 d^{-1}\sigma_\xi^{-2})\beta$$

$$\leq \phi^{(T_1)} + O(n\text{SNR}^2)\beta$$

$$\leq \phi^{(T_1)} + 2\beta \leq 3\beta$$

where the second inequality is by induction and the third inequality is by (34). The last inequality is by the condition of SNR. $\qquad\square$

### C.3 Downstream Task Performance

Recall that after the pre-training stage on the training data at time $T$, the signal learning and noise memorization satisfy

$$\max_r A_r^{(T)} = \widetilde{O}(1/\sqrt{n}),$$

$$\max_r \Psi_{r,i}^{(T)} = \widetilde{\Omega}(1) \text{ for } i \in [n].$$

Then, on the downstream task, the corresponding embedding can be calculated as follows:

$$h_r(\mathbf{x}_{\text{test}}^{(1)}) = \sigma(\langle \mathbf{w}_r^{(T)}, \mathbf{x}_{\text{test}}^{(1)}\rangle) = \widetilde{O}(1/\sqrt{dn}),$$

$$h_r(\mathbf{x}_{\text{test}}^{(2)}) = \sigma(\langle \mathbf{w}_r^{(T)}, \mathbf{x}_{\text{test}}^{(2)}\rangle) = \widetilde{\Omega}(1/\sqrt{d}).$$

Then, it is straightforward to check that the embedding of a finite size of samples during the fine-tuning stage is not linearly separable. Thus, the downstream task performance follows $L_{\mathcal{D}_{\text{test}}}(T^*) = \Theta(1)$.

## D  Multi-Modal Contrastive Learning: Proof of Theorem 4.3

### D.1  First Stage

Similar to the single-modal case, in the first stage, the loss derivative is close to its initial value for both modalities.

**Lemma D.1.** *If* $\max\{\gamma_r^{(t)}, \rho_{r,i}^{(t)}, \widetilde{\gamma}_r^{(t)}, \widetilde{\rho}_{r,i}^{(t)}\} = O(1)$*, there exists a constant* $C_\ell > 1$ *such that*

$$\frac{1}{C_\ell(1 + M)} \leq \ell_i'^{(t)} \leq \frac{C_\ell}{1 + M}$$

$$\frac{1}{C_\ell(M + 1)} \leq \ell_{i,j}'^{(t)} \leq \frac{C_\ell}{1 + M}$$

*for all* $i \in [n]$.

*Proof of Lemma D.1.* The proof follows from Lemma C.2. □

**Lemma D.2.** *Suppose that $\gamma_r^{(t)}, \widetilde{\gamma}_r^{(t)} = O(1)$ and $\rho_{r,i}^{(t)}, \widetilde{\rho}_{r,i}^{(t)} = O(1)$ for all $r \in [m]$ and $i \in [n]$. Under Assumption 4.1, then for any $\delta > 0$, with probability at least $1 - \delta$*

$$|\langle \mathbf{w}_r^{(t)} - \mathbf{w}_r^{(0)}, \boldsymbol{\xi}_i \rangle - \rho_{r,i}^{(t)}| \leq 5\sqrt{\frac{\log(6n^2/\delta)}{d}}n$$

$$|\langle \mathbf{w}_r^{(t)} - \mathbf{w}_r^{(0)}, \boldsymbol{\mu} \rangle - \gamma_r^{(t)}| \leq \mathrm{SNR}\sqrt{\frac{8\log(6n/\delta)}{d}}n,$$

$$|\langle \widetilde{\mathbf{w}}_r^{(t)} - \widetilde{\mathbf{w}}_r^{(0)}\widetilde{\boldsymbol{\xi}}_i \rangle - \widetilde{\rho}_{r,i}^{(t)}| \leq 5\sqrt{\frac{\log(6n^2/\delta)}{\widetilde{d}}}n$$

$$|\langle \widetilde{\mathbf{w}}_r^{(t)} - \widetilde{\mathbf{w}}_r^{(0)}, \widetilde{\boldsymbol{\mu}} \rangle - \widetilde{\gamma}_r^{(t)}| \leq \widetilde{\mathrm{SNR}}\sqrt{\frac{8\log(6n/\delta)}{\widetilde{d}}}n$$

*for all $r \in [m]$, $i \in [n]$.*

### D.1.1 Dynamics of Signal Learning: Lower Bound

We first analyze the dynamics of signal learning for both two modalities. Similar as in the single-modal learning, we partition the neurons, depending on the initialization, i.e., $\mathcal{U}_+^{(t)} = \{r \in [m] : \langle \mathbf{w}_r^{(t)}, \boldsymbol{\mu} \rangle > 0\}$ and $\mathcal{U}_-^{(t)} = \{r \in [m] : \langle \mathbf{w}_r^{(t)}, \boldsymbol{\mu} \rangle < 0\}$, $\widetilde{\mathcal{U}}_+^{(t)} = \{r \in [m] : \langle \widetilde{\mathbf{w}}_r^{(t)}, \widetilde{\boldsymbol{\mu}} \rangle > 0\}$ and $\widetilde{\mathcal{U}}_-^{(t)} = \{r \in [m] : \langle \widetilde{\mathbf{w}}_r^{(t)}, \widetilde{\boldsymbol{\mu}} \rangle < 0\}$.

**Lemma D.3.** *Under Assumption 4.1 and the same condition as Lemma D.2, for all $t > 0$, we have*

(1) $\mathcal{U}_+^{(t)} \cap \widetilde{\mathcal{U}}_+^{(t)} = \mathcal{U}_+^{(0)} \cap \widetilde{\mathcal{U}}_+^{(0)}$ *and* $\gamma_r^{(t)} \geq 0$, $\widetilde{\gamma}_r^{(t)} \geq 0$ *and are increasing.*

(2) $\mathcal{U}_-^{(t)} \cap \widetilde{\mathcal{U}}_-^{(t)} = \mathcal{U}_-^{(0)} \cap \widetilde{\mathcal{U}}_-^{(0)}$ *and* $\gamma_r^{(t)} \leq 0$, $\widetilde{\gamma}_r^{(t)} \leq 0$ *and are decreasing.*

(3) *For* $r \in \mathcal{U}_+^{(0)} \cap \widetilde{\mathcal{U}}_-^{(0)}$ *or* $r \in \mathcal{U}_-^{(0)} \cap \widetilde{\mathcal{U}}_+^{(0)}$, *there exists a time* $t' > 0$ *such that* $r \in \mathcal{U}_+^{(t')} \cap \widetilde{\mathcal{U}}_+^{(t')}$ *or* $r \in \mathcal{U}_-^{(t')} \cap \widetilde{\mathcal{U}}_-^{(t')}$.

*Proof of Lemma D.3.* We first analyze the neurons $r \in \mathcal{U}_+^{(t)} \cap \widetilde{\mathcal{U}}_+^{(t)}$. By the update of $\gamma_r^{(t)}$

$$\gamma_r^{(t+1)} = \gamma_r^{(t)} + \frac{\eta}{nm\tau}\sum_{i=1}^{n}(1 - \ell_i'^{(t)})\sigma(\langle \widetilde{\mathbf{w}}_r^{(t)}, y_i\widetilde{\boldsymbol{\mu}} \rangle)\sigma'(\langle \mathbf{w}_r^{(t)}, y_i\boldsymbol{\mu} \rangle)y_i\|\boldsymbol{\mu}\|_2^2$$

$$- \frac{\eta}{nm\tau}\sum_{i=1}^{n}\sum_{j\neq i}^{M}\ell_{i,j}'^{(t)}\sigma(\langle \widetilde{\mathbf{w}}_r^{(t)}, y_j\widetilde{\boldsymbol{\mu}} \rangle)\sigma'(\langle \mathbf{w}_r^{(t)}, y_i\boldsymbol{\mu} \rangle)y_i\|\boldsymbol{\mu}\|_2^2$$

$$= \gamma_r^{(t)} + \frac{\eta}{nm\tau}\sum_{i:y_i=1}(1 - \ell_i'^{(t)})\sigma(\langle \widetilde{\mathbf{w}}_r^{(t)}, \widetilde{\boldsymbol{\mu}} \rangle)\|\boldsymbol{\mu}\|_2^2 \geq \gamma_r^{(t)}, \tag{35}$$

where the second equality is by $y_i = y_j$ and the derivative of ReLU activation. Similarly for the other modality,

$$\widetilde{\gamma}_r^{(t+1)} = \widetilde{\gamma}_r^{(t)} + \frac{\eta}{nm\tau}\sum_{i=1}^{n}(1 - \ell_i'^{(t)})\sigma(\langle \mathbf{w}_r^{(t)}, y_i\boldsymbol{\mu} \rangle)\sigma'(\langle \widetilde{\mathbf{w}}_r^{(t)}, y_i\widetilde{\boldsymbol{\mu}} \rangle)y_i\|\widetilde{\boldsymbol{\mu}}\|_2^2$$

$$- \frac{\eta}{nm\tau}\sum_{i=1}^{n}\sum_{j\neq i}^{M}\ell_{i,j}'^{(t)}\sigma(\langle \mathbf{w}_r^{(t)}, y_j\boldsymbol{\mu} \rangle)\sigma'(\langle \widetilde{\mathbf{w}}_r^{(t)}, y_i\widetilde{\boldsymbol{\mu}} \rangle)y_i\|\widetilde{\boldsymbol{\mu}}\|_2^2$$

$$= \widetilde{\gamma}_r^{(t)} + \frac{\eta}{nm\tau}\sum_{i:y_i=1}(1 - \ell_i'^{(t)})\sigma(\langle \mathbf{w}_r^{(t)}, \boldsymbol{\mu} \rangle)\|\widetilde{\boldsymbol{\mu}}\|_2^2 \geq \widetilde{\gamma}_r^{(t)}. \tag{36}$$

Hence we see at $t = 0$, the claim is satisfied as $\gamma_r^{(1)} \geq \gamma_r^{(0)}$ for $r \in \mathcal{U}_+^{(0)} \cap \widetilde{\mathcal{U}}_+^{(0)}$. Then we prove the claim by induction. Suppose at iteration $t$ the claims are satisfied, i.e., $\gamma_r^{(t)} \geq \gamma_r^{(t-1)} \geq 0$ for $r \in \mathcal{U}_+^{(0)}$ and $r \in \mathcal{U}_+^{(t)}$. Then we can see from (35)

$$\gamma_r^{(t+1)} \geq \gamma_r^{(t)} \geq 0.$$

Similarly, we can show from (36) that $\widetilde{\gamma}_r^{(t+1)} \geq \widetilde{\gamma}_r^{(t)} \geq 0$.

Then for the inner product, we have

$$\langle \mathbf{w}_r^{(t+1)}, \boldsymbol{\mu} \rangle \overset{(a)}{\geq} \gamma_r^{(t+1)} + \langle \mathbf{w}_r^{(0)}, \boldsymbol{\mu} \rangle - \mathrm{SNR} \sqrt{\frac{8 \log(6n/\delta)}{d}} n$$

$$\overset{(b)}{\geq} 0.99 \langle \mathbf{w}_r^{(0)}, \boldsymbol{\mu} \rangle > 0,$$

where inequality (a) is by Lemma D.2, and inequality (b) is by Lemma B.2. Similarly, the same result holds for the other modality. Together, it shows $r \in \mathcal{U}_+^{(t+1)} \cap \widetilde{\mathcal{U}}_+^{(t+1)}$ and the induction is complete.

Similarly, we can use the same strategy to prove claim (2) for $r \in \mathcal{U}_-^{(0)} \cap \widetilde{\mathcal{U}}_-^{(0)}$.

Next, we analyze the case where $r \in \mathcal{U}_+^{(t)} \cap \widetilde{\mathcal{U}}_-^{(t)}$.

$$\gamma_r^{(t+1)} = \gamma_r^{(t)} + \frac{\eta}{nm\tau} \sum_{i=1}^{n} (1 - \ell_i'^{(t)}) \sigma(\langle \widetilde{\mathbf{w}}_r^{(t)}, y_i \widetilde{\boldsymbol{\mu}} \rangle) \sigma'(\langle \mathbf{w}_r^{(t)}, y_i \boldsymbol{\mu} \rangle) y_i \|\boldsymbol{\mu}\|_2^2$$

$$- \frac{\eta}{nm\tau} \sum_{i=1}^{n} \sum_{j \neq i}^{M} \ell_{i,j}'^{(t)} \sigma(\langle \widetilde{\mathbf{w}}_r^{(t)}, y_j \widetilde{\boldsymbol{\mu}} \rangle) \sigma'(\langle \mathbf{w}_r^{(t)}, y_i \boldsymbol{\mu} \rangle) y_i \|\boldsymbol{\mu}\|_2^2$$

$$= \gamma_r^{(t)} - \frac{\eta}{nm\tau} \sum_{i:y_i=1}^{n} \sum_{j \neq i}^{M} \ell_{i,j}'^{(t)} \sigma(\langle \widetilde{\mathbf{w}}_r^{(t)}, y_j \widetilde{\boldsymbol{\mu}} \rangle) \|\boldsymbol{\mu}\|_2^2 < \gamma_r^{(t)}, \tag{37}$$

where the second equality follows from $y_i \neq y_j$. Similarly,

$$\widetilde{\gamma}_r^{(t+1)} = \widetilde{\gamma}_r^{(t)} + \frac{\eta}{nm\tau} \sum_{i=1}^{n} (1 - \ell_i'^{(t)}) \sigma(\langle \mathbf{w}_r^{(t)}, y_i \boldsymbol{\mu} \rangle) \sigma'(\langle \widetilde{\mathbf{w}}_r^{(t)}, y_i \widetilde{\boldsymbol{\mu}} \rangle) y_i \|\widetilde{\boldsymbol{\mu}}\|_2^2$$

$$- \frac{\eta}{nm\tau} \sum_{i=1}^{n} \sum_{j \neq i}^{M} \ell_{i,j}'^{(t)} \sigma(\langle \mathbf{w}_r^{(t)}, y_j \boldsymbol{\mu} \rangle) \sigma'(\langle \widetilde{\mathbf{w}}_r^{(t)}, y_i \widetilde{\boldsymbol{\mu}} \rangle) y_i \|\widetilde{\boldsymbol{\mu}}\|_2^2$$

$$= \widetilde{\gamma}_r^{(t)} + \frac{\eta}{nm\tau} \sum_{i:y_i=-1}^{n} \sum_{j=1}^{M} \ell_{i,j}'^{(t)} \sigma(\langle \mathbf{w}_r^{(t)}, y_j \boldsymbol{\mu} \rangle) \|\widetilde{\boldsymbol{\mu}}\|_2^2 > \widetilde{\gamma}_r^{(t)}. \tag{38}$$

Then it is clear that the change of $\gamma_r^{(t)}, \widetilde{\gamma}_r^{(t)}$ does not align with the sign of its initialization. Therefore there exists some time $t' \geq 0$ such that either $r \in \mathcal{U}_+^{(t')} \cap \widetilde{\mathcal{U}}_+^{(t')}$ or $r \in \mathcal{U}_-^{(t')} \cap \widetilde{\mathcal{U}}_-^{(t')}$. We prove this claim by contradiction. Suppose for all $t \geq 0$, $r \in \mathcal{U}_+^{(t)} \cap \widetilde{\mathcal{U}}_-^{(t)}$, then by (37) and (38), we see $\gamma_r^{(t+1)} \leq \gamma_r^{(t)} \leq 0$ and $\widetilde{\gamma}_r^{(t+1)} \geq \widetilde{\gamma}_r^{(t)} \geq 0$. Because $\langle \mathbf{w}_r^{(t)}, \boldsymbol{\mu} \rangle \leq 1.01 \langle \mathbf{w}_r^{(0)}, \boldsymbol{\mu} \rangle + \gamma_r^{(t)}$ and $\langle \widetilde{\mathbf{w}}_r^{(t)}, \widetilde{\boldsymbol{\mu}} \rangle \geq 0.99 \langle \widetilde{\mathbf{w}}_r^{(0)}, \widetilde{\boldsymbol{\mu}} \rangle + \widetilde{\gamma}_r^{(t)}$, this raises a contradiction as either $\langle \mathbf{w}_r^{(t)}, \boldsymbol{\mu} \rangle \leq 0$ or $\langle \widetilde{\mathbf{w}}_r^{(t)}, \widetilde{\boldsymbol{\mu}} \rangle \geq 0$. $\square$

With Lemma D.3 at hand, we are ready to demonstrate the lower bound of the growth rate for signal learning.

**Lemma D.4.** *With the same condition as in Lemma C.2 and Lemma C.3 and $n \geq 2500 \log(4/\delta)$, define $A_r^{(t)} = \gamma_r^{(t)} + \langle \mathbf{w}_r^{(0)}, \boldsymbol{\mu} \rangle$ for $r \in \mathcal{U}_+^{(0)}$; and $A_r^{(t)} = -\gamma_r^{(t)} - \langle \mathbf{w}_r^{(0)}, \boldsymbol{\mu} \rangle$ for $r \in \mathcal{U}_-^{(0)}$. Similarly, we define $\widetilde{A}_r^{(t)} = \widetilde{\gamma}_r^{(t)} + \langle \widetilde{\mathbf{w}}_r^{(0)}, \widetilde{\boldsymbol{\mu}} \rangle$ for $r \in \widetilde{\mathcal{U}}_+^{(0)}$ and $\widetilde{A}_r^{(t)} = -\widetilde{\gamma}_r^{(t)} - \langle \widetilde{\mathbf{w}}_r^{(0)}, \widetilde{\boldsymbol{\mu}} \rangle$ for $r \in \widetilde{\mathcal{U}}_-^{(0)}$. Then with probability at least $1 - \delta$, we have*

$$A_r^{(t)} \geq \left(1 + \frac{0.48\eta \|\boldsymbol{\mu}\|_2^2 C_\mu}{m\tau}\right)^t (A_r^{(0)} + \widetilde{A}_r^{(0)}/C_\mu) - 1$$

$$\widetilde{A}_r^{(t)} \geq \left(1 + \frac{0.48\eta\|\boldsymbol{\mu}\|_2^2 C_\mu}{m\tau}\right)^t (C_\mu A_r^{(0)} + \widetilde{A}_r^{(0)}) - 1.$$

*Proof.* Without loss of generality, we consider $r \in \mathcal{U}_+^{(0)} \cap \widetilde{\mathcal{U}}_+^{(0)}$. Then by the update of $\gamma_r^{(t)}, \widetilde{\gamma}_r^{(t)}$, we have from (35) and (36)

$$\gamma_r^{(t+1)} = \gamma_r^{(t)} + \frac{\eta}{nm\tau} \sum_{i:y_i=1} (1 - \ell_i'^{(t)}) \sigma(\langle \widetilde{\mathbf{w}}_r^{(t)}, \widetilde{\boldsymbol{\mu}}\rangle) \|\boldsymbol{\mu}\|_2^2 \tag{39}$$

$$\widetilde{\gamma}_r^{(t+1)} = \widetilde{\gamma}_r^{(t)} + \frac{\eta}{nm\tau} \sum_{i:y_i=1} (1 - \ell_i'^{(t)}) \sigma(\langle \mathbf{w}_r^{(t)}, \boldsymbol{\mu}\rangle) \|\widetilde{\boldsymbol{\mu}}\|_2^2 \tag{40}$$

To achieve the lower bound, we define

$$A_r^{(t)} \triangleq \gamma_r^{(t)} + \langle \mathbf{w}_r^{(0)}, \boldsymbol{\mu}\rangle, \quad \widetilde{A}_r^{(t)} = \widetilde{\gamma}_r^{(t)} + \langle \widetilde{\mathbf{w}}_r^{(0)}, \widetilde{\boldsymbol{\mu}}\rangle.$$

By (39), we have the lower bound of update equation,

$$A_r^{(t+1)} = A_r^{(t)} + \frac{\eta}{nm\tau} \sum_{i:y_i=1} (1 - \ell_i'^{(t)}) \sigma(\langle \widetilde{\mathbf{w}}_r^{(t)}, \widetilde{\boldsymbol{\mu}}\rangle) \|\boldsymbol{\mu}\|_2^2$$

$$\geq A_r^{(t)} + \frac{\eta}{nm\tau} \left(\frac{n}{2} - O(\sqrt{n})\right)\left(1 - \frac{C_\ell}{1+M}\right) \widetilde{A}_r^{(t)} \|\boldsymbol{\mu}\|_2^2$$

$$\geq A_r^{(t)} + \frac{0.48\eta\|\boldsymbol{\mu}\|_2^2}{m\tau} \widetilde{A}_r^{(t)}.$$

The inequality is by Lemma B.3, Lemma D.1 where we recall $c_2 = 1 - \frac{C_\ell}{M+1}$. Similarly, by (40) we arrive at

$$\widetilde{A}_r^{(t+1)} \geq \widetilde{A}_r^{(t)} + \frac{0.48\eta\|\widetilde{\boldsymbol{\mu}}\|_2^2}{m\tau} A_r^{(t)} = \widetilde{A}_r^{(t)} + \frac{0.48\eta C_\mu^2 \|\boldsymbol{\mu}\|_2^2}{m\tau} A_r^{(t)}.$$

Then by combining the above two inequalities, we conclude that

$$C_\mu A_r^{(t+1)} + \widetilde{A}_r^{(t+1)} = C_\mu A_r^{(t)} + \widetilde{A}_r^{(t)} + \frac{0.48\eta\|\boldsymbol{\mu}\|_2^2}{m\tau} (C_\mu \widetilde{A}_r^{(t)} + C_\mu^2 A_r^{(t)})$$

$$\geq \left(1 + \frac{0.48\eta\|\boldsymbol{\mu}\|_2^2 C_\mu}{m\tau}\right)(C_\mu A_r^{(t)} + \widetilde{A}_r^{(t)}).$$

Thus, we see the joint dynamics of $\gamma_r^{(t)}$ and $\widetilde{\gamma}_r^{(t)}$ exhibits exponential growth, i.e.,

$$C_\mu A_r^{(t)} + \widetilde{A}_r^{(t)} \geq \left(1 + \frac{0.48\eta\|\boldsymbol{\mu}\|_2^2 C_\mu}{m\tau}\right)^t (A_r^{(0)} + \widetilde{A}_r^{(0)}). \tag{41}$$

To characterize the dynamics of $\gamma_r^{(t)}$ individually, we next track the dynamics of $C_\mu A_r^{(t)} - \widetilde{A}_r^{(t)}$ by subtracting (40) from (39), which gives

$$C_\mu A_r^{(t+1)} - \widetilde{A}_r^{(t+1)} = C_\mu A_r^{(t)} - \widetilde{A}_r^{(t)} + \frac{\eta\|\boldsymbol{\mu}\|_2^2}{nm\tau} \sum_{i:y_i=1} (1 - \ell_i'^{(t)})(C_\mu \sigma(\langle \mathbf{v}_r^{(t)}, \widetilde{\boldsymbol{\mu}}\rangle) - C_\mu^2 \sigma(\langle \mathbf{w}_r^{(t)}, \boldsymbol{\mu}\rangle))$$

$$\geq \left(1 - \frac{\eta C_\mu\|\boldsymbol{\mu}\|_2^2}{nm\tau} \sum_{i:y_i=1} (1 - \ell_i'^{(t)})\right)(C_\mu A_r^{(t)} - \widetilde{A}_r^{(t)}).$$

Then from the the propagation, we notice that the gap $C_\mu A_r^{(t)} - \widetilde{A}_r^{(t)}$ exhibits exponential decay regardless of the sign. Thus, we have

$$C_\mu A_r^{(t)} - \widetilde{A}_r^{(t)} \geq \left(1 - \frac{\eta C_\mu\|\boldsymbol{\mu}\|_2^2}{nm\tau} \sum_{i:y_i=1} (1 - \ell_i'^{(t)})\right)^t (C_\mu A_r^{(0)} - \widetilde{A}_r^{(0)}). \tag{42}$$

Therefore, combining (41) and (42) yields:

$$C_\mu A_r^{(t)} + \widetilde{A}_r^{(t)} \geq \left(1 + \frac{0.48\eta\|\boldsymbol{\mu}\|_2^2 C_\mu}{m\tau}\right)^t (C_\mu A_r^{(0)} + \widetilde{A}_r^{(0)}),$$

$$C_\mu A_r^{(t)} - \widetilde{A}_r^{(t)} \geq \left(1 - \frac{\eta\|\boldsymbol{\mu}\|_2^2}{nm\tau} \sum_{i:y_i=1} (1 - \ell_i'^{(t)})\right)^t \left(C_\mu A_r^{(0)} - \widetilde{A}_r^{(0)}\right).$$

Then it concludes that

$$A_r^{(t)} \geq \left(1 + \frac{0.48\eta\|\boldsymbol{\mu}\|_2^2 C_\mu}{m\tau}\right)^t (A_r^{(0)} + \widetilde{A}_r^{(0)}/C_\mu) + \left(1 - \frac{\eta\|\boldsymbol{\mu}\|_2^2}{nm\tau} \sum_{i:y_i=1} (1 - \ell_i'^{(t)})\right)^t \left(A_r^{(0)} - \widetilde{A}_r^{(0)}/C_\mu\right)$$

$$\geq \left(1 + \frac{0.48\eta\|\boldsymbol{\mu}\|_2^2 C_\mu}{m\tau}\right)^t (A_r^{(0)} + \widetilde{A}_r^{(0)}/C_\mu) - |A_r^{(0)} - \widetilde{A}_r^{(0)}/C_\mu|,$$

$$\widetilde{A}_r^{(t)} \geq \left(1 + \frac{0.48\eta\|\boldsymbol{\mu}\|_2^2 C_\mu}{m\tau}\right)^t (C_\mu A_r^{(0)} + \widetilde{A}_r^{(0)}) - \left(1 - \frac{\eta\|\boldsymbol{\mu}\|_2^2}{nm\tau} \sum_{i:y_i=1} (1 - \ell_i'^{(t)})\right)^t \left(C_\mu A_r^{(0)} - \widetilde{A}_r^{(0)}\right)$$

$$\geq \left(1 + \frac{0.48\eta\|\boldsymbol{\mu}\|_2^2 C_\mu}{m\tau}\right)^t (C_\mu A_r^{(0)} + \widetilde{A}_r^{(0)}) - |C_\mu A_r^{(0)} - \widetilde{A}_r^{(0)}|$$

where we use the fact that $\left(1 - \frac{\eta\|\boldsymbol{\mu}\|_2^2}{nm\tau} \sum_{i:y_i=1} (1 - \ell_i'^{(t)})\right)^t \leq 1$. Next, we derive an upper bound on $|A_r^{(0)} - \widetilde{A}_r^{(0)}/C_\mu|$, $|C_\mu A_r^{(0)} - \widetilde{A}_r^{(0)}|$ as follows.

$$|A_r^{(0)} - \widetilde{A}_r^{(0)}/C_\mu| \leq |A_r^{(0)}| + |\widetilde{A}_r^{(0)}|/C_\mu \leq 2\sqrt{2\log(8m/\delta)}\sigma_0\|\boldsymbol{\mu}\|_2 \leq 1$$

where we recall that $C_\mu\|\boldsymbol{\mu}\|_2 = \|\widetilde{\boldsymbol{\mu}}\|_2$ and the second inequality is by the condition on $\sigma_0 \leq 0.5(2\log(8m/\delta))^{-1/2}\|\boldsymbol{\mu}\|_2^{-1} = \widetilde{O}(\|\boldsymbol{\mu}\|_2^{-1})$. Similarly, we can show $|C_\mu A_r^{(0)} - \widetilde{A}_r^{(0)}| \leq 1$ and thus we obtain

$$A_r^{(t)} \geq \left(1 + \frac{0.48\eta\|\boldsymbol{\mu}\|_2^2 C_\mu}{m\tau}\right)^t (A_r^{(0)} + \widetilde{A}_r^{(0)}/C_\mu) - 1 \tag{43}$$

$$\widetilde{A}_r^{(t)} \geq \left(1 + \frac{0.48\eta\|\boldsymbol{\mu}\|_2^2 C_\mu}{m\tau}\right)^t (C_\mu A_r^{(0)} + \widetilde{A}_r^{(0)}) - 1 \tag{44}$$

$\square$

### D.1.2 Dynamics of Noise Memorization: Upper Bound

In order to characterize the growth of $\rho_{r,i}^{(t)}, \widetilde{\rho}_{r,i}^{(t)}$, we partition the samples according to its sign of inner product.

$$\mathcal{I}_{r,+}^{(t)} = \{i \in [n] : \langle \mathbf{w}_r^{(t)}, \boldsymbol{\xi}_i \rangle > 0\}, \quad \widetilde{\mathcal{I}}_{r,+}^{(t)} = \{i \in [n] : \langle \widetilde{\mathbf{w}}_r^{(0)}, \widetilde{\boldsymbol{\xi}}_i \rangle > 0\},$$
$$\mathcal{I}_{r,-}^{(t)} = \{i \in [n] : \langle \mathbf{w}_r^{(t)}, \boldsymbol{\xi}_i \rangle < 0\}, \quad \widetilde{\mathcal{I}}_{r,-}^{(t)} = \{i \in [n] : \langle \mathbf{v}_r^{(0)}, \widetilde{\boldsymbol{\xi}}_i \rangle < 0\}.$$

We further define

$$B_{r,+,+}^{(t)} \triangleq \sum_{i:y_i=1} (\rho_{r,i}^{(t)} + \langle \mathbf{w}_r^{(0)}, \boldsymbol{\xi}_i \rangle)\mathbb{1}_{i \in \mathcal{I}_{r,+}^{(t)} \cap \widetilde{\mathcal{I}}_{r,+}^{(t)}},$$

$$B_{r,+,-}^{(t)} \triangleq \sum_{i:y_i=1} (\rho_{r,i}^{(t)} + \langle \mathbf{w}_r^{(0)}, \boldsymbol{\xi}_i \rangle)\mathbb{1}_{i \in \mathcal{I}_{r,+}^{(t)} \cap \widetilde{\mathcal{I}}_{r,-}^{(t)}},$$

$$B_{r,-,+}^{(t)} \triangleq \sum_{i:y_i=-1} (\rho_{r,i}^{(t)} + \langle \mathbf{w}_r^{(0)}, \boldsymbol{\xi}_i \rangle)\mathbb{1}_{i \in \mathcal{I}_{r,+}^{(t)} \cap \widetilde{\mathcal{I}}_{r,+}^{(t)}},$$

$$B_{r,-,-}^{(t)} \triangleq \sum_{i:y_i=-1} (\rho_{r,i}^{(t)} + \langle \mathbf{w}_r^{(0)}, \boldsymbol{\xi}_i \rangle)\mathbb{1}_{i \in \mathcal{I}_{r,+}^{(t)} \cap \widetilde{\mathcal{I}}_{r,-}^{(t)}},$$

On the other hand, we define

$$\widetilde{B}_{r,+,+}^{(t)} \triangleq \sum_{i:y_i=1} (\widetilde{\rho}_{r,i}^{(t)} + \langle \widetilde{\mathbf{w}}_r^{(0)}, \widetilde{\boldsymbol{\xi}}_i \rangle)\mathbb{1}_{i \in \widetilde{\mathcal{I}}_{r,+}^{(t)} \cap \mathcal{I}_{r,+}^{(t)}},$$

$$\widetilde{B}_{r,+,-}^{(t)} \triangleq \sum_{i:y_i=1} (\widetilde{\rho}_{r,i}^{(t)} + \langle \widetilde{\mathbf{w}}_r^{(0)}, \widetilde{\boldsymbol{\xi}}_i \rangle)\mathbb{1}_{i \in \widetilde{\mathcal{I}}_{r,+}^{(t)} \cap \mathcal{I}_{r,-}^{(t)}},$$

$$\widetilde{B}_{r,-,+}^{(t)} \triangleq \sum_{i:y_i=-1} (\widetilde{\rho}_{r,i}^{(t)} + \langle \widetilde{\mathbf{w}}_r^{(0)}, \widetilde{\boldsymbol{\xi}}_i \rangle)\mathbb{1}_{i \in \widetilde{\mathcal{I}}_{r,+}^{(t)} \cap \mathcal{I}_{r,+}^{(t)}},$$

$$\widetilde{B}_{r,-,-}^{(t)} \triangleq \sum_{i:y_i=-1} (\widetilde{\rho}_{r,i}^{(t)} + \langle \widetilde{\mathbf{w}}_r^{(0)}, \widetilde{\boldsymbol{\xi}}_i \rangle) \mathbb{1}_{i \in \widetilde{\mathcal{I}}_{r,+}^{(t)} \cap \mathcal{I}_{r,-}^{(t)}}.$$

Before we state the main result, we provide some useful lemmas.

**Lemma D.5.** *Suppose $\delta > 0$, the with probability at least $1 - \delta$, we have*

$$B_{r,+,+}^{(0)} \le \frac{1}{2}\sigma_0\sigma_\xi\sqrt{dn}(1 + \sqrt{\log(1/\delta)/2}), \quad B_{r,-,+}^{(0)} \le \frac{1}{2}\sigma_0\sigma_\xi\sqrt{dn}(1 + \sqrt{\log(1/\delta)/2}).$$

$$\widetilde{B}_{r,+,+}^{(0)} \le \frac{1}{2}\sigma_0\sigma_{\widetilde{\xi}}\sqrt{dn}(1 + \sqrt{\log(1/\delta)/2}), \quad \widetilde{B}_{r,-,+}^{(0)} \le \frac{1}{2}\sigma_0\sigma_{\widetilde{\xi}}\sqrt{dn}(1 + \sqrt{\log(1/\delta)/2}).$$

*Proof.* By Bernstein's inequality, for arbitrary $t > 0$, we have

$$P(|B_{r,+,+}^{(0)} - \frac{1}{2}\sigma_0\sigma_\xi\sqrt{nd}| > t) \le \exp(-\frac{t^2}{2n/4\sigma_0^2\sigma_\xi^2 d}).$$

Setting $t = \frac{1}{2}\sigma_0\sigma_\xi\sqrt{nd\log(1/\delta)/2}$, we further have

$$B_{r,+}^{(0)} \le \frac{1}{2}\sigma_0\sigma_\xi\sqrt{dn}(1 + \sqrt{\log(1/\delta)/2}).$$

Similarly, the same result holds for $B_{r,-}^{(0)}$. $\qquad\qquad\square$

**Lemma D.6.** *Under the condition $d \ge \frac{300\sqrt{2n^3\log(6n^2/\delta)}}{\sigma_0\sigma_\xi\sqrt{\pi}\log(1/(1-(\delta/m)^2))}$, then with probability at least $1 - \delta$, it satisfies*

$$\widetilde{B}_{r,+,+}^{(0)} > 150\sqrt{\frac{\log(6n^2/\delta)}{d}}n^2, \quad B_{r,+,+}^{(0)} > 150\sqrt{\frac{\log(6n^2/\delta)}{d}}n^2.$$

*Proof of Lemma D.6.* Here we want to show $\widetilde{B}_{r,+,+}^{(0)} \ge 150\sqrt{\frac{\log(6n^2/\delta)}{d}}n^2$ with high probability. To see this, because $\widetilde{B}_{r,+,+}^{(0)}$ is a random variable with positive mean and variance $\sigma_0^2\sigma_\xi^2 dn/4$, by Lemma B.1, we compute

$$\mathbb{P}\big(\widetilde{B}_{r,+,+}^{(0)} \le t\big) \le \sqrt{1 - \exp\big(-\frac{8t^2}{\pi\sigma_0^2\sigma_\xi^2 dn}\big)},$$

Thus when $d \ge \frac{300\sqrt{2n^3\log(6n^2/\delta)}}{\sigma_0\sigma_\xi\sqrt{\pi}\log(1/(1-(\delta/m)^2))}$, we have $\mathbb{P}\big(\widetilde{B}_{r,+,+}^{(0)} \le 150\sqrt{\frac{\log(6n^2/\delta)}{d}}n^2\big) \le \delta/m$ and by union bound, we have with probability at least $1 - \delta$, it holds $\widetilde{B}_{r,+,+}^{(0)} > 150\sqrt{\frac{\log(6n^2/\delta)}{d}}n^2$. $\quad\square$

**Lemma D.7.** *Under the condition $n \ge \frac{8\log(1/\delta)}{\log(1/(1-(\delta/m)^2))}$, then with probability at least $1 - \delta$, it satisfies*

$$|\langle \mathbf{w}_r^{(0)}, \boldsymbol{\xi}_i \rangle| > \frac{B_{r,+,+}^{(0)}}{n}, \quad |\langle \widetilde{\mathbf{w}}_r^{(0)}, \widetilde{\boldsymbol{\xi}}_i \rangle| > \frac{\widetilde{B}_{r,+,+}^{(0)}}{n}.$$

*Proof of Lemma D.7.* Here we want to show $|\langle \mathbf{w}_r^{(0)}, \boldsymbol{\xi}_i \rangle| > \frac{B_{r,+,+}^{(0)}}{n}$ with high probability. To see this, because $\langle \mathbf{w}_r^{(0)}, \boldsymbol{\xi}_i \rangle$ is a random variable with positive mean and variance $\sigma_0^2\sigma_\xi^2 d$. Besides, by Lemma C.9, we have

$$\frac{B_{r,+,+}^{(0)}}{n} \le \frac{\sigma_0\sigma_\xi\sqrt{dn}(1 + \sqrt{\log(1/\delta)/2})}{n}.$$

By Lemma B.1, we compute

$$\mathbb{P}\big(|\langle \mathbf{w}_r^{(0)}, \boldsymbol{\xi}_i \rangle| \le t\big) \le \sqrt{1 - \exp\big(-\frac{8t^2}{\pi\sigma_0^2\sigma_\xi^2 d}\big)},$$

Thus when $n \geq \frac{8\log(1/\delta)}{\log(1/(1-(\delta/m)^2))}$, we have

$$\mathbb{P}\big(|\langle \mathbf{w}_r^{(0)}, \boldsymbol{\xi}_i\rangle| \leq \frac{\sigma_0 \sigma_\xi \sqrt{dn}(1+\sqrt{\log(1/\delta)/2})}{n}\big) \leq \delta/m$$

and by union bound, we have with probability at least $1 - \delta$, it holds $|\langle \mathbf{w}_r^{(0)}, \boldsymbol{\xi}_i\rangle| > \frac{B_{r,+,+}^{(0)}}{n}$.

Similarly, we can conclude that $|\langle \widetilde{\mathbf{w}}_r^{(0)}, \widetilde{\boldsymbol{\xi}}_i\rangle| > \frac{\widetilde{B}_{r,+,+}^{(0)}}{n}$. $\qquad\square$

**Lemma D.8.** *Under Assumption 4.1, with probability at least $1 - \delta$, we have*

$$B_{r,+,+}^{(t)} \leq (1 + \frac{1.05\eta}{nm\tau})^t B_{r,+,+}^{(0)}, \quad B_{r,+,-}^{(t)} \leq B_{r,+,-}^{(0)},$$

$$B_{r,-,+}^{(t)} \leq (1 + \frac{1.05\eta}{nm\tau})^t B_{r,-,+}^{(0)}, \quad B_{r,-,-}^{(t)} \leq B_{r,-,-}^{(0)},$$

$$\widetilde{B}_{r,+,+}^{(t)} \leq (1 + \frac{1.05\eta}{nm\tau})^t \widetilde{B}_{r,+,+}^{(0)}, \quad \widetilde{B}_{r,+,-}^{(t)} \leq \widetilde{B}_{r,+,-}^{(0)},$$

$$\widetilde{B}_{r,-,+}^{(t)} \leq (1 + \frac{1.05\eta}{nm\tau})^t \widetilde{B}_{r,-,+}^{(0)}, \quad \widetilde{B}_{r,-,-}^{(t)} \leq \widetilde{B}_{r,-,-}^{(0)}.$$

*Proof of Lemma D.8.* According to the iteration equations for $\rho_{r,i}^{(t)}$ and $\widetilde{\rho}_{r,i}^{(t)}$, we have

$$B_{r,+,+}^{(t+1)} = B_{r,+,+}^{(t)} + \frac{\eta}{nm\tau} \sum_{i:y_i=1} \left[ (1 - \ell_i'^{(t)})\langle \widetilde{\mathbf{w}}_r^{(t)}, \widetilde{\boldsymbol{\xi}}_i\rangle - \sum_{j:y_j=-1} \ell_{i,j}'^{(t)}\langle \widetilde{\mathbf{w}}_r^{(t)}, \widetilde{\boldsymbol{\xi}}_j\rangle \mathbb{1}_{j\in\widetilde{\mathcal{I}}_{r,+}} \right] \mathbb{1}_{i\in\mathcal{I}_{r,+}\cap\widetilde{\mathcal{I}}_{r,+}} \|\boldsymbol{\xi}_i\|_2^2$$

$$\leq B_{r,+,+}^{(t)} + \frac{\eta}{nm\tau} \sum_{i:y_i=1} \mathbb{1}_{i\in\mathcal{I}_{r,+}\cap\widetilde{\mathcal{I}}_{r,+}} \left[ \frac{C_\ell(M+1)-1}{C_\ell(M+1)}(\langle \widetilde{\mathbf{w}}_r^{(0)}, \widetilde{\boldsymbol{\xi}}_i\rangle + \widetilde{\rho}_{r,i}^{(t)} + 6\sqrt{\frac{\log(6n^2/\delta)}{d}}n) \right.$$

$$\left. - \sum_{j:y_j=-1} \frac{1}{C_\ell(M+1)}(\langle \widetilde{\mathbf{w}}_r^{(0)}, \widetilde{\boldsymbol{\xi}}_j\rangle + \widetilde{\rho}_{r,j}^{(t)} - 6\sqrt{\frac{\log(6n^2/\delta)}{d}}n)\mathbb{1}_{j\in\widetilde{\mathcal{I}}_{r,+}} \right](\sigma_\xi^2 d + \sigma_\xi^2\sqrt{d\log(6n^2/\delta)})$$

$$\leq B_{r,+,+}^{(t)} + \frac{\eta}{n\tau m} \left[ \frac{C_\ell(M+1)-1}{C_\ell(M+1)}(\widetilde{B}_{r,+,+}^{(t)} + \sum_{i:y_i=1}\mathbb{1}_{i\in\mathcal{I}_{r,+}\cap\widetilde{\mathcal{I}}_{r,+}} 6\sqrt{\frac{\log(6n^2/\delta)}{d}}n)) \right.$$

$$\left. - \sum_{i:y_i=1}\mathbb{1}_{i\in\mathcal{I}_{r,+}\cap\widetilde{\mathcal{I}}_{r,+}} \frac{1}{C_\ell(M+1)}(\widetilde{B}_{r,-,+}^{(t)} + \widetilde{B}_{r,-,-}^{(t)} - \sum_{j\neq i} 6\sqrt{\frac{\log(6n^2/\delta)}{d}}n) \right](\sigma_\xi^2 d + \sigma_\xi^2\sqrt{d\log(6n^2/\delta)})$$

$$\leq B_{r,+,+}^{(t)} + \frac{\eta}{nm\tau}\left[ 1.01 B_{r,+,+}^{(t)} - \frac{0.99^2}{4C_l}(\widetilde{B}_{r,-,+}^{(t)} + \widetilde{B}_{r,-,-}^{(t)}) \right] 1.035\sigma_\xi^2 d$$

$$\leq B_{r,+,+}^{(t)} + \frac{\eta}{nm\tau}\left[ 1.05 B_{r,+,+}^{(t)} - \frac{1}{4}(\widetilde{B}_{r,-,+}^{(t)} + \widetilde{B}_{r,-,-}^{(t)}) \right]\sigma_\xi^2 d,$$

where the first inequality is by Lemma C.2, Lemma C.1 and Lemma B.5. Furthermore, the second inequality is by Lemma C.8 and the definition of $\widetilde{B}_{r,-,+}^{(t)}$ and $\widetilde{B}_{r,-,-}^{(t)}$. The third inequality is by Lemma D.6, $\widetilde{B}_{r,+,+}^{(0)} > 150\sqrt{\frac{\log(6n^2/\delta)}{d}}n^2$, $d > 10000\log(6n^2/\delta)$, and $n \geq 1280000\log(4/\delta)$. The last inequality is by choosing $C_\ell = 1.01$.

Next, we establish the following inequality

$$B_{r,+,-}^{(t+1)} = B_{r,+,-}^{(t)} - \frac{\eta}{nm\tau} \sum_{i:y_i=1} \left[ \sum_{j:y_j=-1} \ell_{i,j}'^{(t)}\langle \widetilde{\mathbf{w}}_r^{(t)}, \widetilde{\boldsymbol{\xi}}_j\rangle \mathbb{1}_{j\in\widetilde{\mathcal{I}}_{r,+}} \right] \mathbb{1}_{i\in\mathcal{I}_{r,+}\cap\widetilde{\mathcal{I}}_{r,-}} \|\boldsymbol{\xi}_i\|_2^2$$

$$\leq B_{r,+,-}^{(t)} - \frac{\eta}{nm\tau} \sum_{i:y_i=1} \mathbb{1}_{i\in\mathcal{I}_{r,+}\cap\widetilde{\mathcal{I}}_{r,-}} \left[ \sum_{j:y_j=-1} \frac{1}{C_\ell(M+1)}(\langle \widetilde{\mathbf{w}}_r^{(0)}, \widetilde{\boldsymbol{\xi}}_j\rangle + \widetilde{\rho}_{r,j}^{(t)} - 6\sqrt{\frac{\log(6n^2/\delta)}{d}}n)\mathbb{1}_{j\in\widetilde{\mathcal{I}}_{r,+}} \right]$$

$$(\sigma_\xi^2 d - \sigma_\xi^2\sqrt{d\log(6n^2/\delta)})$$

$$\leq B_{r,+,-}^{(t)} - \frac{\eta}{n\tau m} \sum_{i:y_i=1} \left[ \mathbb{1}_{i\in\mathcal{I}_{r,+}\cap\widetilde{\mathcal{I}}_{r,+}} \frac{1}{C_\ell(M+1)} (\widetilde{B}_{r,-,+}^{(t)} + \widetilde{B}_{r,-,-}^{(t)} - \sum_{j\neq i} 6\sqrt{\frac{\log(6n^2/\delta)}{d}} n) \right]$$

$$(\sigma_\xi^2 d - \sigma_\xi^2 \sqrt{d\log(6n^2/\delta)})$$

$$\leq B_{r,+,-}^{(t)} - \frac{\eta}{nm\tau} \left[ \frac{0.99^2}{4C_l} (\widetilde{B}_{r,-,+}^{(t)} + \widetilde{B}_{r,-,-}^{(t)}) \right] 0.99\sigma_\xi^2 d$$

$$\leq B_{r,+,-}^{(t)} - \frac{\eta}{nm\tau} \left[ 0.24(\widetilde{B}_{r,-,+}^{(t)} + \widetilde{B}_{r,-,-}^{(t)}) \right] \sigma_\xi^2 d,$$

where the first inequality is by Lemma C.2, Lemma C.1 and Lemma B.5. Furthermore, the second inequality is by Lemma C.8 and the definition of $\widetilde{B}_{r,-,+}^{(t)}$ and $\widetilde{B}_{r,-,-}^{(t)}$. The third inequality is by Lemma D.6 and $\widetilde{B}_{r,+,+}^{(0)} > 150\sqrt{\frac{\log(6n^2/\delta)}{d}} n^2$, $d > 10000\log(6n^2/\delta)$, and $n \geq 1280000\log(4/\delta)$. Similarly, we have,

$$B_{r,-,+}^{(t+1)} \leq B_{r,-,+}^{(t)} + \frac{\eta}{nm\tau} \left[ 1.05 B_{r,-,+}^{(t)} - 0.25(\widetilde{B}_{r,+,+}^{(t)} + \widetilde{B}_{r,+,-}^{(t)}) \right] \sigma_\xi^2 d,$$

$$B_{r,-,-}^{(t+1)} \leq B_{r,-,-}^{(t)} - \frac{\eta}{nm\tau} \left[ 0.24(\widetilde{B}_{r,+,+}^{(t)} + \widetilde{B}_{r,+,-}^{(t)}) \right] \sigma_\xi^2 d$$

For the second modality, with the same derivative, we could obtain the following inequalities:

$$\widetilde{B}_{r,+,+}^{(t+1)} \leq \widetilde{B}_{r,+,+}^{(t)} + \frac{\eta}{nm\tau} \left[ 1.05 \widetilde{B}_{r,+,+}^{(t+1)} - 0.25(B_{r,-,+}^{(t)} + B_{r,-,-}^{(t)}) \right] \sigma_\xi^2 d,$$

$$\widetilde{B}_{r,+,-}^{(t+1)} \leq \widetilde{B}_{r,+,+}^{(t)} - \frac{\eta}{nm\tau} \left[ 0.24(B_{r,-,+}^{(t)} + B_{r,-,-}^{(t)}) \right] \sigma_\xi^2 d,$$

$$\widetilde{B}_{r,-,+}^{(t+1)} \leq \widetilde{B}_{r,-,+}^{(t)} + \frac{\eta}{nm\tau} \left[ 1.05 \widetilde{B}_{r,-,+}^{(t+1)} - 0.25(B_{r,+,+}^{(t)} + B_{r,+,-}^{(t)}) \right] \sigma_\xi^2 d,$$

$$\widetilde{B}_{r,-,+}^{(t+1)} \leq \widetilde{B}_{r,-,-}^{(t)} - \frac{\eta}{nm\tau} \left[ 0.24(B_{r,+,+}^{(t)} + B_{r,+,-}^{(t)}) \right] \sigma_\xi^2 d,$$

which completes the proof. □

We define

$$\Psi_{r,i}^{(t)} \triangleq \rho_{r,i}^{(t)} + \langle \mathbf{w}_r^{(0)}, \boldsymbol{\xi}_i \rangle, i \in \mathcal{I}_{r,+}^{(t)}$$

$$\widetilde{\Psi}_{r,i}^{(t)} \triangleq \rho_{r,i}^{(t)} + \langle \widetilde{\mathbf{w}}_r^{(0)}, \widetilde{\boldsymbol{\xi}}_i \rangle, i \in \widetilde{\mathcal{I}}_{r,+}^{(t)}$$

**Lemma D.9.** *Under Assumption 4.1, with probability at least $1 - \delta$, for all $0 \leq t \leq T_1$,*

*(1) for $i \in \mathcal{I}_{r,-}^{(0)} \cap \widetilde{\mathcal{I}}_{r,-}^{(0)}$, we have $\rho_{r,i}^{(t)} = 0$, $\mathcal{I}_{r,-}^{(t)} = \mathcal{I}_{r,-}^{(0)}$ and $\widetilde{\rho}_{r,i}^{(t)} = 0$, $\widetilde{\mathcal{I}}_{r,-}^{(t)} = \widetilde{\mathcal{I}}_{r,-}^{(0)}$.*

*(2) for $i \in \mathcal{I}_{r,-}^{(0)} \cap \widetilde{\mathcal{I}}_{r,+}^{(0)}$, we have $\rho_{r,i}^{(t)} = 0$, $\mathcal{I}_{r,-}^{(t)} = \mathcal{I}_{r,-}^{(0)}$ and $\widetilde{\rho}_{r,i}^{(t)} \leq 0$.*

*(3) for $i \in \mathcal{I}_{r,+}^{(0)} \cap \widetilde{\mathcal{I}}_{r,-}^{(0)}$, we have $\widetilde{\rho}_{r,i}^{(t)} = 0$, $\widetilde{\mathcal{I}}_{r,-}^{(t)} = \widetilde{\mathcal{I}}_{r,-}^{(0)}$ and $\rho_{r,i}^{(t)} \leq 0$.*

*(4) for $i \in \mathcal{I}_{r,+}^{(0)} \cap \widetilde{\mathcal{I}}_{r,+}^{(0)}$, we have*

$$\Psi_{r,i}^{(t)} \leq (1 + \frac{1.06\eta\sigma_\xi^2 d}{nm\tau})^t (\Psi_{r,i}^{(t)} + \widetilde{\Psi}_{r,i}^{(t)}), \tag{45}$$

$$\widetilde{\Psi}_{r,i}^{(t)} \geq \frac{101C_\ell}{M+1} (B_{r,+,+}^{(t)} + B_{r,+,-}^{(t)}). \tag{46}$$

*Proof of Lemma D.9.* We partition the dynamics of $\rho_{r,i}^{(t)}$ and $\widetilde{\rho}_{r,i}^{(t)}$ into one of the four cases according to its initialization (1) $i \in \mathcal{I}_{r,+}^{(0)} \cap \widetilde{\mathcal{I}}_{r,-}^{(0)}$ (2) $i \in \mathcal{I}_{r,-}^{(0)} \cap \widetilde{\mathcal{I}}_{r,+}^{(0)}$, (3) $i \in \mathcal{I}_{r,+}^{(0)} \cap \widetilde{\mathcal{I}}_{r,+}^{(0)}$, (4) $i \in \mathcal{I}_{r,-}^{(0)} \cap \widetilde{\mathcal{I}}_{r,-}^{(0)}$.

(1) In the case of $i \in \mathcal{I}_{r,-}^{(0)} \cap \widetilde{\mathcal{I}}_{r,-}^{(0)}$, it holds that

$$\rho_{r,i}^{(t)} = 0, \ \mathcal{I}_{r,-}^{(t)} = \mathcal{I}_{r,-}^{(0)}; \quad \widetilde{\rho}_{r,i}^{(t)} = 0, \ \widetilde{\mathcal{I}}_{r,-}^{(t)} = \widetilde{\mathcal{I}}_{r,-}^{(0)}.$$

The proof is by the induction method. It is clear when $t = 0$, $\rho_{r,i}^{(0)} = 0$ and $\widetilde{\rho}_{r,i}^{(0)} = 0$. Therefore, the induction argument holds at the initial step.

Suppose at iteration $t$, we have $\rho_{r,i}^{(t)} = 0$ and $\widetilde{\rho}_{r,i}^{(t)} = 0$. Then we have

$$\langle \widetilde{\mathbf{w}}_r^{(t)}, \widetilde{\boldsymbol{\xi}}_i \rangle \leq \frac{1}{2} \langle \widetilde{\mathbf{w}}_r^{(0)}, \widetilde{\boldsymbol{\xi}}_i \rangle + \widetilde{\rho}_{r,i}^{(t)} = \frac{1}{2} \langle \widetilde{\mathbf{w}}_r^{(0)}, \widetilde{\boldsymbol{\xi}}_i \rangle < 0,$$

$$\langle \mathbf{w}_r^{(t)}, \boldsymbol{\xi}_i \rangle \leq \frac{1}{2} \langle \mathbf{w}_r^{(0)}, \boldsymbol{\xi}_i \rangle + \rho_{r,i}^{(t)} = \frac{1}{2} \langle \mathbf{w}_r^{(0)}, \boldsymbol{\xi}_i \rangle < 0.$$

Thus by the update of $\widetilde{\rho}_{r,i}^{(t+1)}$, we have

$$\widetilde{\rho}_{r,i}^{(t+1)} = \widetilde{\rho}_{r,i}^{(t)} + \frac{\eta}{nm\tau}(1 - \ell_i'^{(t)})\sigma(\langle \mathbf{w}_r^{(t)}, \boldsymbol{\xi}_i \rangle)\sigma'(\langle \widetilde{\mathbf{w}}_r^{(t)}, \widetilde{\boldsymbol{\xi}}_i \rangle)\|\widetilde{\boldsymbol{\xi}}_i\|_2^2$$
$$- \frac{\eta}{nm\tau} \sum_{j \neq i}^{M} \ell_{i,j}'^{(t)}\sigma(\langle \mathbf{w}_r^{(t)}, \boldsymbol{\xi}_j \rangle)\sigma'(\langle \widetilde{\mathbf{w}}_r^{(t)}, \widetilde{\boldsymbol{\xi}}_i \rangle)\|\widetilde{\boldsymbol{\xi}}_i\|_2^2 = 0.$$

Here we have used $\langle \widetilde{\mathbf{w}}_r^{(t)}, \widetilde{\boldsymbol{\xi}}_i \rangle < 0$. Similarly,

$$\rho_{r,i}^{(t+1)} = \rho_{r,i}^{(t)} + \frac{\eta}{nm\tau}(1 - \ell_i'^{(t)})\sigma(\langle \widetilde{\mathbf{w}}_r^{(t)}, \widetilde{\boldsymbol{\xi}}_i \rangle)\sigma'(\langle \mathbf{w}_r^{(t)}, \boldsymbol{\xi}_i \rangle)\|\boldsymbol{\xi}_i\|_2^2$$
$$- \frac{\eta}{nm\tau} \sum_{j=1}^{M} \ell_{i,j}'^{(t)}\sigma(\langle \widetilde{\mathbf{w}}_r^{(t)}, \widetilde{\boldsymbol{\xi}}_j \rangle)\sigma'(\langle \mathbf{w}_r^{(t)}, \boldsymbol{\xi}_i \rangle)\|\boldsymbol{\xi}_i\|_2^2 = 0.$$

(2) In the case of $i \in \mathcal{I}_{r,-}^{(0)} \cap \widetilde{\mathcal{I}}_{r,+}^{(0)}$, We use induction. It is clear when $t = 0$, we have $\rho_{r,i}^{(0)} = 0$.

Suppose at iteration $t$, it holds that $\rho_{r,i}^{(t)} = 0$ Then by the propagation of $\rho_{r,i}^{(t)}$, we have

$$\rho_{r,i}^{(t+1)} = \rho_{r,i}^{(t)} + \frac{\eta}{nm\tau}(1 - \ell_i'^{(t)})\sigma(\langle \widetilde{\mathbf{w}}_r^{(t)}, \widetilde{\boldsymbol{\xi}}_i \rangle)\sigma'(\langle \mathbf{w}_r^{(t)}, \boldsymbol{\xi}_i \rangle)\|\boldsymbol{\xi}_i\|_2^2$$
$$- \frac{\eta}{nm\tau} \sum_{j \neq i}^{M} \ell_{i,j}'^{(t)}\sigma(\langle \widetilde{\mathbf{w}}_r^{(t)}, \widetilde{\boldsymbol{\xi}}_j \rangle)\sigma'(\langle \mathbf{w}_r^{(t)}, \boldsymbol{\xi}_i \rangle)\|\boldsymbol{\xi}_i\|_2^2 = 0.$$

On the other hand,

$$\widetilde{\rho}_{r,i}^{(t+1)} = \widetilde{\rho}_{r,i}^{(t)} + \frac{\eta}{nm\tau}(1 - \ell_i'^{(t)})\sigma(\langle \mathbf{w}_r^{(t)}, \boldsymbol{\xi}_i \rangle)\sigma'(\langle \widetilde{\mathbf{w}}_r^{(t)}, \widetilde{\boldsymbol{\xi}}_i \rangle)\|\widetilde{\boldsymbol{\xi}}_i\|_2^2$$
$$- \frac{\eta}{nm\tau} \sum_{j \neq i}^{M} \ell_{i,j}'^{(t)}\sigma(\langle \mathbf{w}_r^{(t)}, \boldsymbol{\xi}_j \rangle)\sigma'(\langle \widetilde{\mathbf{w}}_r^{(t)}, \widetilde{\boldsymbol{\xi}}_i \rangle)\|\widetilde{\boldsymbol{\xi}}_i\|_2^2$$
$$= \widetilde{\rho}_{r,i}^{(t)} - \frac{\eta}{nm\tau} \left[ \sum_{j \in \mathcal{I}_{r,+}} \ell_{i,j}'^{(t)}\sigma(\langle \mathbf{w}_r^{(t)}, \boldsymbol{\xi}_j \rangle) \right] \|\widetilde{\boldsymbol{\xi}}_i\|_2^2$$
$$\leq \widetilde{\rho}_{r,i}^{(t)},$$

where the first equation is by $\sigma(\langle \mathbf{w}_r^{(t)}, \boldsymbol{\xi}_i \rangle) \geq 0$.

(3) In the case of $i \in \mathcal{I}_{r,+}^{(0)} \cap \widetilde{\mathcal{I}}_{r,-}^{(0)}$, we use induction to prove such claim. It is clear when $t = 0$, $\rho_{r,i}^{(0)} = 0$. Suppose at iteration $t$, we have $\widetilde{\rho}_{r,i}^{(t)} = 0$. Then we have

$$\langle \widetilde{\mathbf{w}}_r^{(t)}, \widetilde{\boldsymbol{\xi}}_i \rangle \leq \frac{1}{2} \langle \widetilde{\mathbf{w}}_r^{(0)}, \widetilde{\boldsymbol{\xi}}_i \rangle + \widetilde{\rho}_{r,i}^{(t)} = \frac{1}{2} \langle \widetilde{\mathbf{w}}_r^{(0)}, \widetilde{\boldsymbol{\xi}}_i \rangle < 0.$$

Thus by the update of $\widehat{\rho}_{r,i}^{(t+1)}$:

$$\widehat{\rho}_{r,i}^{(t+1)} = \widehat{\rho}_{r,i}^{(t)} + \frac{\eta}{nm\tau}(1 - \ell_i'^{(t)})\sigma(\langle \mathbf{w}_r^{(t)}, \boldsymbol{\xi}_i \rangle)\sigma'(\langle \widetilde{\mathbf{w}}_r^{(t)}, \widetilde{\boldsymbol{\xi}}_i \rangle)\|\widetilde{\boldsymbol{\xi}}_i\|_2^2$$

$$- \frac{\eta}{nm\tau}\sum_{j \neq i}^{M} \ell_{i,j}'^{(t)}\sigma(\langle \mathbf{w}_r^{(t)}, \boldsymbol{\xi}_j \rangle)\sigma'(\langle \widetilde{\mathbf{w}}_r^{(t)}, \widetilde{\boldsymbol{\xi}}_i \rangle)\|\widetilde{\boldsymbol{\xi}}_i\|_2^2 = 0.$$

On the other hand,

$$\rho_{r,i}^{(t+1)} = \rho_{r,i}^{(t)} + \frac{\eta}{nm\tau}(1 - \ell_i'^{(t)})\sigma(\langle \widetilde{\mathbf{w}}_r^{(t)}, \widetilde{\boldsymbol{\xi}}_i \rangle)\sigma'(\langle \mathbf{w}_r^{(t)}, \boldsymbol{\xi}_i \rangle)\|\boldsymbol{\xi}_i\|_2^2$$

$$- \frac{\eta}{nm\tau}\sum_{j \neq i}^{M} \ell_{i,j}'^{(t)}\sigma(\langle \widetilde{\mathbf{w}}_r^{(t)}, \widetilde{\boldsymbol{\xi}}_j \rangle)\sigma'(\langle \mathbf{w}_r^{(t)}, \boldsymbol{\xi}_i \rangle)\|\boldsymbol{\xi}_i\|_2^2,$$

$$= \rho_{r,i}^{(t)} - \frac{\eta}{nm\tau}\left[\sum_{j \in \widetilde{\mathcal{I}}_{r,+}} \ell_{i,j}'^{(t)}\sigma(\langle \widetilde{\mathbf{w}}_r^{(t)}, \widetilde{\boldsymbol{\xi}}_j \rangle)\right]\|\boldsymbol{\xi}_i\|_2^2$$

$$\leq \rho_{r,i}^{(t)},$$

where the first equation is by $\sigma(\langle \widetilde{\mathbf{w}}_r^{(t)}, \widetilde{\boldsymbol{\xi}}_i \rangle) \geq 0$.

(4) Finally, in the case of $i \in \mathcal{I}_{r,+}^{(0)} \cap \widetilde{\mathcal{I}}_{r,+}^{(0)}$, we have

$$\Psi_{r,i}^{(t+1)} \leq \Psi_{r,i}^{(t)} + \frac{\eta}{nm\tau}(1 - \ell_i'^{(t)})\sigma(\langle \widetilde{\mathbf{w}}_r^{(t)}, \widetilde{\boldsymbol{\xi}}_i \rangle)\sigma'(\langle \mathbf{w}_r^{(t)}, \boldsymbol{\xi}_i \rangle)\|\boldsymbol{\xi}_i\|_2^2$$

$$\leq \Psi_{r,i}^{(t)} + \frac{1.05\eta\sigma_\xi^2 d}{nm\tau}\widetilde{\Psi}_{r,i}^{(t)}.$$

Similarly, for the other modality,

$$\widetilde{\Psi}_{r,i}^{(t+1)} \leq \widetilde{\Psi}_{r,i}^{(t)} + \frac{\eta}{nm\tau}(1 - \ell_i'^{(t)})\sigma(\langle \mathbf{w}_r^{(t)}, \boldsymbol{\xi}_i \rangle)\sigma'(\langle \widetilde{\mathbf{w}}_r^{(t)}, \widetilde{\boldsymbol{\xi}}_i \rangle)\|\widetilde{\boldsymbol{\xi}}_i\|_2^2$$

$$\leq \widetilde{\Psi}_{r,i}^{(t)} + \frac{1.05\eta\sigma_\xi^2 d}{nm\tau}\Psi_{r,i}^{(t)}.$$

Together, we can achieve that

$$\Psi_{r,i}^{(t)} + \widetilde{\Psi}_{r,i}^{(t)} \leq (1 + \frac{1.05\eta\sigma_\xi^2 d}{nm\tau})^t (\Psi_{r,i}^{(0)} + \widetilde{\Psi}_{r,i}^{(0)}). \tag{47}$$

On the other hand size, we calculate the upper bound of $\Psi_{r,i}^{(t)} - \widetilde{\Psi}_{r,i}^{(t)}$ as follows

$$\Psi_{r,i}^{(t+1)} - \widetilde{\Psi}_{r,i}^{(t+1)} = \Psi_{r,i}^{(t)} - \widetilde{\Psi}_{r,i}^{(t)} + \frac{\eta}{nm\tau}(1 - \ell_i'^{(t)})\sigma(\langle \widetilde{\mathbf{w}}_r^{(t)}, \widetilde{\boldsymbol{\xi}}_i \rangle)\sigma'(\langle \mathbf{w}_r^{(t)}, \boldsymbol{\xi}_i \rangle)\|\boldsymbol{\xi}_i\|_2^2$$

$$- \frac{\eta}{nm\tau}(1 - \ell_i'^{(t)})\sigma(\langle \mathbf{w}_r^{(t)}, \boldsymbol{\xi}_i \rangle)\sigma'(\langle \widetilde{\mathbf{w}}_r^{(t)}, \widetilde{\boldsymbol{\xi}}_i \rangle)\|\widetilde{\boldsymbol{\xi}}_i\|_2^2$$

$$- \frac{\eta}{nm\tau}\sum_{j \neq i}^{M} \ell_{i,j}'^{(t)}\sigma(\langle \widetilde{\mathbf{w}}_r^{(t)}, \widetilde{\boldsymbol{\xi}}_i \rangle)\sigma'(\langle \mathbf{w}_r^{(t)}, \boldsymbol{\xi}_j \rangle)\|\boldsymbol{\xi}_i\|_2^2$$

$$+ \frac{\eta}{nm\tau}\sum_{j \neq i}^{M} \ell_{i,j}'^{(t)}\sigma(\langle \mathbf{w}_r^{(t)}, \boldsymbol{\xi}_j \rangle)\sigma'(\langle \widetilde{\mathbf{w}}_r^{(t)}, \widetilde{\boldsymbol{\xi}}_i \rangle)\|\widetilde{\boldsymbol{\xi}}_i\|_2^2$$

$$\leq \Psi_{r,i}^{(t)} - \widetilde{\Psi}_{r,i}^{(t)} - \frac{0.96\eta\sigma_\xi^2 d}{nm\tau}(\Psi_{r,i}^{(t)} - \widetilde{\Psi}_{r,i}^{(t)}) + \frac{1.01\eta\sigma_\xi^2 d}{nm\tau}\frac{C_\ell}{M+1}(B_{r,+,+}^{(t)} + B_{r,+,-}^{(t)})$$

$$\leq (1 - \frac{0.95\eta\sigma_\xi^2 d}{nm\tau})(\Psi_{r,i}^{(t)} - \widetilde{\Psi}_{r,i}^{(t)}),$$

where the first inequality is by Lemma D.2 and Lemma D.1, the second inequality is by induction (46). Therefore we conclude that

$$\Psi_{r,i}^{(t)} - \widetilde{\Psi}_{r,i}^{(t)} \leq (1 - \frac{0.95\eta\sigma_\xi^2 d}{nm\tau})^t (\Psi_{r,i}^{(0)} - \widetilde{\Psi}_{r,i}^{(0)}). \tag{48}$$

Combining (47) and (48) yields

$$\Psi_{r,i}^{(t)} \leq (1 + \frac{1.05\eta\sigma_\xi^2 d}{nm\tau})^t(\Psi_{r,i}^{(0)} + \widetilde{\Psi}_{r,i}^{(0)}) + (1 - \frac{0.95\eta\sigma_\xi^2 d}{nm\tau})^t(\Psi_{r,i}^{(0)} - \widetilde{\Psi}_{r,i}^{(0)})$$

$$\leq (1 + \frac{1.06\eta\sigma_\xi^2 d}{nm\tau})^t(\Psi_{r,i}^{(0)} + \widetilde{\Psi}_{r,i}^{(0)}).$$

$\square$

### D.1.3 Signal Learning: Proof of Lemma 5.4

Before proving Lemma 5.4, we require the following lower bound for the initialization. Recall the definition that $A_r^{(t)} = \gamma_r^{(t)} + \langle \mathbf{w}_r^{(0)}, \boldsymbol{\mu} \rangle$ for $r \in \mathcal{U}_+^{(0)}$; and $A_r^{(t)} = -\gamma_r^{(t)} - \langle \mathbf{w}_r^{(0)}, \boldsymbol{\mu} \rangle$ for $r \in \mathcal{U}_-^{(0)}$. Similarly, we have $\widetilde{A}_r^{(t)} = \widetilde{\gamma}_r^{(t)} + \langle \widetilde{\mathbf{w}}_r^{(0)}, \widetilde{\boldsymbol{\mu}} \rangle$ for $r \in \widetilde{\mathcal{U}}_+^{(0)}$ and $\widetilde{A}_r^{(t)} = -\widetilde{\gamma}_r^{(t)} - \langle \widetilde{\mathbf{w}}_r^{(0)}, \widetilde{\boldsymbol{\mu}} \rangle$ for $r \in \widetilde{\mathcal{U}}_-^{(0)}$.

**Lemma D.10.** *Suppose $\delta > 0$ and $m \geq \widetilde{\Omega}(1)$. Then with probability at least $1 - \delta$, we have*

$$\frac{1}{m} \sum_{r \in \mathcal{U}_+^{(0)} \cap \widetilde{\mathcal{U}}_+^{(0)}} (A_r^{(0)} + \widetilde{A}_r^{(0)}/C_\mu) \geq 0.2\sigma_0\|\boldsymbol{\mu}\|_2$$

$$\frac{1}{m} \sum_{r \in \mathcal{U}_-^{(0)} \cap \widetilde{\mathcal{U}}_-^{(0)}} (A_r^{(0)} + \widetilde{A}_r^{(0)}/C_\mu) \geq 0.2\sigma_0\|\boldsymbol{\mu}\|_2$$

*Proof of Lemma D.10.* We first note that $\langle \mathbf{w}_r^{(0)}, \boldsymbol{\mu} \rangle \sim \mathcal{N}(0, \sigma_0^2\|\boldsymbol{\mu}\|_2^2)$ and $\langle \widetilde{\mathbf{w}}_r^{(0)}, \widetilde{\boldsymbol{\mu}} \rangle \sim \mathcal{N}(0, \sigma_0^2\|\widetilde{\boldsymbol{\mu}}\|_2^2)$. We define the event $\mathcal{A} = \{r \in [m] : \langle \mathbf{w}_r^{(0)}, \boldsymbol{\mu} \rangle > 0, \langle \widetilde{\mathbf{w}}_r^{(0)}, \widetilde{\boldsymbol{\mu}} \rangle > 0\}$. Then we can compute

$$\mathbb{E}[\langle \mathbf{w}_r^{(0)}, \boldsymbol{\mu} \rangle \mathbb{1}(\mathcal{A}) + \langle \widetilde{\mathbf{w}}_r^{(0)}, \widetilde{\boldsymbol{\mu}}/C_\mu \rangle \mathbb{1}(\mathcal{A})]$$

$$= \mathbb{E}[\langle \mathbf{w}_r^{(0)}, \boldsymbol{\mu} \rangle \mathbb{1}(\mathcal{A})] + \mathbb{E}[\langle \widetilde{\mathbf{w}}_r^{(0)}, \widetilde{\boldsymbol{\mu}}/C_\mu \rangle \mathbb{1}(\mathcal{A})]$$

$$= \frac{1}{2}\mathbb{E}[\langle \mathbf{w}_r^{(0)}, \boldsymbol{\mu} \rangle \mathbb{1}(\langle \mathbf{w}_r^{(0)}, \boldsymbol{\mu} \rangle > 0)] + \frac{1}{2}\mathbb{E}[\langle \widetilde{\mathbf{w}}_r^{(0)}, \widetilde{\boldsymbol{\mu}}/C_\mu \rangle \mathbb{1}(\langle \widetilde{\mathbf{w}}_r^{(0)}, \widetilde{\boldsymbol{\mu}} \rangle > 0)]$$

$$= \frac{\sigma_0\|\boldsymbol{\mu}\|_2}{\sqrt{2\pi}}$$

where we use the independence of neurons in two modalities. Let $S := \sum_{r=1}^m \langle \mathbf{w}_r^{(0)}, \boldsymbol{\mu} \rangle \mathbb{1}(\mathcal{A}) + \langle \widetilde{\mathbf{w}}_r^{(0)}, \widetilde{\boldsymbol{\mu}}/C_\mu \rangle \mathbb{1}(\mathcal{A})$. Then we apply the sub-Gaussian concentration inequality that with probability at least $1 - \delta/2$,

$$\left| \sum_{r \in \mathcal{A}} \left( \langle \mathbf{w}_r^{(0)}, \boldsymbol{\mu} \rangle + \langle \widetilde{\mathbf{w}}_r^{(0)}, \widetilde{\boldsymbol{\mu}}/C_\mu \rangle \right) - \frac{m\sigma_0\|\boldsymbol{\mu}\|_2}{\sqrt{2\pi}} \right| \leq \widetilde{O}(m^{-1/2}).$$

Then suppose $m = \widetilde{\Omega}(1)$, we have

$$\frac{1}{m} \sum_{r \in \mathcal{A}} \left( \langle \mathbf{w}_r^{(0)}, \boldsymbol{\mu} \rangle + \langle \widetilde{\mathbf{w}}_r^{(0)}, \widetilde{\boldsymbol{\mu}}/C_\mu \rangle \right) \geq 0.2\sigma_0\|\boldsymbol{\mu}\|_2.$$

Similarly, we can show the same for the event where $\langle \mathbf{w}_r^{(0)}, \boldsymbol{\mu} \rangle < 0, \langle \widetilde{\mathbf{w}}_r^{(0)}, \widetilde{\boldsymbol{\mu}} \rangle < 0$ and taking the union bound completes the proof. $\square$

*Proof of Lemma 5.4.* From the upper bound on noise memorization (45), we take the maximum over $r, i$, which gives

$$\max_{r,i} \Psi_{r,i}^{(t)} \leq \left(1 + 1.06\frac{\eta\sigma_\xi^2 d}{nm\tau}\right)^t \max_{r,i} \Psi_{r,i}^{(0)}$$

$$\leq \left(1 + 1.06\frac{\eta\sigma_\xi^2 d}{nm\tau}\right)^t 2\sqrt{\log(8mn/\delta)}\sigma_0\sigma_\xi\sqrt{d}.$$

At the same time, for signal learning, from the lower bound on signal learning in Lemma D.4, we have for the first modality that

$$A_r^{(t)} \geq \left(1 + \frac{0.48\eta\|\boldsymbol{\mu}\|_2^2 C_\mu}{m\tau}\right)^t (A_r^{(0)} + \widetilde{A}_r^{(0)}/C_\mu) - 1$$

Taking a summation over the $r \in \mathcal{U}_+^{(0)} \cap \widetilde{\mathcal{U}}_+^{(0)}$, we obtain

$$\frac{1}{m} \sum_{r \in \mathcal{U}_+^{(0)} \cap \widetilde{\mathcal{U}}_+^{(0)}} A_r^{(t)} \geq \left(1 + \frac{0.48\eta\|\boldsymbol{\mu}\|_2^2 C_\mu}{m\tau}\right)^t \frac{1}{m} \sum_{r \in \mathcal{U}_+^{(0)} \cap \widetilde{\mathcal{U}}_+^{(0)}} (A_r^{(0)} + \widetilde{A}_r^{(0)}/C_\mu) - 1$$

$$\geq \left(1 + \frac{0.48\eta\|\boldsymbol{\mu}\|_2^2 C_\mu}{m\tau}\right)^t 0.2\sigma_0\|\boldsymbol{\mu}\|_2 - 1$$

where the second inequality is due to Lemma D.10.

Under the SNR condition $n \cdot \text{SNR}^2 \geq 1.7$ and $C_\mu > 2.66$, we can see there exists a scale difference between $\max_{r,i} \Psi_{r,i}^{(t)}$ and $\frac{1}{m} \sum_{r \in \mathcal{U}_+^{(0)} \cap \widetilde{\mathcal{U}}_+^{(0)}} A_r^{(t)}$ at the end of first stage. Let

$$T_1 = \log\left(20/(\sigma_0\|\boldsymbol{\mu}\|_2)\right) / \log\left(1 + 0.48C_\mu \frac{\eta\|\boldsymbol{\mu}\|_2^2}{m\tau}\right).$$

Then we have $\frac{1}{m} \sum_{r \in \mathcal{U}_+^{(0)} \cap \widetilde{\mathcal{U}}_+^{(0)}} A_r^{(t)}$ reach 3 within $T_1$ iterations. Using similar analysis, we can show at the same time $\frac{1}{m} \sum_{r \in \mathcal{U}_-^{(0)} \cap \widetilde{\mathcal{U}}_-^{(0)}} A_r^{(t)}$, $\frac{1}{m} \sum_{r \in \mathcal{U}_+^{(0)} \cap \widetilde{\mathcal{U}}_+^{(0)}} \widetilde{A}_r^{(t)}$, $\frac{1}{m} \sum_{r \in \mathcal{U}_-^{(0)} \cap \widetilde{\mathcal{U}}_-^{(0)}} \widetilde{A}_r^{(t)}$ also reach 3.

On the other hand, we compute the scale of $\max_{r,i} \Psi_{r,i}^{(T_1)}$ as

$$\max_r \Psi_{r,i}^{(T_1)} \leq \left(1 + 1.06\frac{\eta\sigma_\xi^2 d}{nm\tau}\right)^t 2\sqrt{\log(8mn/\delta)}\sigma_0\sigma_\xi\sqrt{d}$$

$$= \exp\left(\frac{\log(1 + 1.06\frac{\eta\sigma_\xi^2 d}{nm\tau})}{\log(1 + 0.48C_\mu \frac{\eta\|\boldsymbol{\mu}\|_2^2}{m\tau})} \log\left(20/(\sigma_0\|\boldsymbol{\mu}\|_2)\right)\right) 2\sqrt{\log(8mn/\delta)}\sigma_0\sigma_\xi\sqrt{d}$$

$$\leq \exp\left((2.21/(C_\mu n \cdot \text{SNR}^2) + O((\frac{\eta\sigma_\xi^2 d}{nm\tau})^2)) \log\left(20/(\sigma_0\|\boldsymbol{\mu}\|_2)\right)\right) 2\sqrt{\log(8mn/\delta)}\sigma_0\sigma_\xi\sqrt{d}$$

$$\leq \exp\left((2.21/(C_\mu n \cdot \text{SNR}^2) + 0.01) \log\left(20/(\sigma_0\|\boldsymbol{\mu}\|_2)\right)\right) 2\sqrt{\log(8mn/\delta)}\sigma_0\sigma_\xi\sqrt{d}$$

$$\leq \exp\left(0.5 \log\left(20/(\sigma_0\|\boldsymbol{\mu}\|_2)\right)\right) 2\sqrt{\log(8mn/\delta)}\sigma_0\sigma_\xi\sqrt{d}$$

$$= \sqrt{24\log(8mn/\delta)} \frac{\sqrt{\sigma_0}\sigma_\xi\sqrt{d}}{\sqrt{\|\boldsymbol{\mu}\|_2}}$$

$$= \widetilde{O}(n^{-1/2}),$$

where we choose $\eta$ sufficiently small for the second inequality. In third inequality, we have applied the condition that $n\text{SNR}^2 = \Theta(1)$ and $\sigma_0 \leq \frac{1}{\|\boldsymbol{\mu}\|_2}$. The last inequality is by the SNR condition. Because we can choose $n \geq C\log(m/\delta)$ for sufficiently large constant $C$, $\max_{r,i} \Psi_{r,i}^{(T_1)} = o(1)$. $\square$

## D.2 Second Stage

We first show a similar result as in Lemma C.14 for both two modalities.

**Lemma D.11.** *Under conditions, for $0 \leq t \leq T^*$, we have*

$$\|\nabla L_S(\mathbf{W}^{(t)})\|_F^2 \leq O(\max\{\|\boldsymbol{\mu}\|_2^2, \sigma_\xi^2 d\}) L_S(\mathbf{W}^{(t)}),$$

$$\|\nabla L_S(\widetilde{\mathbf{W}}^{(t)})\|_F^2 \leq O(\max\{\|\widetilde{\boldsymbol{\mu}}\|_2^2, \sigma_\xi^2 d\}) L_S(\widetilde{\mathbf{W}}^{(t)}).$$

*Proof of Lemma D.11.* The proof follows from that of Lemma C.14 and hence is omitted for clarity.
$\square$

For notation convenience, we let

$$F_0(\mathbf{W}, \mathbf{x}_i) = \mathrm{Sim}_{\mathbf{h},\mathbf{g}}(\mathbf{x}_i, \widetilde{\mathbf{x}}_i)/\tau$$

$$= \frac{1}{m\tau}\sum_{r=1}^{m}\sigma(\langle \mathbf{w}_r, y_i\boldsymbol{\mu}\rangle)\mathrm{sg}(\sigma(\langle \widetilde{\mathbf{w}}_r, y_i\widetilde{\boldsymbol{\mu}}\rangle)) + \frac{1}{m\tau}\sum_{r=1}^{m}\sigma(\langle \mathbf{w}_r, \boldsymbol{\xi}_i\rangle)\mathrm{sg}(\sigma(\langle \widetilde{\mathbf{w}}_r, \widetilde{\boldsymbol{\xi}}_i\rangle))$$

$$F_j(\mathbf{W}, \mathbf{x}_i) = \mathrm{Sim}_{\mathbf{h},\mathbf{g}}(\mathbf{x}_i, \widetilde{\mathbf{x}}_j)/\tau$$

$$= \frac{1}{m\tau}\sum_{r=1}^{m}\sigma(\langle \mathbf{w}_r, y_i\boldsymbol{\mu}\rangle)\mathrm{sg}(\sigma(\langle \widetilde{\mathbf{w}}_r, y_j\widetilde{\boldsymbol{\mu}}\rangle)) + \frac{1}{m\tau}\sum_{r=1}^{m}\sigma(\langle \mathbf{w}_r, \boldsymbol{\xi}_i\rangle)\mathrm{sg}(\sigma(\langle \widetilde{\mathbf{w}}_r, \widetilde{\boldsymbol{\xi}}_j\rangle)), \text{ for } j = 1, ..., M$$

$$F_0(\widetilde{\mathbf{W}}, \widetilde{\mathbf{x}}_i) = \mathrm{Sim}_{\mathbf{g},\mathbf{h}}(\widetilde{\mathbf{x}}_i, \mathbf{x}_i)/\tau$$

$$= \frac{1}{m\tau}\sum_{r=1}^{m}\mathrm{sg}(\sigma(\langle \mathbf{w}_r, y_i\boldsymbol{\mu}\rangle))\sigma(\langle \widetilde{\mathbf{w}}_r, y_i\widetilde{\boldsymbol{\mu}}\rangle) + \frac{1}{m\tau}\sum_{r=1}^{m}\mathrm{sg}(\sigma(\langle \mathbf{w}_r, \boldsymbol{\xi}_i\rangle))\sigma(\langle \widetilde{\mathbf{w}}_r, \widetilde{\boldsymbol{\xi}}_i\rangle)$$

$$\widetilde{F}_j(\widetilde{\mathbf{W}}, \widetilde{\mathbf{x}}_i) = \mathrm{Sim}_{\mathbf{g},\mathbf{h}}(\widetilde{\mathbf{x}}_i, \mathbf{x}_j)/\tau$$

$$= \frac{1}{m\tau}\sum_{r=1}^{m}\mathrm{sg}(\sigma(\langle \mathbf{w}_r, y_j\boldsymbol{\mu}\rangle))\sigma(\langle \widetilde{\mathbf{w}}_r, y_i\widetilde{\boldsymbol{\mu}}\rangle) + \frac{1}{m\tau}\sum_{r=1}^{m}\mathrm{sg}(\sigma(\langle \mathbf{w}_r, \boldsymbol{\xi}_j\rangle))\sigma(\langle \widetilde{\mathbf{w}}_r, \widetilde{\boldsymbol{\xi}}_i\rangle), \text{ for } j = 1, ..., M$$

It is worth mentioning that $F_j(\mathbf{W}, \mathbf{x}_i) = \widetilde{F}_j(\widetilde{\mathbf{W}}, \widetilde{\mathbf{x}}_i)$ in terms of numerical values. They differ in terms of the derivatives.

We further denote

$$L_S(\mathbf{W}) = -\frac{1}{n}\sum_{i=1}^{n}L_i(\mathbf{W}) = -\frac{1}{n}\sum_{i=1}^{n}\log\left(\frac{e^{\mathrm{Sim}_{\mathbf{h},\mathbf{g}}(\mathbf{x}_i,\widetilde{\mathbf{x}}_i)/\tau}}{e^{\mathrm{Sim}_{\mathbf{h},\mathbf{g}}(\mathbf{x}_i,\widetilde{\mathbf{x}}_i)/\tau} + \sum_{j\neq i}^{M}e^{\mathrm{Sim}_{\mathbf{h},\mathbf{g}}(\mathbf{x}_i,\widetilde{\mathbf{x}}_j)/\tau}}\right),$$

where $L_i(\mathbf{W}) = -\log\left(\frac{e^{\mathrm{Sim}_{\mathbf{h},\mathbf{g}}(\mathbf{x}_i,\widetilde{\mathbf{x}}_i)/\tau}}{e^{\mathrm{Sim}_{\mathbf{h},\mathbf{g}}(\mathbf{x}_i,\widetilde{\mathbf{x}}_i)/\tau} + \sum_{j\neq i}^{M}e^{\mathrm{Sim}_{\mathbf{h},\mathbf{g}}(\mathbf{x}_i,\mathbf{x}_j)/\tau}}\right) = -\log\left(\frac{e^{F_0(\mathbf{W},\mathbf{x}_i)}}{e^{F_0(\mathbf{W},\mathbf{x}_i)} + \sum_{j=1}^{M}e^{F_j(\mathbf{W},\mathbf{x}_i)}}\right)$

$$L_S(\widetilde{\mathbf{W}}) = -\frac{1}{n}\sum_{i=1}^{n}L_i(\widetilde{\mathbf{W}}) = -\frac{1}{n}\sum_{i=1}^{n}\log\left(\frac{e^{\mathrm{Sim}_{\mathbf{g},\mathbf{h}}(\widetilde{\mathbf{x}}_i,\mathbf{x}_i)/\tau}}{e^{\mathrm{Sim}_{\mathbf{g},\mathbf{h}}(\widetilde{\mathbf{x}}_i,\mathbf{x}_i)/\tau} + \sum_{j\neq i}^{M}e^{\mathrm{Sim}_{\mathbf{g},\mathbf{h}}(\widetilde{\mathbf{x}}_i,\mathbf{x}_j)/\tau}}\right),$$

where $L_i(\widetilde{\mathbf{W}}) = -\log\left(\frac{e^{\mathrm{Sim}_{\mathbf{g},\mathbf{h}}(\widetilde{\mathbf{x}}_i,\mathbf{x}_i)/\tau}}{e^{\mathrm{Sim}_{\mathbf{g},\mathbf{h}}(\widetilde{\mathbf{x}}_i,\mathbf{x}_i)/\tau} + \sum_{j\neq i}^{M}e^{\mathrm{Sim}_{\mathbf{g},\mathbf{h}}(\widetilde{\mathbf{x}}_i,\mathbf{x}_j)/\tau}}\right) = -\log\left(\frac{e^{F_0(\widetilde{\mathbf{W}},\widetilde{\mathbf{x}}_i)}}{e^{F_0(\widetilde{\mathbf{W}},\widetilde{\mathbf{x}}_i)} + \sum_{j=1}^{M}e^{F_j(\widetilde{\mathbf{W}},\widetilde{\mathbf{x}}_i)}}\right)$

$$\overline{L}(\mathbf{W}, \mathbf{V}) = L_S(\mathbf{W}) + L_S(\widetilde{\mathbf{W}})$$

Here $\overline{L}(\mathbf{W}, \widetilde{\mathbf{W}})$ is the combined loss function for two modalities.

Let $\theta_r = 1$ if $r \in \mathcal{U}_+^{(0)}$, i.e., $\langle \mathbf{w}_r^{(0)}, \boldsymbol{\mu}\rangle > 0$ and $\theta_r = -1$ if $r \in \mathcal{U}_-^{(0)}$, i.e., $\langle \mathbf{w}_r^{(0)}, \boldsymbol{\mu}\rangle < 0$. Similarly, we let $\widetilde{\theta}_r = 1$ if $r \in \widetilde{\mathcal{U}}_+^{(0)}$ and $\widetilde{\theta}_r = -1$ if $r \in \widetilde{\mathcal{U}}_-^{(0)}$. Then we define

$$\mathbf{w}_r^* = \mathbf{w}_r^{(0)} + 2\tau\log(2M/\epsilon)\cdot\theta_r\cdot\frac{\boldsymbol{\mu}}{\|\boldsymbol{\mu}\|_2^2},$$

$$\widetilde{\mathbf{w}}_r^* = \widetilde{\mathbf{w}}_r^{(0)} + 2\tau\log(2M/\epsilon)\cdot\widetilde{\theta}_r\cdot\frac{\widetilde{\boldsymbol{\mu}}}{\|\widetilde{\boldsymbol{\mu}}\|_2^2}.$$

**Lemma D.12.** *Under Assumption 4.1, we have* $\|\mathbf{W}^{(T_1)} - \mathbf{W}^*\|_F \leq \widetilde{O}(m^{1/2}\|\boldsymbol{\mu}\|_2^{-1})$ *and* $\|\widetilde{\mathbf{W}}^{(T_1)} - \widetilde{\mathbf{W}}^*\|_F \leq \widetilde{O}(m^{1/2}\|\widetilde{\boldsymbol{\mu}}\|_2^{-1})$

*Proof of Lemma D.12.* The proof follows exactly the same as the proof in single-modal case. we include it here for completeness. Without loss of generality, we focus on the case for $\mathbf{W}^{(T_1)}$.

By the scale difference at $T_1$, we have

$$\|\mathbf{W}^{(T_1)} - \mathbf{W}^*\|_F \leq \|\mathbf{W}^{(T_1)} - \mathbf{W}^{(0)}\|_F + \|\mathbf{W}^{(0)} - \mathbf{W}^*\|_F$$

$$\leq \sum_{r}\frac{\gamma_r^{(T_1)}}{\|\boldsymbol{\mu}\|_2} + \sum_{r,i}\frac{|\rho_{r,i}^{(T_1)}|}{\|\boldsymbol{\xi}_i\|_2} + O(m^{1/2}\log(1/\epsilon))\|\boldsymbol{\mu}\|_2^{-1}$$

$$\leq O(m\|\boldsymbol{\mu}\|_2^{-1}) + O(nm\sigma_0) + O(m^{1/2}\log(1/\epsilon)\|\boldsymbol{\mu}\|_2^{-1})$$
$$\leq \widetilde{O}(m^{1/2}\|\boldsymbol{\mu}\|_2^{-1})$$

where the first inequality is by triangle inequality and the second inequality is by decomposition of $\mathbf{W}^{(T_1)}$ and $\mathbf{W}^*$. The third inequality is by the bound on $\gamma_r^{(T_1)}$ and $\rho_{r,i}^{(T_1)}$ and Lemma B.5. The last inequality is by condition on $\sigma_0$. $\qquad\square$

**Lemma D.13.** *Under Assumption 4.1, we have for all $t \in [T_1, T^*]$,*

$$\langle \nabla F_0(\mathbf{W}^{(t)}, \mathbf{x}_i), \mathbf{W}^* \rangle \geq 2\log(2M/\epsilon)$$
$$\langle \nabla F_j(\mathbf{W}^{(t)}, \mathbf{x}_i), \mathbf{W}^* \rangle \leq \log(2M/\epsilon), \text{ for } j = 1, ..., M$$
$$\langle \nabla F_0(\widetilde{\mathbf{W}}^{(t)}, \mathbf{x}_i), \widetilde{\mathbf{W}}^* \rangle \geq 2\log(2M/\epsilon)$$
$$\langle \nabla F_j(\widetilde{\mathbf{W}}^{(t)}, \mathbf{x}_j), \widetilde{\mathbf{W}}^* \rangle \leq \log(2M/\epsilon), \text{ for } j = 1, ..., M$$

*Proof of Lemma D.13.* The proof follows similarly from Lemma C.16 and here we only show the result for the first modality. Based on the definition of $\mathbf{W}^*$ and $F_j(\mathbf{W}^{(t)}, \mathbf{x}_i)$, we can derive for $j = 0$,

$$\langle \nabla F_0(\mathbf{W}^{(t)}, \mathbf{x}_i), \mathbf{W}^* \rangle$$
$$= \sum_{r=1}^{m} \langle \nabla_{\mathbf{w}_r} F_0(\mathbf{W}^{(t)}, \mathbf{x}_i), \mathbf{w}_r^* \rangle$$
$$= \frac{1}{m\tau} \sum_{r=1}^{m} \sigma'(\langle \mathbf{w}_r^{(t)}, y_i\boldsymbol{\mu}\rangle)\sigma(\langle \widetilde{\mathbf{w}}_r^{(t)}, y_i\widetilde{\boldsymbol{\mu}}\rangle)\langle \mathbf{w}_r^*, y_i\boldsymbol{\mu}\rangle + \frac{1}{m\tau} \sum_{r=1}^{m} \sigma'(\langle \mathbf{w}_r^{(t)}, \boldsymbol{\xi}_i\rangle)\sigma(\langle \widetilde{\mathbf{w}}_r^{(t)}, \widetilde{\boldsymbol{\xi}}_i\rangle)\langle \mathbf{w}_r^*, \boldsymbol{\xi}_i\rangle$$
$$= \frac{1}{m\tau} \sum_{r=1}^{m} \sigma'(\langle \mathbf{w}_r^{(t)}, y_i\boldsymbol{\mu}\rangle)\sigma(\langle \widetilde{\mathbf{w}}_r^{(t)}, y_i\widetilde{\boldsymbol{\mu}}\rangle)\Big(\langle \mathbf{w}_r^{(0)}, y_i\boldsymbol{\mu}\rangle + 2\tau\log(2M/\epsilon)\theta_r y_i\Big)$$
$$+ \frac{1}{m\tau} \sum_{r=1}^{m} \sigma'(\langle \mathbf{w}_r^{(t)}, \boldsymbol{\xi}_i\rangle)\sigma(\langle \widetilde{\mathbf{w}}_r^{(t)}, \widetilde{\boldsymbol{\xi}}_i\rangle)\Big(\langle \mathbf{w}_r^{(0)}, \boldsymbol{\xi}_i\rangle + 2\tau\theta_r\log(2M/\epsilon)\langle \boldsymbol{\xi}_i, y_i\boldsymbol{\mu}\rangle\|\boldsymbol{\mu}\|_2^{-2}\Big)$$
$$\geq \underbrace{\frac{1}{m\tau} \sum_{r=1}^{m} \sigma'(\langle \mathbf{w}_r^{(t)}, y_i\boldsymbol{\mu}\rangle)\sigma(\langle \widetilde{\mathbf{w}}_r^{(t)}, y_i\widetilde{\boldsymbol{\mu}}\rangle)2\tau\log(2M/\epsilon)\theta_r y_i}_{I_1} - \underbrace{\frac{1}{m\tau} \sum_{r=1}^{m} \sigma(\langle \widetilde{\mathbf{w}}_r^{(t)}, y_i\widetilde{\boldsymbol{\mu}}\rangle)\widetilde{O}(\sigma_0\|\boldsymbol{\mu}\|_2)}_{I_2}$$
$$- \underbrace{\frac{1}{m\tau} \sum_{r=1}^{m} \sigma(\langle \widetilde{\mathbf{w}}_r^{(t)}, \widetilde{\boldsymbol{\xi}}_i\rangle)2\tau\log(2M/\epsilon)\widetilde{O}(\sigma_\xi\|\boldsymbol{\mu}\|_2^{-1})}_{I_3} - \underbrace{\frac{1}{m\tau} \sum_{r=1}^{m} \sigma(\langle \widetilde{\mathbf{w}}_r^{(t)}, \widetilde{\boldsymbol{\xi}}_i\rangle)\widetilde{O}(\sigma_0\sigma_\xi\sqrt{d})}_{I_4}.$$

First, we can bound $I_2 \leq \widetilde{O}(\sigma_0\|\boldsymbol{\mu}\|_2)$, $I_3 \leq \log(2M/\epsilon)\widetilde{O}(\sigma_\xi\|\boldsymbol{\mu}\|_2^{-1})$, $I_4 \leq \widetilde{O}(\sigma_0\sigma_\xi\sqrt{d})$ by the global bound on $\sigma(\langle \widetilde{\mathbf{w}}_r^{(t)}, \widetilde{\boldsymbol{\xi}}_i\rangle), \sigma(\langle \widetilde{\mathbf{w}}_r^{(t)}, y_i\widetilde{\boldsymbol{\mu}}\rangle) = \widetilde{O}(1)$.

Further, we lower bound $I_1$ as follows. Without loss of generality, we suppose $y_i = 1$, then we have

$$I_1 \geq \frac{1}{m\tau} \sum_{r \in \mathcal{U}_+^{(0)} \cap \widetilde{\mathcal{U}}_+^{(0)}} \sigma(\langle \widetilde{\mathbf{w}}_r^{(t)}, y_i\widetilde{\boldsymbol{\mu}}\rangle)2\tau\log(2M/\epsilon) \geq 4\log(2M/\epsilon)$$

where the last inequality is by Lemma 5.4 and the monotonicity of $\widetilde{\gamma}_r^{(t)}$.

Then we can obtain

$$\langle \nabla F_0(\mathbf{W}^{(t)}, \mathbf{x}_i), \mathbf{W}^* \rangle \geq 4\log(2M/\epsilon) - I_2 - I_3 - I - 4 \geq 2\log(2M/\epsilon).$$

The proof for $F_j(\mathbf{W}^{(t)}, \mathbf{W}^*)$ follows the same argument as in Lemma C.16. $\qquad\square$

**Lemma D.14.** *Under Assumption 4.1, we have for all $t \in [T_1, T^*]$,*

$$\|\mathbf{W}^{(t)} - \mathbf{W}^*\|_F^2 + \|\widetilde{\mathbf{W}}^{(t)} - \widetilde{\mathbf{W}}^*\|_F^2 - \|\mathbf{W}^{(t+1)} - \mathbf{W}^*\|_F^2 - \|\widetilde{\mathbf{W}}^{(t+1)} - \widetilde{\mathbf{W}}^*\|_F^2 \geq \eta\overline{L}(\mathbf{W}^{(t)}, \widetilde{\mathbf{W}}^{(t)}) - 2\eta\epsilon$$

*Proof of Lemma D.14.* First, we see that $\overline{L}(\mathbf{W}^{(t)}, \widetilde{\mathbf{W}}^{(t)}) = L_S(\mathbf{W}^{(t)}) + L_S(\widetilde{\mathbf{W}}^{(t)})$ is decomposable in terms of $\mathbf{W}^{(t)}$ and $\widetilde{\mathbf{W}}^{(t)}$. This suggests that $\nabla_{\mathbf{W}} \overline{L}(\mathbf{W}^{(t)}, \widetilde{\mathbf{W}}^{(t)}) = \nabla L_S(\mathbf{W}^{(t)})$ and $\nabla_{\widetilde{\mathbf{W}}} \overline{L}(\mathbf{W}^{(t)}, \widetilde{\mathbf{W}}^{(t)}) = \nabla \widetilde{L}_S(\widetilde{\mathbf{W}}^{(t)})$. Then following similar analysis as in Lemma C.17, we can first show

$$\langle F_j(\mathbf{W}^{(t)}, \mathbf{x}_i), \mathbf{W}^{(t)} \rangle = F_j(\mathbf{W}^{(t)}, \mathbf{x}_i), \text{ for } j = 0, ..., M, \tag{49}$$

$$\langle F_j(\widetilde{\mathbf{W}}^{(t)}, \widetilde{\mathbf{x}}_i), \widetilde{\mathbf{W}}^{(t)} \rangle = F_j(\widetilde{\mathbf{W}}^{(t)}, \widetilde{\mathbf{x}}_i), \text{ for } j = 0, ..., M. \tag{50}$$

Then by the gradient descent update

$$\|\mathbf{W}^{(t)} - \mathbf{W}^*\|_F^2 - \|\mathbf{W}^{(t+1)} - \mathbf{W}^*\|_F^2$$

$$= 2\eta \langle \nabla L_S(\mathbf{W}^{(t)}), \mathbf{W}^{(t)} - \mathbf{W}^* \rangle - \eta^2 \|\nabla L_S(\mathbf{W}^{(t)})\|_F^2$$

$$= \frac{2\eta}{n} \sum_{i=1}^{n} \sum_{j=0}^{M} \frac{\partial L_i(\mathbf{W}^{(t)})}{\partial F_j(\mathbf{W}^{(t)}, \mathbf{x}_i)} \langle \nabla F_j(\mathbf{W}^{(t)}, \mathbf{x}_i), \mathbf{W}^{(t)} - \mathbf{W}^* \rangle - \eta^2 \|\nabla L_S(\mathbf{W}^{(t)})\|_F^2$$

$$= \frac{2\eta}{n} \sum_{i=1}^{n} \sum_{j=0}^{M} \frac{\partial L_i(\mathbf{W}^{(t)})}{\partial F_j(\mathbf{W}^{(t)}, \mathbf{x}_i)} \Big( F_j(\mathbf{W}^{(t)}, \mathbf{x}_i) - \langle \nabla F_j(\mathbf{W}^{(t)}, \mathbf{x}_i), \mathbf{W}^* \rangle \Big) - \eta^2 \|\nabla L_S(\mathbf{W}^{(t)})\|_F^2$$

$$\geq \frac{2\eta}{n} \sum_{i=1}^{n} \Big( \frac{\partial L_i(\mathbf{W}^{(t)})}{\partial F_0(\mathbf{W}^{(t)}, \mathbf{x}_i)} \big( F_0(\mathbf{W}^{(t)}, \mathbf{x}_i) - 2\log(2M/\epsilon) \big) + \sum_{j=1}^{M} \frac{\partial L_i(\mathbf{W}^{(t)})}{\partial F_j(\mathbf{W}^{(t)}, \mathbf{x}_i)} \big( F_j(\mathbf{W}^{(t)}, \mathbf{x}_i) - \log(2M/\epsilon) \big) \Big)$$

$$- \eta^2 \|\nabla L_S(\mathbf{W}^{(t)})\|_F^2$$

$$\geq \frac{2\eta}{n} \sum_{i=1}^{n} \Big( L_i(\mathbf{W}^{(t)}) + \log\big( \frac{e^{2\log(2M/\epsilon)}}{e^{2\log(2M/\epsilon)} + Me^{\log(2M/\epsilon)}} \big) \Big) - \eta^2 \|\nabla L_S(\mathbf{W}^{(t)})\|_F^2$$

$$= \frac{2\eta}{n} \sum_{i=1}^{n} \big( L_i(\mathbf{W}^{(t)}) - \log(1 + \frac{\epsilon}{2}) \big) - \eta^2 \|\nabla L_S(\mathbf{W}^{(t)})\|_F^2$$

$$\geq \eta L_S(\mathbf{W}^{(t)}) - \eta\epsilon$$

where the third equality is by (49). The first inequality is by Lemma D.13. The second inequality is due to the convexity of negative log-Softmax function. The last inequality is by Lemma D.11 (and the conditions on $\eta$) and $\log(1 + x) \leq x$ for $x \geq 0$.

Similarly, we can show the same for the other modality as

$$\|\widetilde{\mathbf{W}}^{(t)} - \widetilde{\mathbf{W}}^*\|_F^2 - \|\widetilde{\mathbf{W}}^{(t+1)} - \widetilde{\mathbf{W}}^*\|_F^2 \geq \eta L_S(\widetilde{\mathbf{W}}^{(t)}) - \eta\epsilon$$

Combining the two results completes the proof. $\qquad\square$

**Lemma D.15.** *Under Assumption 4.1, let* $T = T_1 + \lfloor \frac{\|\mathbf{W}^{(T_1)} - \mathbf{W}^*\|_F^2 + \|\widetilde{\mathbf{W}}^{(T_1)} - \widetilde{\mathbf{W}}^*\|_F^2}{\eta\epsilon} \rfloor = T_1 + \widetilde{O}(m\eta^{-1}\epsilon^{-1}\|\boldsymbol{\mu}\|_2^{-2})$. *Then we have* $\max_{r,i} |\rho_{r,i}^{(t)}| \leq \sigma_0 \sigma_\xi \sqrt{d}$ *and* $\max_{r,i} |\widetilde{\rho}_{r,i}^{(t)}| \leq \sigma_0 \sigma_\xi \sqrt{d}$ *for all* $t \in [T_1, T]$. *In addition, we have for all* $T_1 \leq t \leq T$,

$$\frac{1}{t - T_1 + 1} \sum_{s=T_1}^{t} \overline{L}(\mathbf{W}^{(t)}, \widetilde{\mathbf{W}}^{(t)}) \leq \frac{\|\mathbf{W}^{(T_1)} - \mathbf{W}^*\|_F^2 + \|\widetilde{\mathbf{W}}^{(T_1)} - \widetilde{\mathbf{W}}^*\|_F^2}{\eta(t - T_1 + 1)} + 2\epsilon.$$

*Therefore, we can find an iterate* $(\mathbf{W}^{(s)}, \widetilde{\mathbf{W}}^{(s)})$ *for* $s \in [T_1, T]$ *with training loss smaller than* $3\epsilon$.

*Proof of Lemma D.15.* By Lemma D.14, for $t \in [T_1, T]$, we have for any $s \leq t$

$$\|\mathbf{W}^{(s)} - \mathbf{W}^*\|_F^2 + \|\widetilde{\mathbf{W}}^{(s)} - \widetilde{\mathbf{W}}^*\|_F^2 - \|\mathbf{W}^{(s+1)} - \mathbf{W}^*\|_F^2 - \|\widetilde{\mathbf{W}}^{(s+1)} - \widetilde{\mathbf{W}}^*\|_F^2 \geq \eta \overline{L}(\mathbf{W}^{(s)}, \widetilde{\mathbf{W}}^{(s)}) - 2\eta\epsilon.$$

Summing the inequality yields

$$\sum_{s=T_1}^{t} \overline{L}(\mathbf{W}^{(s)}, \widetilde{\mathbf{W}}^{(s)}) \leq \frac{\|\mathbf{W}^{(T_1)} - \mathbf{W}^*\|_F^2 + \|\widetilde{\mathbf{W}}^{(T_1)} - \widetilde{\mathbf{W}}^*\|_F^2 + 2\eta\epsilon(t - T_1 + 1)}{\eta}.$$

Dividing both sides by $t - T_1 + 1$ and setting $t = T$ gives

$$\frac{1}{T - T_1 + 1} \sum_{s=T_1}^{t} \overline{L}(\mathbf{W}^{(s)}, \widetilde{\mathbf{W}}^{(s)}) \leq \frac{\|\mathbf{W}^{(T_1)} - \mathbf{W}^*\|_F^2 + \|\widetilde{\mathbf{W}}^{(T_1)} - \widetilde{\mathbf{W}}^*\|_F^2}{\eta(T - T_1 + 1)} + 2\epsilon \leq 3\epsilon.$$

$\square$

### D.3 Downstream Task Performance

Recall that after the pre-training stage on the training data at time $T$, the signal learning and noise memorization satisfy

$$\max_r A_r^{(T)} = \widetilde{\Omega}(1),$$
$$\max_r \Psi_{r,i}^{(T)} = \widetilde{O}(1/\sqrt{n}) \text{ for } i \in [n].$$

Then, on the downstream task, the corresponding embedding can be calculated as follows:

$$h_r(\mathbf{x}_{\text{test}}^{(1)}) = \sigma(\langle \mathbf{w}_r^{(T)}, \mathbf{x}_{\text{test}}^{(1)} \rangle) = \widetilde{\Omega}(1/\sqrt{d}),$$
$$h_r(\mathbf{x}_{\text{test}}^{(2)}) = \sigma(\langle \mathbf{w}_r^{(T)}, \mathbf{x}_{\text{test}}^{(2)} \rangle) = \widetilde{O}(1/\sqrt{dn}).$$

Then, it is straightforward to check that the embedding of a finite size of samples during the fine-tuning stage is linearly separable. Thus, the downstream task performance follows $L_{\mathcal{D}_{\text{test}}}(T^*) = o(1)$.

## E  Additional Experimental Details

We implement our methods using PyTorch. For the software and hardware configurations, we ensure consistent environments for each dataset. We run all the experiments on Linux servers with NVIDIA V100 graphics cards and CUDA 11.2, completing them within one hour.

