# OpenReview forum: "On the Comparison between Multi-modal and Single-modal Contrastive Learning"
_NeurIPS.cc/2024/Conference — NeurIPS 2024 poster_

### Official Review · Reviewer_nN5z · 2024-07-06

**Soundness:** 2
**Presentation:** 1
**Contribution:** 2
**Rating:** 5
**Confidence:** 2

**Summary:**

In this paper, the authors performed theoretical analysis between single-modal and multi-modal contrastive learning. The authors proved theorems that indicates single-modal contrastive learning tend to perform worse on test datasets after converging on the training set, while multi-modal contrastive learning tend to perform better on the test set. They also conducted small-scale simulated experiments that confirmed their theoretical results.

**Strengths:**

The paper studies the important topic of contrastive learning, and put effort into creating a theoretical understanding of the difference between single-modal and multi-modal contrastive learning. They also conducted small-scale simulated experiments that confirmed their theoretical results.

**Weaknesses:**

There are many problems with the proof:

1. The starting point of the proof is Assumption 4.1. However, Assumption 4.1 contains many issues that puts the soundness and correctness of the entire proof as well as the usefulness of the results into question:

(a) Incorrect use of big-O-like notations. Big-O notations are formally defined only as sets of functions with variables, but in Assumption 4.1, the authors used big-O-like notation to bound constants, which doesn't make sense.

(b) Assumption (1) is extremely unrealistic. It basically requires that $d>n^2$, meaning that the input data's dimension is larger than the square of the number of data points we have in the dataset. In any real world scenario, your data's input dimension will not be anywhere close to the square of the size of your dataset. For example, OpenAI CLIP is trained with $n=400M$ data instances, and the instance dimension is in the order of $d<2\times 10^5$, so in this case, $d < \frac{1}{10^{11}}n^2$.

(c) Undefined variable $\sigma_0$. Technically $n$ and $\eta$ is also never formally defined in text, but we can guess that they mean the size of the dataset and learning rate respectively.

(d) It is also unclear how assumption (6) distinguishes between the single-modal and multi-modal features, as the assumption only applies to the first modality's SNR.

2. Theorem 4.2 and 4.3, the main results of the paper, are not well-defined. $\epsilon$ is supposed to be a vector according to Eq(5), but the main result of Theorem 4.2 says that the train loss (which is scalar) is less than $\epsilon$.

3. Confusing definition of the loss derivatives in Eq (8). Are these definitions for $l_i^{(t)}$ or for $l'^{(t)}_{i}$? These expressions does not look like derivatives.

4. Lemma 5.1 is not proven anywhere, but is then directly used in proving Lemma C.3 and Lemma C.7.

5. The overall proof is organized poorly. It's very difficult to understand what each Lemma is trying to achieve and how are the Lemmas connected to each other, especially between the main text and the Appendix. For example, it is very hard to find where Lemma 5.2 (in the main paper) is proved in the Appendix. Perhaps the authors should add references in the main paper to the appendix (for example, state after Lemma 5.2 "proof in Appendix X.X").

6. The proof roadmaps in Section 5 does not lead to conclusions in Theorems mentioned in Section 4. For example, the roadmap in Section 5.1 does not explicitly tell us how we end up with Theorem 4.2.

7. In proof of Lemma B.2, on Line 560, it is unclear what "RHS" is referring to. Is it the RHS of the last equation on the previous page? Or is it the one immediately before it on line 560? If it is the latter case, you shouldn't be allowed to "set" it to $\delta\/m$ since $\delta$ already appears in the expression before.

In addition to problems in the proof, it is also unclear how the "noise memorization" and "signal learning" evaluation numbers are defined in the experiment in Figure 1.

**Questions:**

1. Please address the problems in the weakness section.

2. Does your theorems provide any new insight into future research contrastive learning? Any actionable suggestions?

**Limitations:**

Discussion is adequate.

---

> ### Author Rebuttal · Authors · 2024-08-07
>
> We would like to thank the reviewer for their detailed and constructive feedback on our paper. Below, we provide detailed responses to the main comments
>
> ---
>
> **W1(a)** Incorrect use of big-O-like notations.
>
> **R1(a)** We appreciate the reviewer's attention to the details of our notation. We have followed standard usage of big-O notation, such as in [1,2,3,4]. Big-O notation is indeed a mathematical tool used to describe the asymptotic behavior of functions. While it is most commonly used to describe functions in terms of variables, it can also be used to describe constant bounds. For instance, the notation $O(1)$ is widely accepted and used to indicate a constant upper bound regardless of the size of the input.
>
> ---
>
> **W1(b)** $d>n^2$ is unrealistic.
>
> **R1(b)** We appreciate the reviewer's concern regarding the assumption $d> n^2$. Our work seeks to provide a theoretical understanding of the difference between multi-modal and single-modal contrastive learning in the over-parameterization regime. The high dimensional setting we adopted ensures sufficient over-parameterization, as the number of trainable parameters is $d \times m$. This setting is common in theoretical studies to simplify analysis and highlight key phenomena. Similar high-dimensional settings have been adopted in relevant prior works [3,4,5,6].
>
> ---
>
> **W1(c)** Undefined variable σ_0, n, η.
>
> **R1(c)** Thank you for bringing this to our attention. $\sigma_0$ represents the initialization strength of the weights of the neural network, where $\mathbf{w}^{(0)}_r \sim \mathcal{N}(0, \sigma^2_0 I)$. $n$ is the number of training samples and $\eta$ is the learning rate. We will ensure that these variables are explicitly defined and clearly explained in the revised version of the manuscript.
>
> ---
>
> **W1(d)**  Unclear how assumption (6) distinguishes between the single-modal and multi-modal features.
>
> **R1(d)** Assumption (6) specifically applies to the first modality's SNR. Assumption (7), on the other hand, specifies the relationship between the signal strengths of the two modalities, given that we assume the noise scale is the same across both modalities (as stated on Line 123). Together, assumptions (6) and (7) establish the requirements on the SNR for the second modality, thereby distinguishing the roles of single-modal and multi-modal features.
>
> ---
>
> **W2** ϵ is a vector according to Eq(5), but is scalar in Theorem 4.2.
>
> **R2** In our paper, we use bold-faced letters to denote vectors and matrices, while non-bold letters represent scalars (line 106). Therefore, the scalar $\epsilon$ and the vector $\boldsymbol{\epsilon}$ are distinct variables. We will ensure to clarify this notation in the revised version of the manuscript to avoid any confusion.
>
> ---
>
> **W3**  Confusing definition of the loss derivatives in Eq (8).
>
> **R3** We use $\ell'^{(t)}_i$ to denote the 'effective' loss derivative, which is a component of the gradient of $L$ with respect to $L$ with respect to $\mathbf{w}_r^{(t)}$. The complete form of gradient is given in Eq. (10) in the main paper. We will ensure to clarify this notation in the revised manuscript to avoid any confusion.
>
> ---
>
> **W4** Lemma 5.1 is not proven.
>
> **R4** Lemma 5.1 is derived directly from the weight decomposition (Eq 11) and gradient descent training (Eq 7). Similar results can be found in the literature [3,7]. We will include the proof of Lemma 5.1 in the appendix of the revised manuscript to ensure completeness and clarity. Thank you for pointing this out.
>
> ---
>
> **W5**: Hard to find where Lemma 5.2 is proved in the Appendix.
>
> **R5**: Thank you for the comments. We will make sure to restructure the proof to improve the clarity and add explanations for each lemma. For the proof of Lemma 5.2, it can be found in Appendix C.1.3. Apart from the contents of the appendix which we have already provided, we will add references to Lemma 5.2.
>
> ---
>
> **W6**: The roadmap in Section 5.1 does not explicitly tell us how we end up with Theorem 4.2.
>
> **R6**: We would like to highlight that proof roadmaps are to provide *major* steps and lemmas (rather than all of them) that lead to a final conclusion. And we leave the remaining part in the appendix given the limited space in the main paper. In our revised version, we will add more explanations by including the remaining proof sketches that lead to Theorems in Section 4.
>
> Here we make an explanation using single-modal learning (Section 5.1) as an example. After we have shown convergence of the dynamics in the second stage, we can show the maximum signal learning is on the order of $\tilde O(1/\sqrt{n})$, while maximum noise memorization is on the constant order (Lemma C.17). This means for downstream tasks, the embedding of a test data would be dominated by the noise, overshadowing the signal. Thus the embedding is not linearly separable, which gives a constant test loss.
>
> ---
>
> **W7**:  Line 560, it is unclear what "RHS" is referring to.
>
> **R7**: the RHS denotes RHS of the inequality
>
> $ \mathbb{P} \big( |\langle \mathbf{w}^{(0)}_{r} , \boldsymbol{\mu} \rangle| \leq c \big) \leq 2 \sqrt{1 - \exp\big(  - \frac{2 c^2}{\sigma_0^2 \\| \boldsymbol{\mu} \\|_2^2 \pi} \big)}$
>
> The $\delta$ here is defined in line 558 of Lemma B.2. In the proof of Lemma B.2, it is the first time to use $\delta$, thus it is valid. Similarly, the $\delta$s in Lemma B.3, B.4, B.5 are all defined within the scope of the respective lemma.
>
> ---
>
> **W8**:It is also unclear how the "noise memorization" and "signal learning" are defined in the experiment in Figure 1.
>
> **R8**: The signal learning and noise memorization refer to the coefficients $\gamma_r^{(t)}$ and $\rho_{r,i}^{(t)}$, which we have already explicitly defined in Line 214. In the experiments, we track the coefficients along the optimization path to plot the results.
>
> ---
>
> **Q**: Does your theorems provide any new insight into future research contrastive learning?
>
> **A**: Please refer to the global rebuttal

---

> > ### Comment · Reviewer_nN5z · 2024-08-07
> > **Additional Discussion**
> >
> > Thank you so much for your replies!
> >
> > I still have some remaining concerns about Lemma B.2. Specifically, there still seems to be a double-definition of $c$ and $\delta$. You first defined $c =100n∥µ∥_2σ^{−1}_ξd^{−1}\sqrt{8log(6n/δ)}$ on line 560, so $c$ is defined depending on $\delta$; but then, you set RHS (which is an expression containing $c$) directly as $\frac{\delta}{m}$, which shouldn't be allowed.
> >
> > The version available in the Appendix is too condensed for readers to fully grasp the details. Can you please provide a line-by-line detailed proof of Lemma B.2? This will help me better assess the correctness of the proofs too. Thank you!

---

> ### Author Response · Authors · 2024-08-07
>
> **References**
>
> [1] Allen-Zhu, Zeyuan, et al. A convergence theory for deep learning via over-parameterization. In International conference on machine learning. 2019.
>
> [2] Allen-Zhu, Zeyuan. Learning and generalization in overparameterized neural networks, going beyond two layers. In Advances in neural information processing systems. 2019.
>
> [3] Cao, Yuan, et al. Benign overfitting in two-layer convolutional neural networks. In Advances in neural information processing systems. 2022.
>
> [4] Meng, Xuran, et al. Benign Overfitting in Two-Layer ReLU Convolutional Neural Networks for XOR Data. In International Conference on Machine Learning. 2024.
>
> [5] Wen, Z. and Li, Y., 2021, July. Toward understanding the feature learning process of self-supervised contrastive learning. In International Conference on Machine Learning (pp. 11112-11122). PMLR.
>
> [6] Xu, Z., Wang, Y., Frei, S., Vardi, G., & Hu, W. (2023). Benign overfitting and grokking in relu networks for xor cluster data. ICLR 2024.
>
> [7] Kou, Yiwen, et al. "Benign overfitting in two-layer ReLU convolutional neural networks." International Conference on Machine Learning. PMLR, 2023.

---

> ### Author Response · Authors · 2024-08-08
> **Detailed Response to Follow-Up Question on Lemma B.2**
>
> Thank for your the follow-up question. Below, we provide a line-by-line proof for Lemma B.2.
>
> *Proof of Lemma B.2.* By Lemma B.1 (anti-concentration result for Gaussian random variable), and because $\langle \mathbf{w}^{(0)}_r, \mathbf{\mu} \rangle \sim \mathcal{N}(0, \sigma_0^2 \| \mathbf{\mu} \|^2_2)$, we can show
>
> $$
>     \mathbb{P} \big( |\langle \mathbf{w}^{(0)}_{r} , \mathbf{\mu} \rangle| \leq c \big) \leq 2 \sqrt{1 - \exp\big(  - \frac{2 c^2}{\sigma_0^2 \| \mathbf{\mu} \|_2^2 \pi} \big)}.
> $$
>
>
> We first set $c = 100 \cdot \text{SNR} \sqrt{\frac{8\log(6n/\delta)}{d}} n$ and plug it into the RHS of the above inequality, which becomes
>
> $$\text{RHS} = 2 \sqrt{1 - \exp\big(- \frac{16\times100^2 \cdot \log(6n/\delta) n^2}{\sigma_0^2 \sigma_\xi^2 d^2 \pi} \big)}.$$
>
> Then we can verify that when $d$ satisfies the condition on Line 561, i.e., $d \geq \frac{400n}{\sigma_0 \sigma_\xi} \sqrt{\frac{\log(6n/\delta)}{-\pi \log(1- \delta^2/(4m^2))}}$, we can verify that
>
> $$\text{RHS} \leq \frac{\delta}{m}.$$
>
> This suggests for a single neuron $r \in [m]$, we have
>
> $$\mathbb{P}( |\langle \mathbf{w}_r^{(0)}, \mathbf{\mu} \rangle| \leq c ) \leq \frac{\delta}{m}.$$
>
> Taking the union bound for all the neurons $r \in [m]$, we have
>
> $$ \mathbb{P}  \big( \cup\_{r =1}^m \{ | \langle \mathbf{w}\_r^{(0)}, \mathbf{\mu} \rangle | \leq c \} \big)
> \leq \sum_{r=1}^m \mathbb{P}( |\langle \mathbf{w}_r^{(0)}, \mathbf{\mu} \rangle| \leq c ) = \delta.
> $$
>
> This then implies that
>
>
> $$\mathbb{P}( \cap_{r =1}^m \{|\langle \mathbf{w}_r^{(0)}, \mathbf{\mu} \rangle| \geq c \} ) \geq 1 -\delta.$$
>
> This concludes that with probability at least $1-\delta$, for all neurons $r \in [m]$, we have $|\langle \mathbf{w}_r^{(0)}, \mathbf{\mu}\rangle| \geq c = 100 \cdot \text{SNR} \sqrt{\frac{8\log(6n/\delta)}{d}} n$.
>
> We hope the above line-by-line proof resolves your question and we believe the wording "set RHS to $\delta/m$" is what has caused the confusion. We will remove such wroding and make the proof clearer in the revised version.
>
> If you have any further questions, we are more than willing to address them. If you find that our responses have sufficiently addressed your concerns and questions, please consider increasing your score. Thank you for your time.

---

> > ### Comment · Reviewer_nN5z · 2024-08-09
> > **Continued Discussion**
> >
> > Thank you for your response! Your line-by-line proof of Lemma B.2. is much clearer than what was available in the paper, and this allowed me to verify that the proof of Lemma B.2. is correct, and I continued to verify the correctness of the proofs afterwards that depends on it.
> >
> > One problem I found is the inconsistency of scopes of symbols. There are global definitions/assumptions of symbols from the main paper, then some additional assumptions/definitions are made from the Lemma statement, which makes it confusing as readers are unsure which assumption still holds within the proofs of each Lemma. For example, the symbol "d" is bounded within Assumption 4.1, but is then again given a lower bound in Lemmas B2,B4,B5,C3,C5,C6, etc. The lower bounds does not have a clear relative size ordering, and is not necessarily bounded by Assumption 4.1 (such as the assumption of lower bound of d from Lemma C6). Technically, within each lemma's proof, the statements should assume the lower bound of d stated in the Lemma, not Assumption 4.1, unless explicitly stated. Therefore, for each following lemma that depends on a previous lemma, the lower bound assumption on d must be stronger than the one from the previous lemma. There are also different assumptions on the size n (lemma C4, lemma C10)
> > Therefore, can you (1) clarify the scopes of Assumptions/Definitions of all variables, and (2) explain why the earlier lemmas can be used in the proofs of later lemmas without proof of tighter/equal bound assumptions?
> >
> > Also, regarding R6, it is not trivial to prove Lemma 5.2 from Lemma C.17, and I am unable to find the intermediate steps of this proof. Similarly, there needs to be proof of Lemma 5.1 from Lemma C.11, and steps of the proof of Theorem 4.2 from Lemma 5.1 and 5.2. If there is no space in the main paper, you should include them in the appendix at least. (same situation appear for Lemma D.* and theorem 4.3 and Lemma 5.3/5.4)
> >
> > An additional comment/suggestion: the current full proof is very poorly written and poorly organized. They are extremely difficult to read. Some of the variable definitions/assumptions occurred more than 20 pages ago, and there should be reminders of the definitions/assumptions. The current proof writing quality would not be admissible to any mathematical journals where reviewers actually check the correctness of your proofs. If everything is written step-by-step and well-defined like the Lemma B.2. proof that you rewrited above, the writing quality would be good enough to allow reviewers to verify the correctness of your proofs.

---

> > > ### Author Response · Authors · 2024-08-09
> > >
> > > Thank you for your detailed feedback. We appreciate the opportunity to clarify the assumptions and structure of our proofs.
> > >
> > > **Global and Local Assumption**:
> > >
> > > Assumption 4.1 is indeed a global assumption that applies to the entire paper. The assumptions made within specific lemmas do not conflict with this global assumption. In other words, the assumptions in specific lemmas can be weaker but are still compatible with Assumption 4.1. If there is any uncertainty about which assumption should be applied within a lemma, Assumption 4.1 should be considered as the default.
> > >
> > > **Regarding the Assumption on $d$**:
> > >
> > > Assumption 4.1 provides a unified lower bound that encompasses the requirements from various lemmas, such as B.2, B.4, B.5, C.3, C.5, and C.6. The lower bound on $d$ in Lemma C.6 is indeed bounded by Assumption 4.1.
> > >
> > > Specifically, the condition on $d$ in Lemma C.6 can be shown to satisfy Assumption 4.1:
> > >
> > > $   \sqrt{\frac{300 \log(6n^2/\delta)}{-\log(1-\delta^2/4n^2) \pi \sigma_0^2\sigma_\xi^2}}
> > > = \tilde{\Theta}(1/(\sigma_0 \sigma_\xi ) )
> > > $
> > >
> > > which is consistent with Assumption 4.1.
> > >
> > > **Inheritance of Conditions in Lemmas**:
> > >
> > > When a lemma is used in the proof of a subsequent lemma, the conditions from the earlier lemma are inherited. This ensures that the assumptions remain consistent and valid throughout the proofs.
> > >
> > >
> > > **Proof Details and Organization**:
> > >
> > > Regarding Lemma 5.2 and Theorem 4.2, Lemma 5.2 ensures that until convergence of the dynamics in the second stage, the maximum signal learning is on the order of $\tilde{O}(n^{-1/2})$, while the maximum noise memorization remains constant (as shown in Lemma C.17). The results in Theorem 4.2 are derived through the arguments provided in Section C.3.
> > >
> > > In the appendix, we have organized the material using section and subsection titles to provide a clear structure. While some steps in the proofs are streamlined, this approach aligns with standard literature in deep learning theory [1, 2, 3]. We are more than willing to address any specific questions or concerns you may have.
> > >
> > > We hope this clarification resolves your concerns. If you have any further questions or suggestions, please feel free to ask.
> > >
> > > [1] Allen-Zhu, Zeyuan, et al. A convergence theory for deep learning via over-parameterization. ICML 2019.
> > >
> > > [2] Cao, Yuan, et al. Benign overfitting in two-layer convolutional neural networks. NeurIPS 2022.
> > >
> > > [3]  Xu, Z., Wang, Y., Frei, S., Vardi, G., & Hu, W. (2023). Benign overfitting and grokking in relu networks for xor cluster data. ICLR 2024.

---

> > > > ### Comment · Reviewer_nN5z · 2024-08-10
> > > > **Scores increased**
> > > >
> > > > Thank you very much for your detailed response! With your clarifications on the Assumptions, I was able to verify the correctness of more Lemmas in the Appendix. Therefore, I believe that your overall deductions are likely correct, and I raised my scores to marginally positive.
> > > >
> > > > I still believe that the proofs need to include more details and the assumptions need to be clearly stated, such as "inheritance of conditions in lemma" that you mentioned above, to make verification of correctness by reviewers possible.

---

> > > > > ### Author Response · Authors · 2024-08-11
> > > > >
> > > > > Thank you very much for your positive feedback and for taking the time to review our clarifications. We are glad that our response helped in verifying the correctness of the lemmas.
> > > > >
> > > > > We appreciate your suggestion to include more details in the proofs and to clearly state assumptions. In our revised version, we will ensure that:
> > > > >
> > > > > - Assumptions and conditions are explicitly stated and clearly linked across lemmas.
> > > > > - The inheritance of conditions is made explicit, so readers can easily track how assumptions carry forward throughout the proofs.
> > > > > - Additional details are included in the proofs to make the logical steps more transparent and easier to follow.
> > > > >
> > > > > Thank you again for your constructive feedback and for increasing your score.

---

### Official Review · Reviewer_7UFF · 2024-07-12

**Soundness:** 3
**Presentation:** 3
**Contribution:** 3
**Rating:** 6
**Confidence:** 5

**Summary:**

This paper presents a theoretical framework to understand the differences between multi-modal and single-modal contrastive learning approaches. It emphasizes the impact of signal-to-noise ratio (SNR) on the generalizability of these learning methods in downstream tasks. The authors argue that multi-modal learning, through the cooperation of modalities, can achieve better feature learning and performance in downstream tasks compared to single-modal learning.

**Strengths:**

- This paper explore an interesting problem. As to why multi-modal contrastive learning might be more effective than its single-modal, this paper provides a feature learning theory framework for analyzing differences between multi-modal and single-modal contrastive learning which is valuable for the machine learning community.
- This paper delves into the nuanced dynamics of multi-modal and single-modal contrastive learning by employing a data generation model that incorporates signal and noise, thereby enabling an in-depth study of the optimization trajectories under gradient descent.
- The authors notably identify the signal-to-noise ratio (SNR) as a pivotal determinant of the generalizability across learning methods, and compellingly demonstrates that the orchestrated interplay between modalities in multi-modal contrastive learning fosters superior generalization in downstream tasks, underscoring the importance of harmonizing diverse data streams for effective feature learning.

**Weaknesses:**

- **Strong assumptions made in the theoretical analysis** The analysis relies on some strict assumptions, such as specific signal-to-noise ratios and network initialization conditions, which might not hold in real-world applications.
- **Some influential previous works are not introduced** such as UMT/UME[1], QMF[2], MMParato[3], and ReconBoost[4].

[1] On Uni-Modal Feature Learning in Supervised Multi-Modal Learning. ICML 2023.

[2] Provable Dynamic Fusion for Low-Quality Multimodal Data. ICML2023

[3] MMPareto: Boosting Multimodal Learning with Innocent Unimodal Assistance. ICML 2024

[4] ReconBoost: Boosting Can Achieve Modality Reconcilement. ICML 2024.

**Questions:**

The theoretical analysis is predicated on rather strong assumptions. If the volume of data and the complexity of the models are significantly increased, would conclusions still be applicable?

---

> ### Author Rebuttal · Authors · 2024-08-07
>
> We would like to thank the reviewer for their constructive feedback and thoughtful comments on our paper. We appreciate the opportunity to address the points raised and clarify our contributions. Below, we provide detailed responses to the main comments and questions.
>
> ---
>
> **W1**: Strong assumptions made in the theoretical analysis The analysis relies on some strict assumptions, such as specific signal-to-noise ratios and network initialization conditions, which might not hold in real-world applications.
>
> **R1**: The data model adopted in this work aligns with related works in the field. Specifically, we use small weight initialization from a Gaussian distribution, ensuring that the network with gradient descent can effectively perform feature learning, as demonstrated in [a,b]. If large initialization were used, the training might fall into the lazy training regime or neural tangent kernel (NTK) regime, which has different learning dynamics and generalization properties [c,d]. The specific SNR values were chosen to create a scenario where we can make a meaningful comparison between single-modal and multi-modal contrastive learning. Similar strategies have been adopted in prior works [e,f].
>
> ---
>
> **W2**: Some influential previous works are not introduced such as UMT/UME[1], QMF[2], MMParato[3], and ReconBoost[4].
>
> **R2**: We thank the reviewer for pointing out these valuable references. Indeed, the mentioned works provide significant theoretical and empirical insights into multimodal learning:
>
> [1] "On Uni-Modal Feature Learning in Supervised Multi-Modal Learning" (UMT/UME) explores feature learning in multimodal contexts, although its setting and claims differ from ours.
>
> [2] "Provable Dynamic Fusion for Low-Quality Multimodal Data" (QMF) focuses on a popular multimodal fusion framework from a generalization perspective.
>
> [3] "MMPareto: Boosting Multimodal Learning with Innocent Unimodal Assistance" identifies previously ignored gradient conflicts between multimodal and unimodal learning objectives through optimization analysis.
>
> [4] "ReconBoost: Boosting Can Achieve Modality Reconcilement" explores a novel multimodal alternating learning paradigm that aims to reconcile the exploitation of unimodal features with the exploration of cross-modal interactions.
>
> These studies are relevant to our work, and we will cite them and add a discussion in our revised version to provide a more comprehensive overview of the related literature and to position our contributions in the broader context of multimodal learning research.
>
> ---
>
> **Q**: The theoretical analysis is predicated on rather strong assumptions. If the volume of data and the complexity of the models are significantly increased, would conclusions still be applicable?
>
> **A**: Our theoretical analysis indeed can cover scenarios with large data sizes (when sample size $n$ and dimension $d$ increases). This ensures that our conclusions about training dynamics and generalization hold even with an increase in the volume of data.
>
> We believe that the main insights regarding the benefits of modality cooperation and the importance of signal-to-noise ratio (SNR) are likely to extend to more complex data distributions (such as non-linearly separable) as well as more complex model architecture (such as transformers). The core principles of multi-modal learning, such as leveraging complementary information from different modalities to enhance signal learning, are not inherently tied to the simplicity of the data model. We acknowledge that more complex scenarios require further investigation and we consider this an interesting direction for future work.
>
> ---
>
> **References**
>
> [a] Cao, Yuan, et al. "Benign overfitting in two-layer convolutional neural networks." Advances in neural information processing systems 35 (2022): 25237-25250.
>
> [b] Suzuki, Taiji, et al. "Feature learning via mean-field langevin dynamics: classifying sparse parities and beyond." Advances in Neural Information Processing Systems 36 (2024).
>
> [c] Jacot, Arthur, Franck Gabriel, and Clément Hongler. "Neural tangent kernel: Convergence and generalization in neural networks." Advances in neural information processing systems 31 (2018).
>
> [d] Chizat, Lenaic, Edouard Oyallon, and Francis Bach. "On lazy training in differentiable programming." Advances in neural information processing systems 32 (2019).
>
> [e] Wen, Zixin, and Yuanzhi Li. "Toward understanding the feature learning process of self-supervised contrastive learning." International Conference on Machine Learning. PMLR, 2021.
>
> [f] Zou, Difan, et al. "Understanding the generalization of adam in learning neural networks with proper regularization." arXiv preprint arXiv:2108.11371 (2021).

---

> > ### Comment · Reviewer_7UFF · 2024-08-13
> >
> > I appreciate the author's great efforts in the rebuttal phase. I will keep my initial score as a final decision.

---

> > > ### Author Response · Authors · 2024-08-13
> > >
> > > Thank you for your positive support and for taking the time to review our rebuttal. We appreciate your thoughtful consideration throughout the review process.

---

### Official Review · Reviewer_bcfY · 2024-07-13

**Soundness:** 3
**Presentation:** 3
**Contribution:** 3
**Rating:** 6
**Confidence:** 3

**Summary:**

This paper provides a theoretical analysis comparing single-modal and multi-modal contrastive learning. The authors develop a unified framework to analyze the optimization dynamics and generalization capabilities of both approaches. Key findings include:

- Both single-modal and multi-modal contrastive learning can achieve small training error.
- Multi-modal contrastive learning generalizes better to downstream tasks compared to single-modal learning.
- The advantage of multi-modal learning comes from cooperation between modalities and higher quality data in the second modality.

The analysis is based on a two-stage optimization process and uses a signal-noise data generation model. Theoretical results are supported by synthetic experiments.

**Strengths:**

- Develops a unified framework to analyze both single-modal and multi-modal contrastive learning, allowing for direct comparisons.

- Provides detailed theoretical analysis, including convergence guarantees and generalization bounds.

- Identifies key factors (signal-to-noise ratio, cooperation between modalities) that explain the superior performance of multi-modal learning.

- Supports theoretical findings with synthetic experiments that align well with the analysis.

**Weaknesses:**

- The paper could benefit from more intuitive explanations of the key theoretical results to make them more accessible to a broader audience. For example, can you provide intuition for why the cooperation between modalities leads to better signal learning in the multi-modal case? How might this insight be leveraged to improve single-modal contrastive learning?

- The paper lacks discussion on how the insights could be applied to improve existing contrastive learning methods or guide the development of new approaches. Do your theoretical insights suggest any practical strategies for improving multi-modal contrastive learning, such as ways to select or preprocess data to increase the effective signal-to-noise ratio?

- The theoretical setup and assumption of the linear data generation model are somewhat simple and restricted. Do you expect the main insights to hold for more complex data distributions?

- The experimental validation is limited to synthetic data. Including experiments on real-world datasets, even if simplified, would strengthen the paper's impact.

**Questions:**

Line 205: “On the contrary, augmentation often maintains the same SNR as the original data, so single-modal learning hardly benefits from the augmentation and can only memorize the noise from the data.” This claim significantly contradicts empirical experiments such as SimCLR and MoCo. How would you justify this?

**Limitations:**

See weaknesses.

---

> ### Author Rebuttal · Authors · 2024-08-07
>
> We would like to thank the reviewer for their constructive feedback and insightful comments on our paper. We appreciate the opportunity to address the points raised and clarify our contributions. Below, we provide detailed responses to the main comments and questions.
>
> ---
>
> **W1**: Can you provide intuition for why the cooperation between modalities leads to better signal learning in the multi-modal case? How might this insight be leveraged to improve single-modal contrastive learning?
>
> **R1**: Both the single-modal and multi-modal contrastive learning aim to maximize the similarity between positive pairs while minimizing the similarity between the negative ones. The difference is single-modal learning forms positive pairs by data augmentation while multi-modal learning forms positive pairs by the correspondence between the two modalities. Usually, the data augmentation does not change the signal-to-noise ratio (SNR), while the two modalities can have different SNR. Due to the use of contrastive loss, the learning of signals in one modality highly depends on the signal learning in the other modality. Thus, when the other modality has a higher SNR, the learning of the first modality is lifted by aligning with the second modality. This cooperative learning leads to better overall signal learning compared to single-modal contrastive learning.
>
> To improve single-modal contrastive learning, one could potentially simulate the effect of multi-modal learning by augmenting the single modality with additional synthetic views that enhance the signal-to-noise ratio, thereby improving the overall learning process.
>
> ---
>
> **W2**: Do your theoretical insights suggest any practical strategies for improving multi-modal contrastive learning?
>
> **R2**:  Please refer to the global rebuttal
>
> ---
>
> **W3**: Do you expect the main insights to hold for more complex data distributions?
>
> **R3**: While our current analysis is based on a linear data generation model, we believe that the main insights regarding the benefits of modality cooperation and the importance of signal-to-noise ratio (SNR) are likely to extend to more complex data distributions. The core principles of multi-modal learning, such as leveraging complementary information from different modalities to enhance signal learning, are not inherently tied to the simplicity of the data model.
>
> To ensure broader applicability, future work will focus on validating these insights with non-linear and more realistic data models. This will involve experimenting with a variety of data distributions and network architectures to confirm that the advantages of multi-modal cooperation and improved SNR hold true in more practical and complex scenarios.
>
> ---
>
> **W4**: The experimental validation is limited to synthetic data.
>
> **R4**: We have extended the comparison to realistic image data, ColoredMNIST [1,2], which is a typical benchmark studying the generalization capability under distribution shifts. The task is a 10-class classification that recognizes the number of the colored MNIST images. The two modalities are image and text that describes the images. We implement the multi-modal learning following the practice of [2], where we consider an ideal language encoder that successfully encodes the caption of the images into one-hot labels of colors and digits. For single-modal learning, we follow the implementation of the SimCLR paper to construct a set of augmentations to learn the representations. Then, under a mild and realistic distribution shift, we verify that CLIP archives an out-of-distribution test accuracy of 82.13\%, which outperforms that of SimCLR 12.68\%.
>
> | Model | Train Accuracy | Test Accuracy |
> |-------|----------------|---------------|
> | SimCLR | 88.43% | 12.68% |
> | CLIP   | 87.77% | 82.13% |
>
>
> ---
>
> **Q**: Line 205: “On the contrary, augmentation often maintains the same SNR as the original data, so single-modal learning hardly benefits from the augmentation and can only memorize the noise from the data.” This claim significantly contradicts empirical experiments such as SimCLR and MoCo. How would you justify this?
>
> **A**: Our argument does not contradict the success of single-modal contrastive learning methods such as SimCLR and MoCo. The statement is made in a comparative context with multi-modal contrastive learning. In our analysis, when comparing single-modal to multi-modal contrastive learning, the augmentation in the single-modal case often does not improve the signal-to-noise ratio (SNR) as effectively as having a second, complementary modality would. Therefore, in scenarios where the SNR is not significantly enhanced by augmentation, single-modal learning may perform worse than multi-modal learning.
>
> Having said that, this does not imply that single-modal methods like SimCLR and MoCo fail in general. Empirical evidence shows that these methods are quite effective, particularly when the number of training samples is large and the SNR is sufficiently high. Our theoretical framework can indeed provide guarantees for the success of single-modal contrastive learning under such conditions.
>
>
> ---
>
> **References**
>
>
> [1] Invariant Risk Minimization arXiv 2019.
>
> [2]  Do CLIPs Always Generalize Better than ImageNet Models? arXiv 2024.

---

> > ### Comment · Reviewer_bcfY · 2024-08-08
> >
> > Thanks for the author's response! Most of my concerns have been addressed. One remaining question: could you elaborate more on the experimental setting for ColoredMNIST experiments? What do the training data & test data look like?

---

> > > ### Author Response · Authors · 2024-08-09
> > >
> > > Thank you for your positive feedback and further questions regarding the ColoredMNIST experiment. We are happy to provide additional details:
> > >
> > > - Image: The ColoredMNIST dataset is a variation of the standard MNIST dataset, where each digit is assigned a specific color based on its label. We consider a 10-class classification task in order to better match with the realistic setting. Specifically, we have 10 colors to color 10 digits, and introduce spurious correlations via label noises following the literature.
> > >
> > >   - For the training set, 10% of labels will be clipped to a random class. For images with class '0' (or '1'), they will be colored as red (or green) with a probability of 77.5%, and as another random color with a probability of 22.5%. The coloring scheme introduces a spurious correlation.
> > >   - For the test set, 10% of labels will be clipped to a random class. For images with class '0' (or '1'), they will be colored as green (or red) with a probability of 77.5%, and as another random color with a probability of 22.5%. The coloring scheme can be considered as reversing the training spurious correlations. Therefore, the evaluation on testsets can reflect to what extent the model learns to use the spurious features, i.e., colors, to classify images.
> > >
> > > - Text:  We also consider an ideal language encoder that successfully encodes the captions of the images into one-hot labels of colors and digits.
> > >
> > >
> > >
> > > As a result, we can claim that the effective SNR of invariant features (the shape of the digit) will be degraded under the impact of the injected color. Therefore, the performance of SimCLR may be suboptimal as it cannot effectively utilize the information of the digit's shape. On the other hand, CLIP demonstrates a better capacity for handling this scenario.
> > >
> > > We hope this additional explanation clarifies the experimental setup for the ColoredMNIST experiments. If you have any further questions, please feel free to ask. If our answer has clarified your concerns, we would appreciate it if you could consider reevaluating our submission.

---

> > > > ### Comment · Reviewer_bcfY · 2024-08-12
> > > >
> > > > I thank the authors for their detailed explanation. The newly added MNIST experiments provide great intuition into how multi-modal information benefits over uni-modal information. I'll raise my score to 6.

---

> > > > > ### Author Response · Authors · 2024-08-12
> > > > >
> > > > > Thank you for your positive feedback. We are pleased that our ColoredMNIST experiment provided valuable intuition and helped clarify the benefits of multi-modal information. We appreciate your support and are grateful for your consideration in raising the score.

---

### Author Rebuttal · Authors · 2024-08-07

Dear Reviewers and ACs

Thank you all for your time and constructive feedback! We are truly encouraged to see many positive comments on our work, such as the the *unified framework for an interesting problem* (Reviewer bcfY, Reviewer 7UFF, Reviewer nN5z), *thorough theoretical analysis* (Reviewer bcfY, Reviewer 7UFF), *identification of key factors* (Reviewer bcfY, Reviewer 7UFF), and *supportive simulations* (Reviewer bcfY, Reviewer nN5z).

Your insights have greatly helped us strengthen our manuscript. Here, we provide a response to a common question raised by Reviewer bcfY (W2) and Reviewer nN5z (Q):


**Q**: Do your theoretical insights suggest any practical strategies for improving multi-modal contrastive learning?/ Does your theorems provide any new insight into future research contrastive learning? Any actionable suggestions?

**A**: Our theoretical results highlight that increasing the effective signal-to-noise ratio (SNR) across modalities is crucial for improving multi-modal contrastive learning. This leads to two practical strategies for developing new approaches:

- **Selecting or Generating High-Quality Data Pairs** Ensuring that the signal is strong and well-aligned across modalities can significantly improve the performance. For instance, improving the quality of aligned image-caption samples by filtering out poorly aligned pairs used for training can enhance the overall learning process [1,2,3]

- **Improving SNR for the Text Modality** Enhancing the descriptiveness of the captions can boost the SNR for the text modality, leading to better performance in multi-modal contrastive learning [4,5,6]

Should the reviewers have any further suggestions or wish to discuss any points in more detail, we would be more than delighted to continue our productive exchange. Once again, we deeply appreciate the reviewers' time and valuable comments.

Best regards,

Authors of Submission 555


---

**References**

[1] Schuhmann, Christoph, et al. "Laion-5b: An open large-scale dataset for training next generation image-text models." Advances in Neural Information Processing Systems 35 (2022): 25278-25294.

[2] Nguyen, Thao, et al. "Quality not quantity: On the interaction between dataset design and robustness of clip." Advances in Neural Information Processing Systems 35 (2022): 21455-21469.

[3] Gadre, S.Y., Ilharco, G., Fang, A., Hayase, J., Smyrnis, G., Nguyen, T., Marten, R., Wortsman, M., Ghosh, D., Zhang, J. and Orgad, E., 2024. Datacomp: In search of the next generation of multimodal datasets. Advances in Neural Information Processing Systems, 36.

[4] Santurkar, Shibani, et al. "Is a caption worth a thousand images? a study on representation learning." The Eleventh International Conference on Learning Representations. 2023.

[5] Nguyen, Thao, et al. "Improving multimodal datasets with image captioning." Advances in Neural Information Processing Systems 36 (2024).

[6] Fan, L., Krishnan, D., Isola, P., Katabi, D. and Tian, Y., 2024. Improving clip training with language rewrites. Advances in Neural Information Processing Systems, 36.

---

### Comment · Area_Chair_YS4B · 2024-08-07
**Please Discuss**

Dear Reviewers,

the authors have provided an extensive rebuttal for all reviewers. Please have a look at their responses to see if they address your concerns. If you have further questions, you can now start a discussion with the authors.

Your AC

---

### Decision · Program_Chairs · 2024-09-25

**Decision:**

Accept (poster)

**Comment:**

The authors develop a unified framework to analyze both single-modal and multi-modal contrastive learning. They explore why multi-modal contrastive learning might be more effective than its single-modal counter part. The reviewers noted some concerns about the clarity of the proofs and the underlying assumptions (which are found to be too strong), but the rebuttal resolved the main questions and concerns. The authors are urged to incorporate all suggestions and additional clarifications into the paper or the appendix of the paper. Despite the concerns, the theoretical framework can be valuable for the machine learning community.